# ERROR FEEDBACK FOR MUON AND FRIENDS

**Kaja Gruntkowska, Alexander Gaponov, Zhirayr Tovmasyan & Peter Richtárik**
King Abdullah University of Science and Technology (KAUST)
Thuwal, Kingdom of Saudi Arabia

## ABSTRACT

Recent optimizers like Muon, Scion, and Gluon have pushed the frontier of large-scale deep learning by exploiting layer-wise linear minimization oracles (LMOs) over non-Euclidean norm balls, capturing neural network structure in ways traditional algorithms cannot. Yet, no principled distributed framework exists for these methods, and communication bottlenecks remain unaddressed. The very few distributed variants are heuristic, with no convergence guarantees in sight. We introduce EF21-Muon, the first communication-efficient, non-Euclidean LMO-based optimizer with rigorous convergence guarantees. EF21-Muon supports stochastic gradients, momentum, and bidirectional compression with error feedback–marking the first extension of error feedback beyond the Euclidean setting. It recovers Muon/Scion/Gluon when compression is off and specific norms are chosen, providing the first efficient distributed implementation of this powerful family. Our theory covers non-Euclidean smooth and the more general $(L^0, L^1)$–smooth setting, matching best-known Euclidean rates and enabling faster convergence under suitable norm choices. We further extend the analysis to layer-wise (generalized) smoothness regimes, capturing the anisotropic structure of deep networks. Experiments on `NanoGPT` benchmarking EF21-Muon against uncompressed Muon/Scion/Gluon demonstrate up to $7\times$ communication savings with no accuracy degradation.

## 1 INTRODUCTION

Over the past decade, Adam and its variants (Kingma & Ba, 2015; Loshchilov & Hutter, 2019) have established themselves as the cornerstone of optimization in deep learning. Yet emerging evidence suggests that this dominance may be giving way to a new class of optimizers better suited to the geometry and scale of modern deep networks. Leading this shift are Muon (Jordan et al., 2024) and methods inspired by it–Scion (Pethick et al., 2025b) and Gluon (Riabinin et al., 2025b)–which replace Adam's global moment estimation with layer-wise, geometry-aware updates via *linear minimization oracles* (LMOs) over non-Euclidean norm balls. Though relatively new, these optimizers are already gaining traction–supported by a growing body of theoretical insights, community adoption, and empirical success–particularly in training large language models (LLMs) (Liu et al., 2025; Pethick et al., 2025b; Shah et al., 2025; Thérien et al., 2025; Moonshot AI, 2025).

Despite this momentum, the development of these algorithms remains less mature than that of more established methods. Significant gaps persist–both in theory and practice–that must be addressed to fully realize their potential and make them truly competitive for the demands of ultra-scale learning.

**Scaling Up.** Modern machine learning (ML) thrives on scale. Today's state-of-the-art models rely on massive datasets and complex architectures, often requiring weeks or even months of training (Touvron et al., 2023; Comanici et al., 2025). This scale imposes new demands on optimization methods, which must not only be effective at navigating complex nonconvex landscapes but also efficient in distributed, resource-constrained environments. Since training on a single machine is no longer feasible (Dean et al., 2012; You et al., 2017), distributed computing has become the default. Mathematically, this task is commonly modeled as the (generally non-convex) optimization problem

$$\min_{X \in \mathcal{S}} \left\{ f(X) := \frac{1}{n} \sum_{j=1}^{n} f_j(X) \right\}, \qquad f_j(X) := \mathbb{E}_{\xi_j \sim \mathcal{D}_j} [f_j(X; \xi_j)] \tag{1}$$

Table 1: Summary of convergence guarantees. **Algorithm**: Deterministic = EF21-Muon with deterministic gradients (Algorithm 2), Stochastic = EF21-Muon with stochastic gradients (Algorithms 1 and 3); **Smooth**: ✓ = (layer-wise) smooth setting (Assumptions 3 and 6), ✗ = (layer-wise) generalized smooth setting (Assumptions 4 and 8); **Rate** = rate of convergence to achieve $\min_{k=0,\ldots,K} \mathbb{E}\left[\left\|\nabla f(X^k)\right\|_\star\right] \leq \varepsilon$; **Eucl.** = recovers the state-of-the-art guarantees in the Euclidean case; **Non-comp.** = recovers the state-of-the-art uncompressed guarantees.

| Algorithm | Result | Layer-wise | Smooth | Rate | Eucl. | Non-comp. |
|---|---|---|---|---|---|---|
| Deterministic | Theorem 3 | ✗ | ✓ | $\mathcal{O}\left(\frac{1}{K^{1/2}}\right)$ | ✓ | ✓ |
| | Theorem 14 | ✓ | | | ✓ | ✓ |
| | Theorem 4 | ✗ | ✗ | | ✓ | ✓ |
| | Theorem 17 | ✓ | | | ✓ | ✓ |
| Stochastic | Theorem 5 | ✗ | ✓ | $\mathcal{O}\left(\frac{1}{K^{1/4}}\right)$ | ✓ | ✓ |
| | Theorem 19 | ✓ | | | ✓ | ✓ |
| | Theorem 6 | ✗ | ✗ | | ✓ | ✓ |
| | Theorem 24 | ✓ | | | ✓ | ✓ |

where $X \in \mathcal{S}$ represents the model parameters, $n \geq 1$ is the number of workers/clients/machines, and $f_j(X)$ is the loss of the model ($X$) on the data ($\mathcal{D}_j$) stored on worker $j \in [n] := \{1, \ldots, n\}$. We consider the general heterogeneous setting, where the local objectives $f_j$ may differ arbitrarily across machines, reflecting real-world scenarios such as multi-datacenter pipelines or federated learning (McMahan et al., 2017; Konečný et al., 2016). Here, $\mathcal{S}$ is a $d$-dimensional vector space equipped with an inner product $\langle \cdot, \cdot \rangle : \mathcal{S} \times \mathcal{S} \to \mathbb{R}$ and the standard Euclidean norm $\|\cdot\|_2$. Furthermore, we endow $\mathcal{S}$ with an arbitrary norm $\|\cdot\| : \mathcal{S} \to \mathbb{R}_{\geq 0}$. The corresponding dual norm $\|\cdot\|_\star : \mathcal{S} \to \mathbb{R}_{\geq 0}$ is defined via $\|X\|_\star := \sup_{\|Z\| \leq 1} \langle X, Z \rangle$. The general framework introduced in this work gives rise to a variety of interesting algorithms arising from different norm choices. In matrix spaces, a particularly important class is the family of *operator norms*, defined for any $A \in \mathbb{R}^{m \times n}$ by $\|A\|_{\alpha \to \beta} := \sup_{\|Z\|_\alpha = 1} \|AZ\|_\beta$, where $\|\cdot\|_\alpha$ and $\|\cdot\|_\beta$ are some norms on $\mathbb{R}^n$ and $\mathbb{R}^m$, respectively.

**Communication: the Cost of Scale.** In client-server architectures, coordination is centralized, with workers performing local computations and periodically synchronizing with the coordinator (Seide et al., 2014; Alistarh et al., 2017; Khirirat et al., 2018; Stich et al., 2018; Mishchenko et al., 2019; Karimireddy et al., 2019; Mishchenko et al., 2024). While this distributed design unlocks learning at unprecedented scales, it introduces a critical bottleneck: *communication*. The massive size of modern models places a heavy burden on the channels used to synchronize updates across machines, as each step requires transmitting large $d$-dimensional vectors (e.g., parameters or gradients) over links that can be far slower than local computation (Kairouz et al., 2021). Without communication-efficient strategies, this imbalance makes communication a dominant cost, ultimately limiting the efficiency and scalability of distributed optimization.

**Distributed Muon: Bridging the Gap.** The case for communication-efficient distributed training is clear, as is the promise of Muon for deep learning. The natural question is: can we merge the two? Perhaps surprisingly, this intersection remains largely unexplored. Nonetheless, three recent efforts are worth noting. Liu et al. (2025) propose a distributed variant of Muon based on ZeRO-1 (Rajbhandari et al., 2020). Thérien et al. (2025) show that Muon can be used instead of AdamW as the inner optimizer in DiLoCo. The introduced MuLoCo framework is shown to consistently converge faster than the original DiLoCo (Douillard et al., 2023) when pre-training a 220M parameter transformer language model. In parallel, Ahn et al. (2025) introduce Dion, a Muon-inspired algorithm compatible with 3D parallelism that employs low-rank approximations for efficient orthonormalized updates.

While promising empirically, these approaches *lack any formal theoretical guarantees*. Our goal is to bridge this gap by developing a distributed optimizer leveraging non-Euclidean geometry that both *works in practice* and comes with *strong convergence guarantees*. Our central question is:

*Can we efficiently distribute Muon without compromising its theoretical and practical benefits?*

In this work, we provide an affirmative answer through the following **contributions**:

1. **A framework for compressed non-Euclidean distributed optimization.** We propose EF21-Muon, an LMO-based distributed optimizer based on bidirectionally compressed updates with error feedback (Seide et al., 2014; Richtárik et al., 2021). It is communication-efficient (never sending uncompressed messages) and practical, supporting stochasticity and momentum. Parameterized by the norm in the LMO step, EF21-Muon recovers a broad class of compressed methods, and for spectral norms yields *the first communication-efficient distributed variants of* Muon *and* Scion.

2. **Practical deep learning variant.** The main body of this paper presents a simplified version of EF21-Muon that treats all parameters jointly (Algorithm 1), consistent with standard theoretical exposition. Our main algorithms, however, are designed for and analyzed in a *layer-wise* manner (see Algorithms 2 and 3 for the deterministic and stochastic gradient variants, respectively), explicitly modeling the hierarchical structure of neural networks. This allows us to better align with practice (methods like Muon are applied *per layer*) and to introduce *anisotropic modeling assumptions*.

3. **Strong convergence guarantees.** EF21-Muon comes with strong theoretical guarantees (see Table 1) under two smoothness regimes: non-Euclidean smoothness (Theorems 3 and 5) and non-Euclidean $(L^0, L^1)$–smoothness (Theorems 4 and 6). In both cases, our bounds match the state-of-the-art rates for EF21 in the Euclidean setting, while allowing for potentially faster convergence under well-chosen norms. These results are subsumed by a more general analysis of the full layer-wise methods. In Theorems 14 and 19, we prove convergence under *layer-wise non-Euclidean smoothness* (Assumption 6), and extend this to *layer-wise non-Euclidean $(L^0, L^1)$–smoothness* (Assumption 8) in Theorems 17 and 24. This refined treatment allows us to better capture the geometry of deep networks, leading to tighter guarantees.

4. **Non-Euclidean compressors.** EF21-Muon supports standard contractive compressors as well as a new class of non-Euclidean compressors (Section D), which may be of independent interest.

5. **Strong empirical performance.** Experiments training a `NanoGPT` model on the `FineWeb` dataset systematically compare EF21-Muon with multiple compressors against the uncompressed baseline (Muon/Scion/Gluon) and show that compression reduces worker-to-server communication by up to $7\times$ with no loss in accuracy (Sections 5 and G).

**Outline.** Section 2 introduces the necessary preliminaries and reviews Muon (Jordan et al., 2024), placing it within the broader class of LMO-based optimizers. This naturally raises the central question of our work: how can such methods be distributed efficiently? We highlight the main challenges and motivate compression and error feedback as practical solutions (with deeper motivation and an extensive literature review deferred to Section A). Section 3 presents our proposed method, EF21-Muon. In Section 4, we present convergence results in both deterministic and stochastic settings, under two smoothness regimes: standard (non-Euclidean) and $(L^0, L^1)$–smoothness. Finally, Section 5 provides empirical validation, demonstrating the practical benefits of our approach.

## 2    BACKGROUND

We frame problem (1) in an abstract vector space $\mathcal{S}$. In several of our results, the specific structure of $\mathcal{S}$ does not matter. One may simply flatten the model parameters into a $d \times 1$ vector and view $\mathcal{S}$ as $\mathbb{R}^d$. However, in the context of deep learning, it is often useful to *explicitly model the layer-wise structure* (see Section B). Then, $X \in \mathcal{S}$ represents the collection of matrices $X_i \in \mathcal{S}_i := \mathbb{R}^{m_i \times n_i}$ of trainable parameters across all layers $i \in [p]$ of the network with a total number $d := \sum_{i=1}^p m_i n_i$ of parameters. Accordingly, $\mathcal{S}$ is the $d$-dimensional product space $\mathcal{S} := \bigotimes_{i=1}^p \mathcal{S}_i \equiv \mathcal{S}_1 \otimes \cdots \otimes \mathcal{S}_p$, where each $\mathcal{S}_i$ is associated with the trace inner product $\langle X_i, Y_i \rangle_{(i)} := \mathrm{tr}(X_i^\top Y_i)$ for $X_i, Y_i \in \mathcal{S}_i$, and a norm $\|\cdot\|_{(i)}$ (not necessarily induced by this inner product). We write $X = [X_1, \ldots, X_p]$.

**What is Muon?**  Muon, introduced by Jordan et al. (2024), is an optimizer for the hidden layers of neural networks.[1] For clarity of exposition, let us assume that the parameters $X$ represent a single layer of the network (a full layer-wise description is provided in Section B.1). In this setting, Muon updates $X^{k+1} = X^k - t^k U^k (V^k)^\top$, where $t^k > 0$ and the matrices $U^k, V^k$ are derived from the

---

[1] The first and last layers are typically optimized using other optimizers, such as AdamW (Loshchilov & Hutter, 2019)–see Section B.1 for details.

---

**Algorithm 1** EF21-Muon (simplified)

---

1: **Parameters:** radii $t^k > 0$; momentum parameter $\beta \in (0, 1]$; initial iterate $X^0 \in \mathcal{S}$ (stored on the server); initial iterate shift $W^0 = X^0$ (stored on the server and the workers); initial gradient estimators $G_j^0$ (stored on the workers); $G^0 = \frac{1}{n} \sum_{j=1}^n G_j^0$ (stored on the server); initial momentum $M_j^0$ (stored on the workers); worker compressors $\mathcal{C}_j^k$; server compressors $\mathcal{C}^k$

2: **for** $k = 0, 1, \ldots, K - 1$ **do**

3:      $X^{k+1} = \mathrm{LMO}_{\mathcal{B}(X^k, t^k)}\left(G^k\right)$          *Take LMO-type step*

4:      $S^k = \mathcal{C}^k(X^{k+1} - W^k)$          *Compress shifted model on the server*

5:      $W^{k+1} = W^k + S^k$          *Update model shift*

6:      Broadcast $S^k$ to all workers

7:      **for** $j = 1, \ldots, n$ **in parallel do**

8:          $W^{k+1} = W^k + S^k$          *Update model shift*

9:          $M_j^{k+1} = (1 - \beta)M_j^k + \beta \nabla f_j(W^{k+1}; \xi_j^{k+1})$          *Compute momentum*

10:          $R_j^{k+1} = \mathcal{C}_j^k(M_j^{k+1} - G_j^k)$          *Compress shifted gradient*

11:          $G_j^{k+1} = G_j^k + R_j^{k+1}$

12:          Broadcast $R_j^{k+1}$ to the server

13:      **end for**

14:      $G^{k+1} = \frac{1}{n} \sum_{j=1}^n G_j^{k+1} = G^k + \frac{1}{n} \sum_{j=1}^n R_j^{k+1}$          *Compute gradient estimator*

15: **end for**

---

SVD of the momentum matrix $G^k = U^k \Sigma^k (V^k)^\top$. This update rule is, in fact, a special case of a more general one, based on the norm-constrained *linear minimization oracle* (LMO)

$$X^{k+1} = X^k + t^k \mathrm{LMO}_{\mathcal{B}(0,1)}\left(G^k\right), \tag{2}$$

where $\mathcal{B}(X, t) := \{Z \in \mathcal{S} : \|Z - X\| \leq t\}$ and $\mathrm{LMO}_{\mathcal{B}(X,t)}(G) := \arg\min_{Z \in \mathcal{B}(X,t)} \langle G, Z \rangle$. Muon corresponds to the case where $\|\cdot\| = \|\cdot\|_{2\to 2}$ is the spectral (operator) norm, in which case $\mathrm{LMO}_{\mathcal{B}(0,1)}\left(G^k\right) = -U^k(V^k)^T$. Consequently, its recent analyses (Pethick et al., 2025b; Kovalev, 2025; Riabinin et al., 2025b) have shifted focus to the general form (2). Among them, Pethick et al. (2025b) introduce Scion, which extends the LMO update across layers, and Riabinin et al. (2025b) develop Gluon–a general LMO-based framework that subsumes Muon and Scion as special cases while providing stronger convergence guarantees. We adopt this unifying viewpoint by treating all three algorithms as instances of Gluon, which we use as the umbrella term for the entire class.

**The challenges of distributing the LMO.** Distributing (2) is far from trivial, as the limited literature suggests. Even in the relatively well-structured special case of spectral norms, Muon relies on the Newton–Schulz iteration (Kovarik, 1970; Björck & Bowie, 1971), a procedure requiring dense matrix operations that are incompatible with standard parameter-sharding schemes used in LLM training (Ahn et al., 2025). To illustrate the difficulty, consider a deterministic version of (2), where $G^k$ is replaced by the exact gradient $\nabla f(X^k)$. Applied to problem (1), the iteration becomes

$$X^{k+1} = X^k + \mathrm{LMO}_{\mathcal{B}(0,t^k)}\left(\frac{1}{n} \sum_{j=1}^n \nabla f_j(X^k)\right).$$

The most basic approach to distributing this update consists of the following four main steps:

1. Each worker computes its local gradient $\nabla f_j(X^k)$ at iteration $k$.
2. **w2s:** The workers send their gradients $\nabla f_j(X^k)$ to the central server.
3. The server averages these gradients and computes the LMO update.
4. **s2w:** The server sends $X^{k+1}$ (or $\mathrm{LMO}_{\mathcal{B}(0,t^k)}(\cdot)$) back to the workers.

This scheme involves two potentially costly phases: (1) workers-to-server (= w2s) and (2) server-to-workers (= s2w) communication. As each transmitted object resides in $\mathcal{S}$, every iteration involves exchanging dense, $d$-dimensional data, imposing substantial communication overhead that can quickly overwhelm available resources. This is where compression techniques come into play.

**Compression.** Compression is one of the two main strategies for improving communication efficiency in distributed optimization (the other being *local training* (Povey et al., 2014; Moritz et al.,

2015; McMahan et al., 2017)), extensively studied in the Euclidean regime (Alistarh et al., 2017; Horváth et al., 2022; Richtárik et al., 2021). It is typically achieved by applying an operator $\mathcal{C}$ mapping the original dense message $X$ to a more compact representation $\mathcal{C}(X)$. We work with general *biased* (or *contractive*) compressors.

**Definition 1** (Contractive compressor). A (possibly randomized) mapping $\mathcal{C} : \mathcal{S} \to \mathcal{S}$ is a *contractive compression operator* with parameter $\alpha \in (0, 1]$ if

$$\mathbb{E}\left[\|\mathcal{C}(X) - X\|^2\right] \leq (1 - \alpha)\|X\|^2 \qquad \forall X \in \mathcal{S}. \tag{3}$$

**Remark 2.** *The classical definition of a contractive compressor is based on the Euclidean norm, i.e., $\|\cdot\| = \|\cdot\|_2$ in (3). A canonical example in this setting is the TopK compressor, which retains the $K$ largest-magnitude entries of the input vector. In (3), we generalize this to arbitrary norms for greater flexibility. Section D provides examples of such compressors (to our knowledge, not studied in this context before). Depending on the compression objective, we apply (3) with respect to $\|\cdot\|$, $\|\cdot\|_\star$, or $\|\cdot\|_2$, denoting the respective families of compressors as $\mathbb{B}(\alpha)$, $\mathbb{B}_\star(\alpha)$, and $\mathbb{B}_2(\alpha)$.*

**Error Feedback.** To address the communication bottleneck, a natural approach is to apply biased compressors to transmitted gradients. However, this "enhancement" can result in exponential divergence, even in the simple case of minimizing the average of three strongly convex quadratics (Beznosikov et al., 2020, Example 1). A remedy, *Error Feedback* (EF), was introduced by Seide et al. (2014) and for years remained a heuristic with limited theory. This changed with Richtárik et al. (2021), who proposed EF21, the first method to achieve the desirable $\mathcal{O}(1/\sqrt{K})$ rate for expected gradient norms under standard assumptions. Since then, EF21 has inspired many extensions, including EF21-P (Gruntkowska et al., 2023), a primal variant targeting s2w communication.

For a deeper dive into compression and EF, we refer the reader to Appendices A.1 and A.2.

## 3 NON-EUCLIDEAN DISTRIBUTED TRAINING

Marrying geometry-aware updates of Gluon with the communication efficiency enabled by compression promises a potentially high-yield strategy. Yet, from a theoretical standpoint, their compatibility is far from obvious–nothing a priori ensures that these two paradigms can be meaningfully unified.

Most importantly, it is unclear what kind of descent lemma to use. The analysis of EF21 relies on a recursion involving squared Euclidean norms $\|\cdot\|_2^2$, while LMO-based methods naturally yield descent bounds in terms of first powers of norms $\|\cdot\|$–a structure common to all existing analyses (Kovalev, 2025; Pethick et al., 2025b; Riabinin et al., 2025b). We initially adopted the latter approach, but the resulting guarantees failed to recover those of EF21 in the Euclidean case. The pivot point came from reformulating the update via *sharp operators* (Nesterov, 2012; Kelner et al., 2014). For any $G \in \mathcal{S}$, the sharp operator is defined as $G^\sharp := \arg\max_{X \in \mathcal{S}}\{\langle G, X \rangle - \frac{1}{2}\|X\|^2\}$, which is connected to the LMO via the identity $\|G\|_\star \operatorname{LMO}_{\mathcal{B}(0,1)}(G) = -G^\sharp$. Hence, (2) is equivalent to

$$X^{k+1} = X^k + t^k \operatorname{LMO}_{\mathcal{B}(0,1)}(G^k) = X^k - \frac{t^k}{\|G^k\|_\star}(G^k)^\sharp, \tag{4}$$

i.e., a normalized steepest descent step with stepsize $\gamma^k := t^k/\|G^k\|_\star$. We alternate between the sharp operator and LMO formulations, depending on the assumptions at play. Theorems 3 and 5 use the former; Theorems 4 and 6, the latter. We explore this and other reformulations in Section C.

**The algorithm.** Working with compression in non-Euclidean geometry presents several challenges. In addition to the lack of a standard descent lemma, further complications arise from interactions between gradient stochasticity and compression and unknown variance behavior under biased compression. Yet, we develop the *first communication-efficient variant of* Gluon (and by extension, its special cases Muon and Scion), called EF21-Muon, that combines biased compression, gradient stochasticity, and momentum, all while enjoying *strong theoretical guarantees*. A simplified version of the algorithm, applied globally to $X$, is shown in Algorithm 1. A more general, deep learning-oriented layer-wise variant operating in the product space $\mathcal{S} := \bigotimes_{i=1}^p \mathbb{R}^{m_i \times n_i}$ is given in Algorithm 3. For clarity, we focus on the simplified variant throughout the main text; all theoretical results presented here are special cases of the general layer-wise guarantees provided in Section E.

While the pseudocode is largely self-explanatory (for a more detailed description, see Section B.2), we highlight the most important components:

⋄ **Role of Compression.** Compression is key for reducing communication overhead in distributed training. Algorithm 1 adheres to this principle by transmitting the compressed messages $S^k$ and $R_j^k$ only, never the full dense updates. When compression is disabled (i.e., $\mathcal{C}_j^k$, $\mathcal{C}^k$ are identity mappings) and in the single-node setting ($n = 1$), EF21-Muon reduces exactly to Gluon, which in turn recovers Muon and Scion (all of which were originally designed for non-distributed settings).

⋄ **Role of Error Feedback.** Even in the Euclidean setup, biased compression can break distributed GD unless some form of error feedback is used (see Section 2). To remedy this, we adopt a modern strategy inspired by EF21 (Richtárik et al., 2021) for the w2s direction. Its role is to stabilize training and prevent divergence. To reduce s2w communication overhead, we further incorporate the primal compression mechanism of EF21-P (Gruntkowska et al., 2023).

⋄ **Role of Gradient Stochasticity.** In large-scale ML, computing full gradients $\nabla f_j(x)$ is typically computationally infeasible. In practice, they are replaced with stochastic estimates, which drastically reduces per-step computational cost and makes the method scalable to practical workloads.

⋄ **Role of Momentum.** Stochastic gradients inevitably introduce noise into the optimization process. Without further stabilization, this leads to convergence to a neighborhood of the solution only. Momentum mitigates this issue, reducing the variance in the updates and accelerating convergence.

## 4 CONVERGENCE RESULTS

To support our convergence analysis, we adopt standard lower-boundedness assumptions on the global objective $f$, and in certain cases, also on the local functions $f_j$.

**Assumption 1.** *There exist $f^\star \in \mathbb{R}$ such that $f(X) \geq f^\star$ for all $X \in \mathcal{S}$.*

**Assumption 2.** *For all $j \in [n]$, there exist $f_j^\star \in \mathbb{R}$ such that $f_j(X) \geq f_j^\star$ for all $X \in \mathcal{S}$.*

We study two smoothness regimes. The first, standard $L$–smoothness generalized to arbitrary norms (used in Theorems 3 and 5), is the default in virtually all convergence results for Muon and Scion (Kovalev, 2025; Pethick et al., 2025b; Li & Hong, 2025).

**Assumption 3.** *The function $f$ is $L$–smooth, i.e., $\|\nabla f(X) - \nabla f(Y)\|_\star \leq L \|X - Y\|$ for all $X, Y \in \mathcal{S}$. Moreover, the functions $f_j$ are $L_j$–smooth for all $j \in [n]$. We define[2] $\tilde{L}^2 := \frac{1}{n} \sum_{j=1}^n L_j^2$.*

To our knowledge, the only exception departing from this standard setting is the recent work on Gluon (Riabinin et al., 2025b). The authors argue that layer-wise optimizers are designed specifically for deep learning, where the classical smoothness assumption is known to fail (Zhang et al., 2020). Instead, they build upon the $(L^0, L^1)$–smoothness model introduced by Zhang et al. (2020)[3] (Assumption 4), a strictly weaker alternative motivated by empirical observations from NLP training dynamics. Riabinin et al. (2025b) introduce a *layer-wise* variant (Assumption 8), arguing that heterogeneity across network layers requires smoothness constants to vary accordingly. Consistent with this line of work, we provide convergence guarantees under the *layer-wise $(L^0, L^1)$–smoothness* assumption (Theorems 17 and 24). For clarity, the main text treats the case of a generic vector space $\mathcal{S}$, without delving into the product space formulation (see Section B), in which case the assumption reduces to a non-Euclidean variant of asymmetric $(L^0, L^1)$–smoothness from Chen et al. (2023).

**Assumption 4.** *The function $f : \mathcal{S} \mapsto \mathbb{R}$ is $(L^0, L^1)$–smooth, i.e., there exist $L^0, L^1 > 0$ such that*

$$\|\nabla f(X) - \nabla f(Y)\|_\star \leq \left(L^0 + L^1 \|\nabla f(X)\|_\star\right) \|X - Y\| \quad \forall X, Y \in \mathcal{S}.$$

*Moreover, the functions $f_j$, $j \in [n]$, are $(L_j^0, L_j^1)$–smooth. We define $L_{\max}^1 := \max_{j \in [n]} L_j^1$ and $\bar{L}^0 := \frac{1}{n} \sum_{j=1}^n L_j^0$.*

Assumption 4 is strictly more general than Assumption 3, as it allows the smoothness constant to grow with the norm of the gradient, a key property observed in deep learning (Zhang et al., 2020).

---

[2]In theoretical results, $\tilde{L}$ could potentially be improved to the arithmetic mean–see Richtárik et al. (2024).

[3]The original $(L^0, L^1)$–smoothness assumption of Zhang et al. (2020) was defined for twice-differentiable functions via Hessian norms. This notion and our Assumption 4 are closely related–see Chen et al. (2023).

**Deterministic setting.** As a warm-up, we first present the convergence guarantees of Algorithm 2–a deterministic counterpart of Algorithm 1 using deterministic gradients without momentum (though stochasticity may still arise from compression). The first theorem addresses the smooth setting.

**Theorem 3.** *Let Assumptions 1 and 3 hold. Let $\{X^k\}_{k=0}^{K-1}$, $K \geq 1$, be the iterates of Algorithm 2 (with $p = 1$) initialized with $X^0 = W^0$, $G_j^0 = \nabla f_j(X^0)$, $j \in [n]$, and run with $\mathcal{C}^k \in \mathbb{B}(\alpha_P)$, $\mathcal{C}_j^k \in \mathbb{B}_\star(\alpha_D)$ and $0 < \gamma^k \equiv \gamma \leq \left(2L + {}^4\!/\!_{\alpha_D}\sqrt{12 + {}^{66}\!/\!_{\alpha_P^2}}\tilde{L}\right)^{-1}$. Then*

$$\frac{1}{K}\sum_{k=0}^{K-1}\mathbb{E}\left[\left\|\nabla f(X^k)\right\|_\star^2\right] \leq \frac{4\left(f(X^0)-f^\star\right)}{K\gamma}.$$

Theorem 3 is a special case of the general layer-wise result in Theorem 14. To our knowledge, no prior work analyzes comparable compressed methods under general non-Euclidean geometry. In the Euclidean case, our guarantees recover known results (up to constants): without primal compression ($\alpha_P = 1$), they match the rate of Richtárik et al. (2021, Theorem 1); with primal compression, they align with the rate of EF21-BC from Fatkhullin et al. (2021, Theorem 21) (though EF21-Muon and EF21-BC differ algorithmically, and the former does not reduce to the latter in the Euclidean case).

In the generalized smooth setup, we establish convergence without primal compression. However, as we argue in Section D.1, the s2w communication *can* still be made efficient through appropriate norm selection. Indeed, we find that LMOs under certain norms naturally induce *compression-like behavior*.

**Theorem 4.** *Let Assumptions 1, 2 and 4 hold and let $\{X^k\}_{k=0}^{K-1}$, $K \geq 1$, be the iterates of Algorithm 2 (with $p = 1$) initialized with $G_j^0 = \nabla f_j(X^0)$, $j \in [n]$, and run with $\mathcal{C}^k \equiv \mathcal{I}$ (the identity compressor), $\mathcal{C}_j^k \in \mathbb{B}_\star(\alpha_D)$, and $t^k \equiv \frac{\eta}{\sqrt{K+1}}$ for some $\eta > 0$. Then,*

$$\min_{k=0,\ldots,K}\mathbb{E}\left[\left\|\nabla f(X^k)\right\|_\star\right] \leq \frac{\exp\left(4\eta^2 C L_{\max}^1\right)}{\eta\sqrt{K+1}}\delta^0 + \frac{\eta\left(4C\frac{1}{n}\sum_{j=1}^{n}L_j^1\left(f^\star-f_j^\star\right)+C\frac{1}{n}\sum_{j=1}^{n}\frac{L_j^0}{L_j^1}+D\right)}{\sqrt{K+1}},$$

*where $\delta^0 := f(X^0) - f^\star$, $C := \frac{L^1}{2} + \frac{2\sqrt{1-\alpha_D}L_{\max}^1}{1-\sqrt{1-\alpha_D}}$ and $D := \frac{L^0}{2} + \frac{2\sqrt{1-\alpha_D}\tilde{L}^0}{1-\sqrt{1-\alpha_D}}$.*

Theorem 4, a corollary of the layer-wise result in Theorem 17, achieves the same desirable $\mathcal{O}({}^1\!/\!\sqrt{K})$ rate for expected gradient norms as Theorem 3, but with radii $t^k$ that are *independent of problem-specific constants*. If smoothness constants are known in advance, they can be incorporated into the choice of $\eta$ to improve the dependence on these constants in the final rate. In the Euclidean case, our guarantee matches that of $\|\text{EF21}\|$ under $(L^0, L^1)$–smoothness established by Khirirat et al. (2024).

**Stochastic setting.** We now turn to the convergence guarantees of our practical variant of EF21-Muon (Algorithms 1 and 3), which incorporates noisy gradients and momentum. We assume access to a standard stochastic gradient oracles $\nabla f_j(\cdot;\xi_j)$, $\xi_j \sim \mathcal{D}_j$ with bounded variance.

**Assumption 5.** *The stochastic gradient estimators $\nabla f_j(\cdot;\xi_j) : \mathcal{S} \mapsto \mathcal{S}$ are unbiased and have bounded variance. That is, $\mathbb{E}_{\xi_j \sim \mathcal{D}_j}\left[\nabla f_j(X;\xi_j)\right] = \nabla f_j(X)$ for all $X \in \mathcal{S}$ and there exists $\sigma \geq 0$ such that $\mathbb{E}_{\xi_j \sim \mathcal{D}_j}\left[\left\|\nabla f_j(X;\xi_j) - \nabla f_j(X)\right\|_2^2\right] \leq \sigma^2$ for all $X \in \mathcal{S}$.*

Note that the variance bound in Assumption 5 is expressed in terms of the Euclidean norm rather than $\|\cdot\|$ to facilitate the bias-variance decomposition. Nevertheless, since $\mathcal{S}$ is finite-dimensional, the magnitudes measured in $\|\cdot\|_2$ can be related to quantities measured in $\|\cdot\|$ via norm equivalence. That is, there exist $\underline{\rho}, \bar{\rho} > 0$ such that $\underline{\rho}\|X\| \leq \|X\|_2 \leq \bar{\rho}\|X\|$ for all $X \in \mathcal{S}$.

As in the deterministic setting, we begin by analyzing the smooth case.

**Theorem 5.** *Let Assumptions 1, 3 and 5 hold. Let $\{X^k\}_{k=0}^{K-1}$, $K \geq 1$, be the iterates of Algorithm 1 initialized with $X^0 = W^0$, $G_j^0 = M_j^0 = \nabla f_j(X^0;\xi_j^0)$, $j \in [n]$, and run with $\mathcal{C}^k \in \mathbb{B}(\alpha_P)$, $\mathcal{C}_j^k \in \mathbb{B}_2(\alpha_D)$, any $\beta \in (0,1]$, and $0 \leq \gamma^k \equiv \gamma \leq \left(2\sqrt{\zeta} + 2L\right)^{-1}$, where $\zeta := \frac{\bar{\rho}^2}{\underline{\rho}^2}\left(\frac{12}{\beta^2}L^2 + \frac{24(\beta+2)}{\alpha_P^2}L^2 + \frac{36\left(\beta^2+4\right)}{\alpha_D^2}\tilde{L}^2 + \frac{144\beta^2(2\beta+5)}{\alpha_P^2\alpha_D^2}\tilde{L}^2\right)$. Then*

$$\frac{1}{K}\sum_{k=0}^{K-1}\mathbb{E}\left[\left\|\nabla f(X^k)\right\|_\star^2\right] \leq \frac{4\delta^0}{K\gamma} + \frac{24}{K}\left(\frac{1}{\sqrt{n}\beta} + \frac{12\beta}{\alpha_D^2}\right)\sigma\bar{\rho}^2 + 24\left(\frac{1}{n} + \frac{(1-\alpha_D)\beta}{\alpha_D} + \frac{12\beta^2}{\alpha_D^2}\right)\sigma^2\bar{\rho}^2\beta,$$

*where $\delta^0 := f(X^0) - f^\star$.*

Theorem 5 is a special case of Theorem 19. Choosing $\gamma = \left(2\sqrt{\zeta} + 2L\right)^{-1}$ and $\beta = \min\left\{1, \left(\frac{\delta^0 Ln}{\underline{\rho}^2 \sigma^2 K}\right)^{1/2}, \left(\frac{\delta^0 L\alpha_D}{\underline{\rho}^2 \sigma^2 K}\right)^{1/3}, \left(\frac{\delta^0 L\alpha_D^2}{\underline{\rho}^2 \sigma^2 K}\right)^{1/4}\right\}$, it guarantees that

$$\frac{1}{K}\sum_{k=0}^{K-1}\mathbb{E}\left[\left\|\nabla f(X^k)\right\|_\star^2\right] = \mathcal{O}\left(\frac{\delta^0 \bar{\rho}^2 \tilde{L}^0}{\underline{\rho}^2 \alpha_P \alpha_D K} + \left(\frac{\delta^0 \bar{\rho}^4 \sigma^2 L}{\underline{\rho}^2 nK}\right)^{1/2} + \left(\frac{\delta^0 \bar{\rho}^3 \sigma L}{\underline{\rho}^2 \sqrt{\alpha_D} K}\right)^{2/3} + \left(\frac{\delta^0 \bar{\rho}^{8/3} \sigma^{2/3} L}{\bar{\rho}^2 \alpha_D^{2/3} K}\right)^{3/4}\right)$$

(see Corollary 2). In the absence of stochasticity and momentum ($\sigma^2 = 0$, $\beta = 1$), Algorithm 1 reduces to Algorithm 2 (with $p = 1$), and the guarantee in Theorem 5 recovers that of Theorem 3, up to constants (Remark 22). In the Euclidean case without primal compression ($\bar{\rho}^2 = \rho^2 = \alpha_P = 1$), Theorem 5 matches the rate of EF21-SDGM established by Fatkhullin et al. (2023, Theorem 3), again up to constants (Remark 21). Finally, one may employ compressors $\mathcal{C}^k \in \mathbb{B}_2(\alpha_P)$ instead of $\mathcal{C}^k \in \mathbb{B}(\alpha_P)$, though this introduces an additional dependence on $\bar{\rho}^2$ in the constant $\zeta$ (Remark 23).

As in Theorem 4, in the $(L^0, L^1)$–smooth setup, we set $\mathcal{C}^k \equiv \mathcal{I}$.

**Theorem 6.** *Let Assumptions 1, 2, 4 and 5 hold. Let $\{X^k\}_{k=0}^{K-1}$, $K \geq 1$, be the iterates of Algorithm 1 initialized with $M_j^0 = \nabla f_j(X^0; \xi_j^0)$, $G_j^0 = \mathcal{C}_j^0(\nabla f_j(X^0; \xi_j^0))$, $j \in [n]$, and run with $\mathcal{C}^k \equiv \mathcal{I}$ (the identity compressor), $\mathcal{C}_j^k \in \mathbb{B}_2(\alpha_D)$, $\beta = 1/(K+1)^{1/2}$ and $0 \leq t^k \equiv t = \eta/(K+1)^{3/4}$, where*

$$\eta^2 \leq \min\left\{\frac{(K+1)^{1/2}}{6(L^1)^2}, \frac{(K+1)^{1/2}(1-\sqrt{1-\alpha_D})\rho}{24\sqrt{1-\alpha_D}\bar{\rho}(L_{\max}^1)^2}, \frac{\rho}{24\bar{\rho}(L_{\max}^1)^2}, 1\right\}. \textit{ Then}$$

$$\min_{k=0,\dots,K}\mathbb{E}\left[\left\|\nabla f(X^k)\right\|_\star\right] \leq \frac{3\left(f(X^0) - f^\star\right)}{\eta(K+1)^{1/4}} + \frac{\eta L^0}{(K+1)^{3/4}} + \frac{16\sqrt{1-\alpha_D}\bar{\rho}\sigma}{(1-\sqrt{1-\alpha_D})(K+1)^{1/2}} + \frac{8\bar{\rho}\sigma}{\sqrt{n}(K+1)^{1/4}}$$

$$+ \frac{\eta\bar{\rho}}{\rho}\left(\frac{8}{(K+1)^{1/4}} + \frac{8\sqrt{1-\alpha_D}}{(1-\sqrt{1-\alpha_D})(K+1)^{3/4}}\right)\left(\frac{1}{n}\sum_{j=1}^n (L_j^1)^2\left(f^\star - f_j^\star\right) + \bar{L}^0\right).$$

Analogously to Theorem 5, Theorem 6 (a corollary of Theorem 24) establishes an $\mathcal{O}(1/K^{1/4})$ convergence rate, matching state-of-the-art guarantees for SGD-type methods in the non-convex setting (Cutkosky & Mehta, 2020; Sun et al., 2023). Among the terms with the worst scaling in $K$, $3\left(f(X^0) - f^\star\right)/\eta(K+1)^{1/4}$ is standard and reflects the impact of the initial suboptimality. $8\bar{\rho}\sigma/\sqrt{n}(K+1)^{1/4}$ captures gradient stochasticity, scaling linearly with the standard deviation $\sigma$, but decaying with the square root of the number of clients $n$. The term $\frac{1}{n}\sum_{j=1}^n (L_j^1)^2\left(f^\star - f_j^\star\right)$ quantifies client heterogeneity and vanishes when local optima $f_j^\star$ coincide with the global minimum $f^\star$, and otherwise scales with the local smoothness constants $L_j^1$. All compression-driven error terms vanish when compression is disabled ($\alpha_D = 1$). Finally, in the Euclidean case ($\bar{\rho}^2 = \rho^2 = 1$), the rate recovers that of $\|$EF21-SDGM$\|$ from Khirirat et al. (2024, Theorem 2), up to constants.

## 5 EXPERIMENTS

We present key experimental results below, with additional details and extended experiments available in Section G.[4]

**Experimental setup.** All experiments are conducted on 4 NVIDIA Tesla V100-SXM2-32GB GPUs or 4 NVIDIA A100-SXM4-80GB in a Distributed Data Parallel (DDP) setup. The dataset is evenly partitioned across workers, with one worker node acting as the master, aggregating compressed updates from the others. Training and evaluation are implemented in PyTorch,[5] extending open-source codebases (Pethick et al., 2025a; Riabinin et al., 2025a; Karpathy, 2023).

We train a `NanoGPT` model (Karpathy, 2023) with 124M parameters on the `FineWeb10B` dataset (Penedo et al., 2024), using input sequences of length 1024 and a batch size of 256. Optimization is performed with EF21-Muon, using spectral norm LMOs for hidden layers and $\ell_\infty$ norm LMOs for embedding and output layers (which coincide due to weight sharing), following the approach of Pethick et al. (2025b). For spectral norm LMOs, inexact updates are computed with 5 Newton–Schulz iterations (Kovarik, 1970; Björck & Bowie, 1971), as in Jordan et al. (2024).

---

[4]Code for experiments is available here.
[5]PyTorch Documentation: https://pytorch.org/docs/stable/index.html

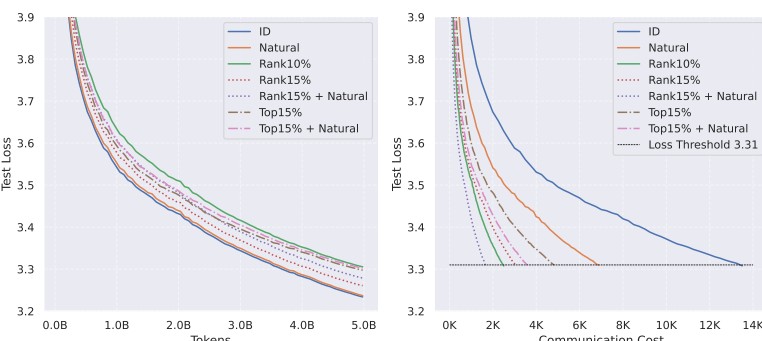

Figure 1: Left: Test loss vs. # of tokens processed. Right: Test loss vs. # of bytes sent from each worker to the server normalized by model size to reach test loss 3.31. Rank/Top$X\%$ = Rank/Top$K$ compressor with sparsification level $X\%$; ID = no compression.

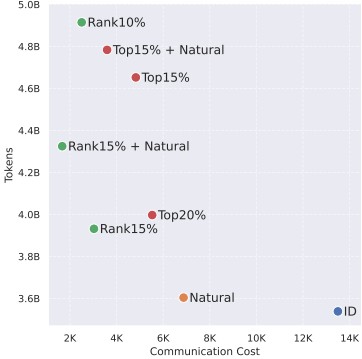

Figure 2: Trade-off between token efficiency and communication cost for different compression setups at a target test loss of 3.31.

Following common practice in communication compression literature, we assume that broadcasting is free and focus on w2s communication. Thus, the server-side compressor is fixed to $\mathcal{I}$, while worker compressors vary among Top$K$, Rank$K$ (Safaryan et al., 2021), Natural compressor (Horváth et al., 2022) and combinations thereof: Top$K$ + Natural compressor of selected elements, and Rank$K$ + Natural compressor applied to all components of the low rank decomposition. These are tested under multiple compression levels and compared against an uncompressed baseline (i.e., standard Scion/Gluon; see Section 3). Learning rates are tuned per optimizer and experimental setting, initialized from the values in the Gluon repository (Riabinin et al., 2025a) (see Section G.3). We adopt the same learning rate scheduler as Karpathy (2023) and fix the momentum parameter to 0.9. Model and optimizer hyperparameters are summarized in Tables 3 and 5, respectively.

**Results.** For Rank$K$ and Top$K$ compressors, we evaluate multiple compression levels (in plots, Rank$X\%$/Top$X\%$ denotes a Rank$K$/Top$K$ compressor with compression level $X\%$). We report experimental results for a 5B-token training budget ($> 40\times$ model size) in Figure 1 (left), and to reach a strong loss threshold of 3.31 in Figures 1 (right) and 2.

Table 2: Communication cost per round (in bytes), normalized relative to the identity compressor.

| Compressor | Relative Cost |
|---|---|
| ID | 1.0000 |
| Natural | 0.5000 |
| Rank20% | 0.2687 |
| Rank15% | 0.2019 |
| Rank15% + Natural | 0.1010 |
| Rank10% | 0.1335 |
| Rank10% + Natural | 0.0667 |
| Rank5% | 0.0667 |
| Top20% | 0.3625 |
| Top15% | 0.2718 |
| Top15% + Natural | 0.1969 |
| Top10% | 0.1812 |
| Top10% + Natural | 0.1312 |
| Top5% | 0.0906 |

The number of tokens required to reach a target loss depends on the compressor. Figure 2 provides a comparison of the numbers of tokens used in the training run to reach a strong test loss threshold of 3.31 plotted against the communication cost (reported as the number of bits transmitted from each worker to the server normalized by the model size), plotted against the w2s communication cost. Shorter 2.5B-token runs are reported in Section G.5 to assess performance under limited training budgets.

In Figure 1, we plot test loss vs. tokens processed, as well as the w2s communication cost required to reach the 3.31 loss threshold. For each compressor, we report its most competitive configuration (see Section G.4 for a detailed ablation). As expected, compression slows convergence in terms of number of training steps, but substantially reduces per-step communication cost (Table 2). Overall, this yields significant **communication savings—up to** $7\times$ for Rank15% + Natural compressor, and roughly $4\times$ for Top15% + Natural compressor—relative to the uncompressed baseline.

ACKNOWLEDGMENTS

The research reported in this publication was supported by funding from King Abdullah University of Science and Technology (KAUST): i) KAUST Baseline Research Scheme, ii) CRG Grant ORFS-CRG12-2024-6460, and iii) Center of Excellence for Generative AI, under award number 5940.

For computer time, this research used the resources of the Supercomputing Laboratory at King Abdullah University of Science & Technology (KAUST) in Thuwal, Saudi Arabia.

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

APPENDIX

CONTENTS

# A  RELATED WORK

## A.1  COMPRESSION

The ML community has developed two dominant strategies to address the communication bottleneck. The first is *compression*, implemented through techniques such as sparsification or quantization (Seide et al., 2014; Alistarh et al., 2017; Beznosikov et al., 2020; Szlendak et al., 2021; Horváth et al., 2022), which reduce communication costs by transmitting lossy representations of dense updates. Compression techniques have been extensively studied in the Euclidean regime. The other approach is *local training* (Mangasarian, 1995; Povey et al., 2014; McMahan et al., 2017), which lowers communication frequency by synchronizing with the server only periodically, after several local updates on the clients. These two approaches can be combined, yielding additional provable benefits by leveraging both mechanisms (Condat et al., 2024). In this work, we focus on compression. Local training introduces a distinct set of challenges and trade-offs, and is orthogonal to our approach.

There are two primary compression objectives in distributed optimization: workers-to-server (w2s) (= uplink) and server-to-workers (s2w) (= downlink) communication. A large body of prior work focuses exclusively on w2s compression, assuming that broadcasting from the server to the workers is either free or negligible (Gorbunov et al., 2021; Szlendak et al., 2021; Tyurin & Richtárik, 2023a; Pirau et al., 2024). This assumption is partly due to analytical convenience, but can also be justified in settings where the server has significantly higher bandwidth, greater computational resources, or when the network topology favors fast downlink speeds (Kairouz et al., 2021). However, in many communication environments, this asymmetry does not hold. For instance, in 4G LTE and 5G networks, the upload and download speeds can be comparable, with the ratio between w2s and s2w bandwidths bounded within an order of magnitude (Huang et al., 2012; Narayanan et al., 2021). In such cases, s2w communication costs become non-negligible, and optimizing for both directions is essential for practical efficiency (Zheng et al., 2019; Liu et al., 2020; Philippenko & Dieuleveut, 2021; Fatkhullin et al., 2021; Gruntkowska et al., 2023; Tyurin & Richtárik, 2023b; Gruntkowska et al., 2024).

## A.2  ERROR FEEDBACK

To address the communication bottleneck, a natural approach is to apply biased compressors to the transmitted gradients. For the standard (Euclidean) GD, which iterates

$$X^{k+1} = X^k - \gamma^k \nabla f(X^k) = X^k - \gamma^k \left( \frac{1}{n} \sum_{j=1}^{n} \nabla f_j(X^k) \right),$$

where $\gamma^k > 0$ is the stepsize, this would yield the update rule

$$X^{k+1} = X^k - \gamma^k \left( \frac{1}{n} \sum_{j=1}^{n} \mathcal{C}_j^k (\nabla f_j(X^k)) \right).$$

Sadly, this "enhancement" can result in exponential divergence, even in simplest setting of minimizing the average of three strongly convex quadratic functions (Beznosikov et al., 2020, Example 1). Empirical evidence of such instability appeared much earlier, prompting Seide et al. (2014) to propose a remedy in the form of an *error feedback* (EF) mechanism, which we refer to as EF14.

Initial theoretical insights into EF14 were established in the simpler single-node setting (Stich et al., 2018; Alistarh et al., 2018). The method was subsequently analyzed in the convex case by Karimireddy et al. (2019); Beznosikov et al. (2020); Gorbunov et al. (2020). Next, Qian et al. (2021) showed that error feedback methods can be combined with Nesterov-style acceleration (Nesterov, 2003), though at the cost of incorporating additional unbiased compression, leading to increased communication overhead per iteration. These analyses were later extended to the nonconvex regime by Stich & Karimireddy (2019). This motivated a series of extensions combining error feedback with additional algorithmic components, such as bidirectional compression (Tang et al., 2019), decentralized training protocols (Koloskova et al., 2019), and the incorporation of momentum either on the client (Zheng et al., 2019) or server side (Xie et al., 2020). While these works advanced the state

of the art, their guarantees relied on strong regularity assumptions, such as bounded gradients (BG) or bounded gradient similarity (BGS), which may be difficult to justify in practical deep learning scenarios.

The limitations of EF14 and its successors were partially overcome by Richtárik et al. (2021), who proposed a refined variant termed EF21. EF21 eliminates the need for strong assumptions such as BG and BGS, relying only on standard assumptions (smoothness of the local functions $f_j$ and the existence of a global lower bound on $f$), while improving the iteration complexity to the desirable $\mathcal{O}(1/\sqrt{K})$ in the deterministic setting. Building on this foundation, a series of extensions and generalizations followed. These include adaptations to partial participation, variance-reduction, proximal setting, and bidirectional compression (Fatkhullin et al., 2021), a generalization from contractive to three-point compressors (Richtárik et al., 2022), support for adaptive compression schemes (Makarenko et al., 2022), and EF21-P–a modification of EF21 from gradient compression to model compression (Gruntkowska et al., 2023). Further developments used EF21 in the design of Byzantine robust methods (Rammal et al., 2024), applied it to Hessian communication (Islamov et al., 2023), and extended the theoretical analysis to the $(L^0, L^1)$–smooth regime (Khirirat et al., 2024).

With this historical overview in place, we now narrow our focus to two developments in the error feedback literature that are particularly relevant to this work: EF21 (Richtárik et al., 2021) and EF21-P (Gruntkowska et al., 2023).

EF21 is a method for w2s communication compression. It aims to solve problem (1) via the iterative process

$$
\begin{aligned}
X^{k+1} &= X^k - \gamma G^k, \\
G_j^{k+1} &= G_j^k + \mathcal{C}_j^k(\nabla f_j(X^{k+1}) - G_j^k), \\
G^{k+1} &= \frac{1}{n}\sum_{j=1}^{n} G_j^{k+1},
\end{aligned}
$$

where $\gamma > 0$ is the stepsize and $\mathcal{C}_j^k \in \mathbb{B}_2(\alpha_D)$ are independent contractive compressors. In the EF21 algorithm, each client $j$ keeps track of a gradient estimator $G_j^k$. At each iteration, the clients compute their local gradient $\nabla f_j(X^{k+1})$, subtract the stored estimator $G_j^k$, and then compresses this difference using a biased compression operator. The compressed update is sent to the server, which aggregates updates from all clients and uses them to update the global model. Concurrently, each client updates its error feedback vector by using the same compressed residual. Importantly, EF21 compresses only the uplink communication (i.e., vectors sent from clients to the server), while downlink communication remains uncompressed. That is, the global model $X^{k+1}$ is transmitted in full precision from the server to all clients, under the assumption that downlink communication is not a bottleneck.

A complementary approach is proposed in the follow-up work of Gruntkowska et al. (2023), which introduces a primal variant of EF21, referred to as EF21-P. Unlike EF21, which targets uplink compression (from workers to server), EF21-P is explicitly designed for s2w compression. The method proceeds via the iterative scheme

$$
X^{k+1} = X^k - \gamma\nabla f(W^k) = X^k - \gamma\frac{1}{n}\sum_{j=1}^{n}\nabla f_j(W^k),
$$

$$
W^{k+1} = W^k + \mathcal{C}^k(X^{k+1} - W^k),
$$

where $\gamma > 0$ is the stepsize and $\mathcal{C}^k \in \mathbb{B}_2(\alpha_P)$ are independent contractive compressors. Analogous to EF21, the EF21-P method employs error feedback to compensate for the distortion introduced by compression. However, rather than correcting gradient estimates, EF21-P maintains and updates an estimate of the model parameters, $W^k$. The server computes the update $X^{k+1}$, but broadcasts only a compressed difference $\mathcal{C}^k(X^{k+1} - W^k)$ to the clients.

In its basic form, EF21-P assumes dense uplink communication–i.e., the clients transmit full gradients $\nabla f_j(W^k)$ to the server. Nonetheless, EF21-P can be naturally extended to bidirectional compression by integrating it with an uplink compression mechanism, enabling full communication efficiency (Gruntkowska et al., 2023).

### A.3 GENERALIZED SMOOTHNESS

A standard assumption in the convergence analysis of gradient-based methods is Lipschitz smoothness of the gradient (Assumption 3). However, many modern learning problems–especially in deep learning–violate this assumption. Empirical evidence has shown non-smoothness in a variety of architectures and tasks, including `LSTM` language modeling, image classification with `ResNet20` (Zhang et al., 2020), and transformer models (Crawshaw et al., 2022). These observations motivated the search for alternative smoothness models that better reflect the behavior of practical objectives.

One such model is $(L^0, L^1)$–smoothness, introduced by Zhang et al. (2020) for twice continuously differentiable functions in the Euclidean setting. The authors define a function $f : \mathbb{R}^d \to \mathbb{R}$ to be $(L^0, L^1)$–smooth if

$$\left\| \nabla^2 f(X) \right\|_2 \leq L^0 + L^1 \left\| \nabla f(X) \right\|_2 \qquad \forall X \in \mathbb{R}^d.$$

This condition generalizes standard Lipschitz smoothness and has been shown empirically to capture deep learning loss landscapes more faithfully than the classical model (Zhang et al., 2020; Crawshaw et al., 2022). Subsequent works extended the above condition beyond the twice differentiable case (Li et al., 2023; Chen et al., 2023). In particular, Chen et al. (2023) introduced asymmetric and symmetric variants of $(L^0, L^1)$–smoothness, where the asymmetric form (a special case of Assumption 4 restricted to Euclidean norms) is given by

$$\left\| \nabla f(X) - \nabla f(Y) \right\|_2 \leq \left( L^0 + L^1 \left\| \nabla f(X) \right\|_2 \right) \left\| X - Y \right\|_2 \quad \forall X, Y \in \mathbb{R}^d.$$

This framework has since been used in the non-Euclidean setting (Pethick et al., 2025c) and adapted to the layer-wise structure of deep networks by Riabinin et al. (2025b), who introduced non-Euclidean *layer-wise* $(L^0, L^1)$–smoothness assumption (Assumption 8). This layer-aware view aligns naturally with LMO-based optimizers that operate on individual parameter groups.

The idea of accounting for the heterogeneous structure of parameters is not unique to the work of Riabinin et al. (2025b). Anisotropic smoothness conditions, where smoothness constants can vary across coordinates or parameter blocks, have been studied extensively, for example in the context of coordinate descent methods (Nesterov, 2012; Richtárik & Takáč, 2014; Nutini et al., 2017). Variants of coordinate-wise or block-wise (generalized) smoothness assumptions have also been used to analyze algorithms such as `signSGD` (Bernstein et al., 2018; Crawshaw et al., 2022), `AdaGrad` (Jiang et al., 2024; Liu et al., 2024), and `Adam` (Xie et al., 2024). These works collectively reinforce the need for smoothness models that reflect the anisotropic geometry of modern neural networks.

# B LAYER-WISE SETUP

So far, we have been operating in an abstract vector space $\mathcal{S}$, without assuming any particular structure. This is the standard approach in the vast majority of the theoretical optimization literature in machine learning, where model parameters are typically flattened into vectors in $\mathbb{R}^d$. However, modern deep networks are inherently structured objects, with a clear *layer-wise* organization. While treating parameters as flat vectors can still yield meaningful convergence guarantees, explicitly modeling this layer-wise structure allows us to formulate assumptions that more accurately reflect the underlying geometry of the model Nesterov (2012); Richtárik & Takáč (2014); Crawshaw et al. (2022); Jiang et al. (2024). This, in turn, can lead to improved theoretical results (Liu et al., 2024; Riabinin et al., 2025b).

A further motivation for adopting the layer-wise perspective is that the algorithms that inspired this work–Muon, Scion, and Gluon–are themselves *layer-wise by design*. Rather than operating on the entire parameter vector, they apply separate LMO updates to each layer or building block independently. This modular treatment is one of the main reasons for their strong empirical performance.

With this motivation in mind, we now turn to solving the optimization problem (1) in a setting where the parameter vector $X \in \mathcal{S}$ represents a collection of matrices $X_i \in \mathcal{S}_i := \mathbb{R}^{m_i \times n_i}$ corresponding to the trainable parameters of each layer $i \in \{1, \ldots, p\}$ in a neural network. For notational convenience, we write $X = [X_1, \ldots, X_p]$ and $\nabla f(X) = [\nabla_1 f(X), \ldots, \nabla_p f(X)]$, where $\nabla_i f(X)$ is the gradient component corresponding to the $i$th layer. Accordingly, $\mathcal{S}$ is the $d$-dimensional product space

$$\mathcal{S} := \bigotimes_{i=1}^p \mathcal{S}_i \equiv \mathcal{S}_1 \otimes \cdots \otimes \mathcal{S}_p,$$

where $d := \sum_{i=1}^p m_i n_i$. Each component space $\mathcal{S}_i$ is equipped with the trace inner product, defied as $\langle X_i, Y_i \rangle_{(i)} := \mathrm{tr}(X_i^\top Y_i)$ for $X_i, Y_i \in \mathcal{S}_i$, and an arbitrary norm $\|\cdot\|_{(i)}$, not necessarily induced by this inner product. We use $\|\cdot\|_{(i)\star}$ to denote the dual norm associated with $\|\cdot\|_{(i)}$ (i.e., $\|X_i\|_{(i)\star} := \sup_{\|Z_i\|_{(i)} \leq 1} \langle X_i, Z_i \rangle_{(i)}$ for any $X_i \in \mathcal{S}_i$). Furthermore, we use $\underline{\rho}_i, \bar{\rho}_i > 0$ to denote the norm equivalence constants such that

$$\underline{\rho}_i \|X_i\|_{(i)} \leq \|X_i\|_2 \leq \bar{\rho}_i \|X_i\|_{(i)} \qquad \forall X_i \in \mathcal{S}_i,$$

(or, equivalently, $\underline{\rho}_i \|X_i\|_2 \leq \|X_i\|_{(i)\star} \leq \bar{\rho}_i \|X_i\|_2$).

**Remark 7.** *In the case of* Muon*, the norms $\|\cdot\|_{(i)}$ are taken to be the spectral norms, i.e., $\|\cdot\|_{(i)} = \|\cdot\|_{2\to 2}$. Since for any matrix $X_i$ of rank at most $r$, we have*

$$\|X_i\|_{2\to 2} \leq \|X_i\|_F \leq \sqrt{r} \|X_i\|_{2\to 2},$$

*in this setting, $\underline{\rho}_i = 1$ and $\bar{\rho}_i = \sqrt{r}$.*

Given the block structure of $X$ across layers, the smoothness assumptions in Assumption 3 can be made more precise by assigning separate constants to each layer.

**Assumption 6** (Layer-wise smoothness). *The function $f : \mathcal{S} \mapsto \mathbb{R}$ is layer-wise $L^0$–smooth with constants $L^0 := (L_1^0, \ldots, L_p^0) \in \mathbb{R}_+^p$, i.e.,*

$$\|\nabla_i f(X) - \nabla_i f(Y)\|_{(i)\star} \leq L_i^0 \|X_i - Y_i\|_{(i)}$$

*for all $i = 1, \ldots, p$ and all $X = [X_1, \ldots, X_p] \in \mathcal{S}$, $Y = [Y_1, \ldots, Y_p] \in \mathcal{S}$.*

**Assumption 7** (Local layer-wise smoothness). *The functions $f_j : \mathcal{S} \mapsto \mathbb{R}$, $j \in [n]$, are layer-wise $L_j^0$–smooth with constants $L_j^0 := (L_{1,j}^0, \ldots, L_{p,j}^0) \in \mathbb{R}_+^p$, i.e.,*

$$\|\nabla_i f_j(X) - \nabla_i f_j(Y)\|_{(i)\star} \leq L_{i,j}^0 \|X_i - Y_i\|_{(i)}$$

*for all $i = 1, \ldots, p$ and all $X = [X_1, \ldots, X_p] \in \mathcal{S}$, $Y = [Y_1, \ldots, Y_p] \in \mathcal{S}$. We define $(\tilde{L}_i^0)^2 := \frac{1}{n} \sum_{j=1}^n (L_{i,j}^0)^2$.*

We invoke Assumptions 6 and 7 in Appendices E.3.1 and E.4.1 to extend Theorems 3 and 5 to the more general setting.

Smoothness is the standard assumption used in virtually all convergence results for Muon and Scion (Kovalev, 2025; Pethick et al., 2025b; Li & Hong, 2025) (except for the recent work on Gluon (Riabinin et al., 2025b)). However, as discussed in Section 4 and Section A.3, this assumption often fails to hold in modern deep learning settings. To address this, we adopt a more flexible and expressive condition: the *layer-wise* $(L^0, L^1)$–*smoothness* assumption (Riabinin et al., 2025b).

**Assumption 8** (Layer-wise $(L^0, L^1)$–smoothness). *The function* $f : \mathcal{S} \mapsto \mathbb{R}$ *is layer-wise* $(L^0, L^1)$– *smooth with constants* $L^0 := (L_1^0, \ldots, L_p^0) \in \mathbb{R}_+^p$ *and* $L^1 := (L_1^1, \ldots, L_p^1) \in \mathbb{R}_+^p$, *i.e.,*

$$\|\nabla_i f(X) - \nabla_i f(Y)\|_{(i)\star} \le \left( L_i^0 + L_i^1 \|\nabla_i f(X)\|_{(i)\star} \right) \|X_i - Y_i\|_{(i)}$$

*for all* $i = 1, \ldots, p$ *and all* $X = [X_1, \ldots, X_p] \in \mathcal{S}$, $Y = [Y_1, \ldots, Y_p] \in \mathcal{S}$.

Since, unlike Gluon, we operate in the distributed setting, we will also need an analogous assumption on the local functions $f_j$.

**Assumption 9** (Local layer-wise $(L^0, L^1)$–smoothness). *The functions* $f_j$, $j \in [n]$, *are layer-wise* $(L_j^0, L_j^1)$–*smooth with constants* $L_j^0 := (L_{1,j}^0, \ldots, L_{p,j}^0) \in \mathbb{R}_+^p$ *and* $L_j^1 := (L_{1,j}^1, \ldots, L_{p,j}^1) \in \mathbb{R}_+^p$, *i.e.,*

$$\|\nabla_i f_j(X) - \nabla_i f_j(Y)\|_{(i)\star} \le \left( L_{i,j}^0 + L_{i,j}^1 \|\nabla_i f_j(X)\|_{(i)\star} \right) \|X_i - Y_i\|_{(i)}$$

*for all* $i = 1, \ldots, p$ *and all* $X = [X_1, \ldots, X_p] \in \mathcal{S}$, $Y = [Y_1, \ldots, Y_p] \in \mathcal{S}$.

*For* $O \in \{0, 1\}$, *we define* $L_{\max,j}^O := \max_{i \in [p]} L_{i,j}^O$, $L_{i,\max}^O := \max_{j \in [n]} L_{i,j}^O$ *and* $\bar{L}_i^0 := \frac{1}{n} \sum_{j=1}^n L_{i,j}^0$.

Riabinin et al. (2025b) present empirical evidence showing that this more flexible, layer-wise approach is essential for accurately modeling the network's underlying structure. They demonstrate that the layer-wise $(L^0, L^1)$–smoothness condition approximately holds along the training trajectory of Gluon in experiments on the `NanoGPT` language modeling task. Motivated by these findings, in Appendices E.3.2 and E.4.2, we provide an analysis within this generalized framework, offering a *full generalization of* Gluon *to bidirectional compression*.

In the stochastic setting, we will also require a layer-wise analogue of Assumption 5.

**Assumption 10.** *The stochastic gradient estimators* $\nabla f_j(\cdot; \xi_j) : \mathcal{S} \mapsto \mathcal{S}$ *are unbiased and have bounded variance. That is,* $\mathbb{E}_{\xi_j \sim \mathcal{D}_j} [\nabla f_j(X; \xi_j)] = \nabla f_j(X)$ *for all* $X \in \mathcal{S}$ *and there exist* $\sigma_i \ge 0$ *such that*

$$\mathbb{E}_{\xi_j \sim \mathcal{D}_j} \left[ \|\nabla_i f_j(X; \xi_j) - \nabla_i f_j(X)\|_2^2 \right] \le \sigma_i^2, \quad \forall X \in \mathcal{S}, \, i = 1, \ldots, p.$$

We permit layer-dependent variance parameters $\sigma_i^2$, motivated by empirical evidence that variance is not uniform across layers. For example, Glentis et al. (2025) observe that, during training of LLaMA 130M with SGD and column-wise normalization (i.e., Gluon using the $\|\cdot\|_{1\to2}$ norm), the final and embedding layers display significantly higher variance.

## B.1 Muon, Scion AND Gluon

Muon, introduced by Jordan et al. (2024), is an optimizer for the hidden layers of neural networks (the first and last layers are trained with AdamW). Unlike traditional element-wise gradient methods, it updates each weight matrix as a whole. Given a layer $X_i$ and the corresponding (stochastic) gradient $G_i$, Muon selects an update that maximizes the alignment with the gradient to reduce loss, while constraining the update's size to avoid excessive model perturbation. This is formulated as a constrained optimization problem over the spectral norm ball:

$$\arg\min_{\Delta X_i} \langle G_i, \Delta X_i \rangle \quad \text{s.t.} \quad \|\Delta X_i\|_{2\to2} \le t_i, \tag{5}$$

where the radius $t_i > 0$ plays a role similar to a stepsize. The optimal update $\Delta X_i$ is obtained by orthogonalizing $G_i$ via its singular value decomposition $G_i = U_i \Sigma_i V_i^T$, leading to

$$\Delta X_i = -t_i U_i V_i^T.$$

This yields the basic update

$$X_i^{k+1} = X_i^k + \Delta X_i^k = X_i^k - t_i^k U_i^k (V_i^k)^T. \tag{6}$$

In practice, computing the SVD exactly at every step is expensive and not GPU-friendly. Muon instead uses Newton–Schulz iterations (Kovarik, 1970; Björck & Bowie, 1971) to approximate the orthogonalization. Combined with momentum, the practical update is

$$M_i^k = (1 - \beta_i) M_i^{k-1} + \beta_i G_i^k, \qquad X_i^{k+1} = X_i^k - t_i^k \text{NewtonSchulz}(M_i^k),$$

where $\beta_i \in (0, 1]$ is the momentum parameter and $M_i^k$ is the momentum-averaged gradient.

While Newton–Schulz iterations and momentum are crucial for practical efficiency, the essence of Muon lies in solving (5)–that is, computing the linear minimization oracle (LMO) over the spectral norm ball. Recall that $\text{LMO}_{\mathcal{B}(X,t)}(G) := \arg\min_{Z \in \mathcal{B}(X,t)} \langle G, Z \rangle$. Then

$$\Delta X_i = \arg\min_{Z_i \in \mathcal{B}_i^{2 \to 2}(0, t_i)} \langle G_i, Z_i \rangle = \text{LMO}_{\mathcal{B}_i^{2 \to 2}(0, t_i)}(G_i)$$

where $\mathcal{B}_i^{2 \to 2}(0, t_i) := \{Z_i \in \mathcal{S}_i : \|Z_i\|_{2 \to 2} \leq t_i\}$ is the spectral norm ball of radius $t_i$ around 0. Thus, the update (6) can equivalently be written as

$$X_i^{k+1} = X_i^k + \text{LMO}_{\mathcal{B}_i^{2 \to 2}(0, t_i^k)}(G_i^k), \tag{7}$$

where $G_i^k$ may be replaced with a momentum term.

Crucially, nothing in this formulation ties us to the spectral norm. The same update structure can be defined over any norm ball, opening the door to an entire family of optimizers whose properties depend on the underlying geometry. This insight has led to several Muon-inspired methods with provable convergence guarantees (Pethick et al., 2025b; Kovalev, 2025; Riabinin et al., 2025b). Scion (Pethick et al., 2025b) removes the restriction to matrix-shaped layers by applying LMO-based updates to all layers, pairing the spectral norm for hidden layers with the $\|\cdot\|_{1 \to \infty}$ norm elsewhere. Gluon (Riabinin et al., 2025b) expands the view even further: it provides a general convergence analysis for LMO updates over arbitrary norm balls, supported by a layer-wise $(L^0, L^1)$-smoothness assumption that captures the heterogeneity of deep learning loss landscapes more accurately than standard smoothness.

## B.2 LAYER-WISE EF21-Muon

The simplified EF21-Muon in Algorithm 1, analyzed in Section 4, omits the layer-wise treatment introduced above. The full structured variant is given in Algorithm 3. Its deterministic counterpart is formalized in Algorithm 2, extending the simplified version studied in Section 4.

Both Algorithms 2 and 3 operate on a per-layer basis. We now briefly describe their structure. For each layer $i$, the parameters are updated via $X_i^{k+1} = \text{LMO}_{\mathcal{B}(X_i^k, t_i^k)}(G_i^k)$ (equivalently, $X_i^{k+1} = X_i^k - \gamma_i^k (G_i^k)^\sharp$, where $\gamma^k = t^k / \|G^k\|_\star$ –see Section C). Next, the algorithms perform the server-to-workers (s2w) compression, following a technique inspired by EF21-P (Gruntkowska et al., 2023). The resulting compressed messages $S_i^k = \mathcal{C}_i^k (X_i^{k+1} - W_i^k)$ are sent to the workers. Each worker then updates the model shift and uses the resulting model estimate $W_i^{k+1}$ to compute the local (stochastic) gradient. This gradient is then used (either directly or within a momentum term) to form the compressed message $R_{i,j}^{k+1}$. This part of the algorithm follows the workers-to-server (w2s) compression strategy of EF21 (Richtárik et al., 2021). The messages $R_{i,j}^{k+1}$ are sent back to the server, which updates the layer-wise gradient estimators via $G_i^{k+1} = \frac{1}{n} \sum_{j=1}^n G_{i,j}^{k+1} = G_i^k + \frac{1}{n} \sum_{j=1}^n R_{i,j}^{k+1}$. This process is repeated until convergence.

---

**Algorithm 2** Deterministic EF21-Muon

---

1: **Parameters:** radii $t_i^k > 0$ / stepsizes $\gamma_i^k$; initial iterate $X^0 = [X_1^0, \ldots, X_p^0] \in \mathcal{S}$ (stored on the server); initial iterate shift $W^0 = X^0$ (stored on the server and the workers); initial gradient estimators $G_j^0 = [G_{1,j}^0, \ldots, G_{p,j}^0] = [\nabla_1 f_j(X^0), \ldots, \nabla_p f_j(X^0)] \in \mathcal{S}$ (stored on the workers), $G^0 = \frac{1}{n} \sum_{j=1}^n G_j^0$ (stored on the server); worker compressors $\mathcal{C}_i^k$; server compressors $\mathcal{C}^k$
2: **for** $k = 0, 1, \ldots, K-1$ **do**
3:     **for** $i = 1, 2, \ldots, p$ **do**
4:         $X_i^{k+1} = \text{LMO}_{\mathcal{B}(X_i^k, t_i^k)}\left(G_i^k\right) = X_i^k - \gamma_i^k \left(G_i^k\right)^\sharp$        *Take LMO-type step*
5:         $S_i^k = \mathcal{C}_i^k(X_i^{k+1} - W_i^k)$        *Compress shifted model on the server*
6:         $W_i^{k+1} = W_i^k + S_i^k$        *Update model shift*
7:         Broadcast $S^k = [S_1^k, \ldots, S_p^k]$ to all workers
8:     **end for**
9:     **for** $j = 1, \ldots, n$ **in parallel do**
10:         **for** $i = 1, 2, \ldots, p$ **do**
11:             $W_i^{k+1} = W_i^k + S_i^k$        *Update model shift*
12:             $R_{i,j}^{k+1} = \mathcal{C}_{i,j}^k(\nabla_i f_j(W^{k+1}) - G_{i,j}^k)$        *Compress shifted gradient*
13:             $G_{i,j}^{k+1} = G_{i,j}^k + R_{i,j}^{k+1}$
14:         **end for**
15:         Broadcast $R_j^{k+1} = [R_{1,j}^{k+1}, \ldots, R_{p,j}^{k+1}]$ to the server
16:     **end for**
17:     **for** $i = 1, \ldots, p$ **do**
18:         $G_i^{k+1} = \frac{1}{n} \sum_{j=1}^n G_{i,j}^{k+1} = G_i^k + \frac{1}{n} \sum_{j=1}^n R_{i,j}^{k+1}$        *Compute gradient estimator*
19:     **end for**
20: **end for**

---

## C    LMO IN MANY GUISES

As outlined in Section 2, the update rule (2)

$$X^{k+1} = X^k + t^k \text{LMO}_{\mathcal{B}(0,1)}\left(G^k\right)$$

admits several equivalent reformulations.

**LMO viewpoint.** The original update (2) is the solution of a simple linear minimization problem over a norm ball

$$X^{k+1} = \text{LMO}_{\mathcal{B}(X^k, t^k)}\left(G^k\right) = \underset{X \in \mathcal{B}(X^k, t^k)}{\arg\min} \left\langle G^k, X \right\rangle,$$

where $\mathcal{B}(X, t) := \{Z \in \mathcal{S} \ : \ \|Z - X\| \leq t\}$. The LMO satisfies

$$\left\langle G, \text{LMO}_{\mathcal{B}(X,t)}\left(G\right)\right\rangle = -t \left\|G\right\|_\star.$$

**Sharp operator viewpoint.** An equivalent perspective is obtained via the *sharp operators* (Nesterov, 2012; Kelner et al., 2014). Define the function $\phi(X) := \frac{1}{2}\|X\|^2$. Its Fenchel conjugate is given by $\phi^\star(G) := \sup_{X \in \mathcal{S}}\{\langle G, X\rangle - \phi(X)\} = \frac{1}{2}\|X\|_\star^2$, and its subdifferential $\partial\phi^\star$ coincides with the sharp operator:

$$\begin{aligned}
\partial\phi^\star(G) &= \{X \in \mathcal{S} : \langle G, X\rangle = \|G\|_\star \|X\|, \|G\|_\star = \|X\|\} \\
&= -\|G\|_\star \text{LMO}_{\mathcal{B}(0,1)}\left(G\right) \\
&= G^\sharp,
\end{aligned}$$

where $G^\sharp := \arg\max_{X \in \mathcal{S}}\{\langle G, X\rangle - \frac{1}{2}\|X\|^2\}$ is the *sharp operator*. Therefore,

$$X^{k+1} = X^k + t^k \text{LMO}_{\mathcal{B}(0,1)}\left(G^k\right) = X^k - \frac{t^k}{\|G^k\|_\star}\left(G^k\right)^\sharp,$$

---

**Algorithm 3** EF21-Muon

---

1: **Parameters:** radii $t_i^k > 0$ / stepsizes $\gamma_i^k$; momentum parameters $\beta_i \in (0, 1]$; initial iterate $X^0 = [X_1^0, \ldots, X_p^0] \in \mathcal{S}$ (stored on the server); initial iterate shift $W^0 = X^0$ (stored on the server and the workers); initial gradient estimators $G_j^0 = [G_{1,j}^0, \ldots, G_{p,j}^0] \in \mathcal{S}$ (stored on the workers); $G^0 = \frac{1}{n} \sum_{j=1}^n G_j^0$ (stored on the server); initial momentum $M_j^0 = [M_{1,j}^0, \ldots, M_{p,j}^0] \in \mathcal{S}$ (stored on the workers); worker compressors $\mathcal{C}_{i,j}^k$; server compressors $\mathcal{C}_i^k$
2: **for** $k = 0, 1, \ldots, K - 1$ **do**
3:     **for** $i = 1, 2, \ldots, p$ **do**
4:         $X_i^{k+1} = \mathrm{LMO}_{\mathcal{B}(X_i^k, t_i^k)}\left(G_i^k\right) = X_i^k - \gamma_i^k \left(G_i^k\right)^\sharp$          *Take LMO-type step*
5:         $S_i^k = \mathcal{C}_i^k(X_i^{k+1} - W_i^k)$          *Compress shifted model on the server*
6:         $W_i^{k+1} = W_i^k + S_i^k$          *Update model shift*
7:         Broadcast $S^k = [S_1^k, \ldots, S_p^k]$ to all workers
8:     **end for**
9:     **for** $j = 1, \ldots, n$ **in parallel do**
10:         **for** $i = 1, 2, \ldots, p$ **do**
11:             $W_i^{k+1} = W_i^k + S_i^k$          *Update model shift*
12:             $M_{i,j}^{k+1} = (1 - \beta_i) M_{i,j}^k + \beta_i \nabla_i f_j(W^{k+1}; \xi_j^{k+1})$          *Compute momentum*
13:             $R_{i,j}^{k+1} = \mathcal{C}_{i,j}^k(M_{i,j}^{k+1} - G_{i,j}^k)$          *Compress shifted gradient*
14:             $G_{i,j}^{k+1} = G_{i,j}^k + R_{i,j}^{k+1}$
15:         **end for**
16:         Broadcast $R_j^{k+1} = [R_{1,j}^{k+1}, \ldots, R_{p,j}^{k+1}]$ to the server
17:     **end for**
18:     **for** $i = 1, 2, \ldots, p$ **do**
19:         $G_i^{k+1} = \frac{1}{n} \sum_{j=1}^n G_{i,j}^{k+1} = G_i^k + \frac{1}{n} \sum_{j=1}^n R_{i,j}^{k+1}$          *Compute gradient estimator*
20:     **end for**
21: **end for**

---

i.e., a normalized steepest descent step with effective stepsize $\gamma^k := t^k / \|G^k\|_\star$.

Two properties of the sharp operator used later are

$$\left\langle X, X^\sharp \right\rangle = \left\| X^\sharp \right\|^2, \qquad \|X\|_\star = \left\| X^\sharp \right\|.$$

**Subdifferential viewpoint.** The negative LMO direction $-\mathrm{LMO}_{\mathcal{B}(0,1)}(A) = \arg\max_{\|Z\|=1} \langle A, Z \rangle$ is a subdifferential of the dual norm $\partial \|\cdot\|_\star (A)$, so (2) can also be written as

$$X^{k+1} = X^k + t^k \mathrm{LMO}_{\mathcal{B}(0,1)}\left(G^k\right) = X^k - t^k H^k$$

for some $H^k \in \partial \|\cdot\|_\star (G^k)$, where by the definition of subdifferential, for any $G^k \neq 0$,

$$\left\langle H^k, G^k \right\rangle = \left\| G^k \right\|_\star, \quad \left\| H^k \right\| = 1. \tag{8}$$

## D  NON-EUCLIDEAN CONTRACTIVE COMPRESSORS

Recall from Definition 1 that a mapping $\mathcal{C} : \mathcal{S} \to \mathcal{S}$ is called a *contractive compression operator* with parameter $\alpha \in (0, 1]$ if, for all $X \in \mathcal{S}$,

$$\mathbb{E}\left[\|\mathcal{C}(X) - X\|^2\right] \leq (1 - \alpha)\|X\|^2. \tag{9}$$

When $\|\cdot\|$ is the Euclidean norm, a wide range of such compressors is available in the literature (Seide et al., 2014; Alistarh et al., 2017; Beznosikov et al., 2020; Richtárik et al., 2021; Szlendak et al., 2021; Horváth et al., 2022). However, when $\|\cdot\|$ is a *non-Euclidean* norm, Euclidean contractivity does not in general imply contractivity with respect to $\|\cdot\|$. Indeed, suppose that $\mathcal{C}$ is contractive with respect to the Euclidean norm. Then, using norm equivalence, for any $X \in \mathcal{S}$,

$$\underline{\rho}^2 \mathbb{E}\left[\|\mathcal{C}(X) - X\|^2\right] \leq \mathbb{E}\left[\|\mathcal{C}(X) - X\|_2^2\right] \leq (1 - \alpha)\|X\|_2^2 \leq \bar{\rho}^2(1 - \alpha)\|X\|^2.$$

Rearranging gives

$$\mathbb{E}\left[\|\mathcal{C}(X) - X\|^2\right] \leq \frac{\bar{\rho}^2}{\underline{\rho}^2}(1 - \alpha)\|X\|^2,$$

and hence $\mathcal{C}$ is *not* contractive with respect to the norm $\|\cdot\|$ unless $\alpha > 1 - \underline{\rho}^2/\bar{\rho}^2$. Consequently, dedicated compressors are needed when working outside the Euclidean setting.

In this section, we first present two simple examples of operators that satisfy condition (9) for *any* norm. These are, however, in general not very practical choices. We then turn to more useful examples of non-Euclidean compressors for several matrix norms of interest.

A simple deterministic example of a contractive compressor is the *scaling* or *damping* operator.

**Definition 8** (Deterministic Damping). For any $X \in \mathcal{S}$, the deterministic damping operator with parameter $\gamma \in (0, 2)$ is defined as

$$\mathcal{C}(X) = \gamma X.$$

For this operator,

$$\mathbb{E}\left[\|\mathcal{C}(X) - X\|^2\right] = (1 - \gamma)^2\|X\|^2,$$

and thus $\mathcal{C}$ satisfies Definition 1 with $\alpha = 1 - (1 - \gamma)^2$ for any $\gamma \in (0, 2)$.

Despite meeting the definition, the deterministic damping operator is of little use in communication-constrained optimization: it merely scales the entire input vector by a constant, without reducing the amount of data to be transmitted. The fact that it formally satisfies the contractive compressor definition is more of a theoretical curiosity. It highlights that the definition captures a broader mathematical property that does not always align with the practical engineering goal of reducing data transmission.

The *random dropout operator* (whose scaled, unbiased variant appears in the literature as the *Bernoulli compressor* (Islamov et al., 2021)) is a simple yet more practically relevant example of a contractive compressor that can reduce communication cost.

**Definition 9** (Random Dropout). For any $X \in \mathcal{S}$, the random dropout operator with a probability parameter $p \in (0, 1]$ is defined as

$$\mathcal{C}(X) = \begin{cases} X & \text{with probability } p, \\ 0 & \text{with probability } 1 - p. \end{cases}$$

Then

$$\mathbb{E}\left[\|\mathcal{C}(X) - X\|^2\right] = (1 - p)\|X\|^2,$$

and hence $\mathcal{C} \in \mathbb{B}(p)$.

The examples of deterministic damping and random dropout apply to any valid norm defined on the space $\mathcal{S}$. However, one can also design compressors directly for the norm of interest. A natural example for both the spectral norm $\|\cdot\|_{2\to 2}$ and the nuclear norm $\|\cdot\|_*$ is based on truncated SVD.

**Definition 10** (Top$K$ SVD compressor). Let $X = U\Sigma V^\top \in \mathbb{R}^{m \times n}$ be a matrix of rank $r$, where $\Sigma = \operatorname{diag}(\sigma_1, \ldots, \sigma_r)$ contains the singular values $\sigma_1 \geq \cdots \geq \sigma_r > 0$. For $K < r$, the *TopK SVD compressor* is defined by

$$\mathcal{C}(X) := U\Sigma_K V^\top,$$

where $\Sigma_K = \operatorname{diag}(\sigma_1, \ldots, \sigma_K, 0, \ldots, 0)$ retains the $K$ largest singular values, setting the rest to zero.

The Top$K$ SVD compressor can be used in conjunction with several commonly used matrix norms:

- **Spectral norm.** The spectral norm, frequently used in LMO-based optimization methods, is defined by $\|X\|_{2\to2} = \sigma_1$. Under this norm, the compression residual is

$$\|X - \mathcal{C}(X)\|_{2\to2} = \sigma_{K+1}.$$

This yields a valid contractive compressor (unless $\sigma_{K+1}^2 = \sigma_1^2$), and Definition 1 is satisfied with parameter $\alpha = 1 - \sigma_{K+1}^2 / \sigma_1^2$.

- **Nuclear norm.** The nuclear norm, dual to the spectral norm, is given by $\|X\|_* = \sum_{i=1}^r \sigma_i$. In this case,

$$\|X - \mathcal{C}(X)\|_* = \sum_{i=K+1}^r \sigma_i,$$

and Definition 1 holds with $\alpha = 1 - \left(\frac{\sum_{i=K+1}^r \sigma_i}{\sum_{i=1}^r \sigma_i}\right)^2$.

- **Frobenius norm.** The Euclidean norm of the matrix can be expressed as $\|X\|_F = \sqrt{\sum_{i=1}^r \sigma_i^2}$. Then,

$$\|X - \mathcal{C}(X)\|_F = \sqrt{\sum_{i=K+1}^r \sigma_i^2}.$$

and so Definition 1 is satisfied with $\alpha = 1 - \frac{\sum_{i=K+1}^r \sigma_i^2}{\sum_{i=1}^r \sigma_i^2}$.

In fact, the Top$K$ SVD compressor is naturally well-suited for a larger family of *Schatten p-norm*, defined in terms of the singular values $\sigma_i$ of a matrix $X$ by

$$\|X\|_{S_p} = \left(\sum_{i=1}^r \sigma_i^p\right)^{1/p}$$

Important special cases include the nuclear norm (or trace norm) for $p = 1$ (i.e., $\|X\|_* = \|X\|_{S_1}$), the Frobenius norm for $p = 2$ (i.e., $\|X\|_F = \|X\|_{S_2}$), and the spectral norm for $p = \infty$ (i.e., $\|X\|_{2\to2} = \|X\|_{S_\infty}$). In general, it is easy to show that the Top$K$ SVD compressor satisfies Definition 1 with respect to the $\|\cdot\|_{S_p}$ norm with

$$\alpha = 1 - \left(\frac{\sum_{i=K+1}^r \sigma_i^p}{\sum_{i=1}^r \sigma_i^p}\right)^{2/p}.$$

**Remark 11.** *For large-scale matrices, computing the exact SVD may be computationally prohibitive. In such cases, one may resort to approximate methods to obtain a stochastic compressor $\widetilde{\mathcal{C}}$ satisfying Definition 1 in expectation:*

$$\mathbb{E}\left[\left\|\widetilde{\mathcal{C}}(X) - X\right\|^2\right] \leq (1 - \alpha + \delta)\|X\|^2,$$

*where $\delta > 0$ quantifies the approximation error and can be made arbitrarily small.*

**Remark 12.** *The expressions for $\alpha$ above depend on the singular values of $X$, and hence $\alpha$ is generally matrix-dependent rather than a uniform constant. For theoretical guarantees, one may take the minimum $\alpha$ observed over a training run. Alternatively, our framework admits a straightforward extension to iteration-dependent compression parameters.*

Beyond Schatten norms, similar ideas can be applied to other structured non-Euclidean norms. Throughout, we let $X_{i:}$, $X_{:j}$, and $X_{ij}$ denote the $i$th row, $j$th column, and $(i, j)$th entry of the matrix $X \in \mathbb{R}^{m \times n}$, respectively.

**Definition 13** (Column-wise $\text{Top}_p K$ compressor)**.** The *column-wise $\text{Top}_p K$ compressor* keeps the $K$ columns with largest $\ell_p$ norm, setting the rest to zero:

$$\mathcal{C}(X)_{:j} = \begin{cases} X_{:j}, & j \in \mathcal{I}_K, \\ 0, & \text{otherwise,} \end{cases}$$

where $\mathcal{I}_K$ indexes the $K$ columns with the largest $\ell_p$ norm.

This operator is naturally suited for the mixed $\ell_{p,q}$ norms ($p, q \geq 1$), defined as

$$\|X\|_{p,q} := \left( \sum_{j=1}^{n} \left( \sum_{i=1}^{m} |X_{ij}|^p \right)^{q/p} \right)^{1/q} = \left( \sum_{j=1}^{n} \|X_{:j}\|_p^q \right)^{1/q},$$

where $\|\cdot\|_p$ is the standard (vector) $\ell_p$ norm. The compression residual satisfies

$$\|X - \mathcal{C}(X)\|_{p,q} = \left( \sum_{j \notin \mathcal{I}_K} \|X_{:j}\|_p^q \right)^{1/q},$$

and hence Definition 1 holds with

$$\alpha = 1 - \left( \frac{\sum_{j \notin \mathcal{I}_K} \|X_{:j}\|_p^q}{\sum_{j=1}^{n} \|X_{:j}\|_p^q} \right)^{2/q}.$$

This general formulation recovers, for example, the $\ell_{2,1}$ norm (commonly used in robust data analysis (Nie et al., 2010)) and the $\ell_{2,2}$ norm (Frobenius norm).

### D.1 Compression via norm selection

A useful perspective on communication reduction in distributed optimization emerges from the connection between compression operators and mappings such as the *sharp operator* and the LMO. Recall that for any norm $\|\cdot\|$ with dual norm $\|\cdot\|_\star$, the sharp operator of $G \in \mathbb{R}^{m \times n}$ is defined as

$$G^\sharp := \arg\max_{X \in \mathbb{R}^{m \times n}} \left\{ \langle G, X \rangle - \frac{1}{2} \|X\|^2 \right\}.$$

Since $\|G\|_\star \, \text{LMO}_{\mathcal{B}(0,1)}(G) = -G^\sharp$, one can view $G^\sharp$ as the LMO over the unit ball of $\|\cdot\|$, scaled by $\|G\|_\star$.

For many norms, $G^\sharp$ naturally acts as a structured compressor. Below, we list several such examples.

- **Nuclear norm.** For the nuclear norm (with dual norm $\|\cdot\|_{2 \to 2}$, the operator/spectral norm), the sharp operator is

$$G^\sharp = \sigma_1 \, u_1 v_1^\top,$$

  where $\sigma_1$, $u_1$, and $v_1$ are the leading singular value and singular vectors of $G$, yielding a *Rank1 compression* via truncated SVD. This operator satisfies Definition 1 with parameter $\alpha = 1/r$, where $r$ is the rank of $G$.

- **Element-wise $\ell_1$ norm.** For the norm $\|X\|_1 = \sum_{i=1}^{m} \sum_{j=1}^{n} |X_{ij}|$ (with dual $\|X\|_\infty = \max_{i,j} |X_{ij}|$), the sharp operator is

$$G^\sharp = \text{Top1}(G) = \|G\|_\infty \, E_{(i^\star j^\star)},$$

  where $(i^\star, j^\star) = \arg\max_{i,j} |G_{ij}|$ and $E_{(i^\star j^\star)}$ is the matrix with a 1 in entry $(i^\star, j^\star)$ and zeros elsewhere. Thus, the sharp operator associated with the $\ell_1$ norm corresponds to *Top1 sparsification*, which satisfies Definition 1 with $\alpha = 1/mn$.

- **Max row sum norm.** For $\|X\|_{\infty\to\infty} = \max_{1\le i\le m}\sum_{j=1}^{n}|X_{ij}|$, the dual norm is $\|X\|_{1,\infty} = \sum_{j=1}^{n}\|X_{:j}\|_{\infty}$, and the sharp operator yields

$$G^{\sharp} = \left(\sum_{j=1}^{n}\|G_{:j}\|_{\infty}\right)[\operatorname{sign}(\operatorname{Top1}(G_{:1}),\ldots,\operatorname{Top1}(G_{:n}))]$$

  i.e., it keeps a single non-zero entry in each column of $G$, with all of these entries equal across columns.

These are only some examples of the compression capabilities of sharp operators. They open the door to compressed server-to-worker communication even in the absence of primal compression, as briefly mentioned in Section 4. Indeed, instead of broadcasting the compressed messages $S^k$ in Algorithms 1 to 3, the server can compute $G^{\sharp}$, transmit this naturally compressed object, and let the workers perform the model update locally. In doing so, we preserve communication efficiency while avoiding the introduction of additional primal compressors.

# E CONVERGENCE ANALYSIS

## E.1 DESCENT LEMMAS

We provide two descent lemmas corresponding to the two smoothness regimes. The first applies to the layer-wise smooth setting.

**Lemma 1** (Descent Lemma I)**.** *Let Assumption 6 hold and consider the update rule* $X_i^{k+1} = X_i^k - \gamma_i^k \left(G_i^k\right)^\sharp$, $i = 1, \ldots, p$, *where* $X^{k+1} = [X_1^{k+1}, \ldots, X_p^{k+1}], X^k = [X_1^k, \ldots, X_p^k], G^k = [G_1^k, \ldots, G_p^k] \in \mathcal{S}$ *and* $\gamma_i^k > 0$. *Then*

$$f(X^{k+1}) \leq f(X^k) + \sum_{i=1}^p \frac{3\gamma_i^k}{2} \left\| \nabla_i f(X^k) - G_i^k \right\|_{(i)\star}^2 - \sum_{i=1}^p \frac{\gamma_i^k}{4} \left\| \nabla_i f(X^k) \right\|_{(i)\star}^2$$
$$- \sum_{i=1}^p \left( \frac{1}{4\gamma_i^k} - \frac{L_i^0}{2} \right) (\gamma_i^k)^2 \left\| G_i^k \right\|_{(i)\star}^2.$$

*Proof.* First, for any $s > 0$, we have

$$\left\| \nabla_i f(X^k) \right\|_{(i)\star}^2 = \left\| \nabla_i f(X^k) - G_i^k + G_i^k \right\|_{(i)\star}^2$$
$$\overset{(28)}{\leq} (1+s) \left\| \nabla_i f(X^k) - G_i^k \right\|_{(i)\star}^2 + \left( 1 + \frac{1}{s} \right) \left\| G_i^k \right\|_{(i)\star}^2,$$

meaning that

$$- \left\| G_i^k \right\|_{(i)\star}^2 \leq \frac{1+s}{1+\frac{1}{s}} \left\| \nabla_i f(X^k) - G_i^k \right\|_{(i)\star}^2 - \frac{1}{1+\frac{1}{s}} \left\| \nabla_i f(X^k) \right\|_{(i)\star}^2$$
$$= s \left\| \nabla_i f(X^k) - G_i^k \right\|_{(i)\star}^2 - \frac{s}{s+1} \left\| \nabla_i f(X^k) \right\|_{(i)\star}^2. \tag{10}$$

Then, using layer-wise smoothness of $f$ and Lemma 14 with $L_i^1 = 0$, we get

$$f(X^{k+1}) \leq f(X^k) + \left\langle \nabla f(X^k), X^{k+1} - X^k \right\rangle + \sum_{i=1}^p \frac{L_i^0}{2} \left\| X_i^k - X_i^{k+1} \right\|_{(i)}^2$$
$$= f(X^k) + \sum_{i=1}^p \left\langle \nabla_i f(X^k), X_i^{k+1} - X_i^k \right\rangle_{(i)} + \sum_{i=1}^p \frac{L_i^0}{2} \left\| X_i^k - X_i^{k+1} \right\|_{(i)}^2$$
$$= f(X^k) - \sum_{i=1}^p \gamma_i^k \left\langle \nabla_i f(X^k) - G_i^k, (G_i^k)^\sharp \right\rangle_{(i)} - \sum_{i=1}^p \gamma_i^k \left\langle G_i^k, (G_i^k)^\sharp \right\rangle_{(i)}$$
$$+ \sum_{i=1}^p \frac{L_i^0}{2} (\gamma_i^k)^2 \left\| (G_i^k)^\sharp \right\|_{(i)\star}^2$$
$$\overset{(33),(34)}{=} f(X^k) - \sum_{i=1}^p \gamma_i^k \left\langle \nabla_i f(X^k) - G_i^k, (G_i^k)^\sharp \right\rangle_{(i)} - \sum_{i=1}^p \frac{\gamma_i^k}{2} \left\| G_i^k \right\|_{(i)\star}^2$$
$$- \sum_{i=1}^p \frac{\gamma_i^k}{2} \left\| G_i^k \right\|_{(i)\star}^2 + \sum_{i=1}^p \frac{L_i^0}{2} (\gamma_i^k)^2 \left\| G_i^k \right\|_{(i)\star}^2$$
$$\overset{(10)}{\leq} f(X^k) - \sum_{i=1}^p \gamma_i^k \left\langle \nabla_i f(X^k) - G_i^k, (G_i^k)^\sharp \right\rangle - \sum_{i=1}^p \frac{\gamma_i^k}{2} \left\| G_i^k \right\|_{(i)\star}^2$$
$$+ \sum_{i=1}^p \frac{\gamma_i^k}{2} s \left\| \nabla_i f(X^k) - G_i^k \right\|_{(i)\star}^2 - \sum_{i=1}^p \frac{\gamma_i^k}{2} \frac{s}{s+1} \left\| \nabla_i f(X^k) \right\|_{(i)\star}^2$$
$$+ \sum_{i=1}^p \frac{L_i^0}{2} (\gamma_i^k)^2 \left\| G_i^k \right\|_{(i)\star}^2.$$

Therefore, applying Fenchel's inequality, we get

$$
\begin{aligned}
f(X^{k+1}) \\
\overset{(29)}{\leq} \quad & f(X^k) + \sum_{i=1}^{p} \left( \frac{\gamma_i^k}{2r} \left\| \nabla_i f(X^k) - G_i^k \right\|_{(i)\star}^2 + \frac{\gamma_i^k r}{2} \left\| (G_i^k)^\sharp \right\|_{(i)}^2 - \frac{\gamma_i^k}{2} \left\| G_i^k \right\|_{(i)\star}^2 \right. \\
& \left. + \frac{\gamma_i^k}{2} s \left\| \nabla_i f(X^k) - G_i^k \right\|_{(i)\star}^2 - \frac{\gamma_i^k}{2} \frac{s}{s+1} \left\| \nabla_i f(X^k) \right\|_{(i)\star}^2 + \frac{L_i^0}{2} (\gamma_i^k)^2 \left\| G_i^k \right\|_{(i)\star}^2 \right) \\
\overset{(34)}{=} \quad & f(X^k) + \sum_{i=1}^{p} \left( \frac{\gamma_i^k}{2r} + \frac{\gamma_i^k s}{2} \right) \left\| \nabla_i f(X^k) - G_i^k \right\|_{(i)\star}^2 - \sum_{i=1}^{p} \frac{\gamma_i^k}{2} \frac{s}{s+1} \left\| \nabla_i f(X^k) \right\|_{(i)\star}^2 \\
& - \sum_{i=1}^{p} \left( \frac{1-r}{2\gamma_i^k} - \frac{L_i^0}{2} \right) (\gamma_i^k)^2 \left\| G_i^k \right\|_{(i)\star}^2
\end{aligned}
$$

for any $r > 0$. Choosing $s = 1$ and $r = 1/2$ finishes the proof. $\qquad \square$

The next lemma is specific to the layer-wise smooth case.

**Lemma 2** (Descent Lemma II). *Let Assumption 8 hold and consider the update rule $X_i^{k+1} = \mathrm{LMO}_{\mathcal{B}(X_i^k, t_i^k)}(G_i^k)$, $i = 1, \ldots, p$, where $X^{k+1} = [X_1^{k+1}, \ldots, X_p^{k+1}], X^k = [X_1^k, \ldots, X_p^k], G^k = [G_1^k, \ldots, G_p^k] \in \mathcal{S}$ and $t_i^k > 0$. Then*

$$
\begin{aligned}
f(X^{k+1}) \quad \leq \quad & f(X^k) + \sum_{i=1}^{p} 2t_i^k \left\| \nabla_i f(X^k) - G_i^k \right\|_{(i)\star} - \sum_{i=1}^{p} t_i^k \left\| \nabla_i f(X^k) \right\|_{(i)\star} \\
& + \sum_{i=1}^{p} \frac{L_i^0 + L_i^1 \left\| \nabla_i f(X^k) \right\|_{(i)\star}}{2} (t_i^k)^2 .
\end{aligned}
$$

*Proof.* Assumption 8 and Lemma 14 give

$$
\begin{aligned}
& f(X^{k+1}) \\
\leq \quad & f(X^k) + \left\langle \nabla f(X^k), X^{k+1} - X^k \right\rangle + \sum_{i=1}^{p} \frac{L_i^0 + L_i^1 \left\| \nabla_i f(X^k) \right\|_{(i)\star}}{2} \left\| X_i^k - X_i^{k+1} \right\|_{(i)}^2 \\
= \quad & f(X^k) + \sum_{i=1}^{p} \left\langle \nabla_i f(X^k), X_i^{k+1} - X_i^k \right\rangle_{(i)} + \sum_{i=1}^{p} \frac{L_i^0 + L_i^1 \left\| \nabla_i f(X^k) \right\|_{(i)\star}}{2} \left\| X_i^k - X_i^{k+1} \right\|_{(i)}^2 \\
= \quad & f(X^k) + \sum_{i=1}^{p} \left( \left\langle \nabla_i f(X^k) - G_i^k, X_i^{k+1} - X_i^k \right\rangle_{(i)} + \left\langle G_i^k, X_i^{k+1} - X_i^k \right\rangle_{(i)} \right) \\
& + \sum_{i=1}^{p} \frac{L_i^0 + L_i^1 \left\| \nabla_i f(X^k) \right\|_{(i)\star}}{2} (t_i^k)^2 \\
\overset{(32)}{=} \quad & f(X^k) + \sum_{i=1}^{p} \left( \left\langle \nabla_i f(X^k) - G_i^k, X_i^{k+1} - X_i^k \right\rangle_{(i)} - t_i^k \left\| G_i^k \right\|_{(i)\star} \right) \\
& + \sum_{i=1}^{p} \frac{L_i^0 + L_i^1 \left\| \nabla_i f(X^k) \right\|_{(i)\star}}{2} (t_i^k)^2 \\
\leq \quad & f(X^k) + \sum_{i=1}^{p} \left( t_i^k \left\| \nabla_i f(X^k) - G_i^k \right\|_{(i)\star} - t_i^k \left\| G_i^k \right\|_{(i)\star} + \frac{L_i^0 + L_i^1 \left\| \nabla_i f(X^k) \right\|_{(i)\star}}{2} (t_i^k)^2 \right),
\end{aligned}
$$

where the last line follows from the Cauchy-Schwarz inequality and the fact that $\left\| X_i^{k+1} - X_i^k \right\|_{(i)} = t_i^k$. Therefore, using triangle inequality, we get

$$
f(X^{k+1})
$$

$$
\begin{aligned}
\leq \quad & f(X^k) + \sum_{i=1}^{p} \left( t_i^k \left\| \nabla_i f(X^k) - G_i^k \right\|_{(i)\star} + t_i^k \left\| \nabla_i f(X^k) - G_i^k \right\|_{(i)\star} - t_i^k \left\| \nabla_i f(X^k) \right\|_{(i)\star} \right) \\
& + \sum_{i=1}^{p} \frac{L_i^0 + L_i^1 \left\| \nabla_i f(X^k) \right\|_{(i)\star}}{2} (t_i^k)^2 \\
= \quad & f(X^k) + \sum_{i=1}^{p} \left( 2 t_i^k \left\| \nabla_i f(X^k) - G_i^k \right\|_{(i)\star} - t_i^k \left\| \nabla_i f(X^k) \right\|_{(i)\star} \right) \\
& + \sum_{i=1}^{p} \frac{L_i^0 + L_i^1 \left\| \nabla_i f(X^k) \right\|_{(i)\star}}{2} (t_i^k)^2 .
\end{aligned}
$$

$\square$

## E.2 AUXILIARY LEMMAS

**Lemma 3.** *The iterates of Algorithm 2 and 3 run with $\mathcal{C}_i^k \in \mathbb{B}(\alpha_P)$ satisfy*

$$
\mathbb{E}\left[ \left\| X_i^{k+1} - W_i^{k+1} \right\|_{(i)}^2 \right] \leq \left( 1 - \frac{\alpha_P}{2} \right) \mathbb{E}\left[ \left\| X_i^k - W_i^k \right\|_{(i)}^2 \right] + \frac{2}{\alpha_P} (\gamma_i^k)^2 \mathbb{E}\left[ \left\| G_i^k \right\|_{(i)\star}^2 \right].
$$

*Proof.* Let $\mathbb{E}_{\mathcal{C}}\left[\cdot\right]$ denote the expectation over the randomness introduced by the compressors. Then

$$
\begin{aligned}
& \mathbb{E}_{\mathcal{C}}\left[ \left\| X_i^{k+1} - W_i^{k+1} \right\|_{(i)}^2 \right] \\
= \quad & \mathbb{E}_{\mathcal{C}}\left[ \left\| W_i^k + \mathcal{C}_i^k(X_i^{k+1} - W_i^k) - X_i^{k+1} \right\|_{(i)}^2 \right] \\
\overset{(1)}{\leq} \quad & (1 - \alpha_P) \left\| X_i^{k+1} - W_i^k \right\|_{(i)}^2 \\
\overset{(28)}{\leq} \quad & (1 - \alpha_P)\left( 1 + \frac{\alpha_P}{2} \right) \left\| X_i^k - W_i^k \right\|_{(i)}^2 + (1 - \alpha_P)\left( 1 + \frac{2}{\alpha_P} \right) \left\| X_i^{k+1} - X_i^k \right\|_{(i)}^2 \\
\overset{(30),(31)}{\leq} \quad & \left( 1 - \frac{\alpha_P}{2} \right) \left\| X_i^k - W_i^k \right\|_2^2 + \frac{2}{\alpha_P} \left\| X_i^{k+1} - X_i^k \right\|_{(i)}^2 .
\end{aligned}
$$

It remains to take full expectation and use the fact that

$$
\left\| X_i^{k+1} - X_i^k \right\|_{(i)} = \gamma_i^k \left\| (G_i^k)^{\sharp} \right\|_{(i)} \overset{(34)}{=} \gamma_i^k \left\| G_i^k \right\|_{(i)\star} .
$$

$\square$

### E.2.1 SMOOTH CASE

**Lemma 4.** *Let Assumptions 7 and 10 hold. Then, the iterates of Algorithm 3 run with $\mathcal{C}_{i,j}^k \in \mathbb{B}_2(\alpha_P)$ satisfy*

$$
\begin{aligned}
\mathbb{E}\left[ \left\| M_{i,j}^{k+1} - G_{i,j}^{k+1} \right\|_2^2 \right] \leq \quad & \left( 1 - \frac{\alpha_D}{2} \right) \mathbb{E}\left[ \left\| M_{i,j}^k - G_{i,j}^k \right\|_2^2 \right] + \frac{6\beta_i^2}{\alpha_D} \mathbb{E}\left[ \left\| M_{i,j}^k - \nabla_i f_j(X^k) \right\|_2^2 \right] \\
& + \frac{6\beta_i^2}{\alpha_D \underline{\rho}_i^2} (L_{i,j}^0)^2 (\gamma_i^k)^2 \mathbb{E}\left[ \left\| G_i^k \right\|_\star^2 \right] \\
& + \frac{6\beta_i^2}{\alpha_D \underline{\rho}_i^2} (L_{i,j}^0)^2 \mathbb{E}\left[ \left\| X_i^{k+1} - W_i^{k+1} \right\|_{(i)}^2 \right] + (1 - \alpha_D)\beta_i^2 \sigma_i^2 .
\end{aligned}
$$

*Proof.* Using the definition of contractive compressors and the algorithm's momentum update rule, we get

$$
\mathbb{E}_{\mathcal{C}}\left[ \left\| M_{i,j}^{k+1} - G_{i,j}^{k+1} \right\|_2^2 \right] = \mathbb{E}_{\mathcal{C}}\left[ \left\| M_{i,j}^{k+1} - G_{i,j}^k - \mathcal{C}_{i,j}^k(M_{i,j}^{k+1} - G_{i,j}^k) \right\|_2^2 \right]
$$

$$\overset{(1)}{\le} \quad (1 - \alpha_D) \left\| M_{i,j}^{k+1} - G_{i,j}^k \right\|_2^2,$$

where $\mathbb{E}_{\mathcal{C}}[\cdot]$ denotes the expectation over the randomness introduced by the compressors. Then, letting $\mathbb{E}_{\xi}[\cdot]$ be the expectation over the stochasticity of the gradients, we have

$$\mathbb{E}\left[ \left\| M_{i,j}^{k+1} - G_{i,j}^{k+1} \right\|_2^2 \right]$$

$$\le \quad \mathbb{E}\left[ \mathbb{E}_{\mathcal{C}}\left[ \left\| M_{i,j}^{k+1} - G_{i,j}^{k+1} \right\|_2^2 \right] \right]$$

$$\le \quad (1 - \alpha_D)\mathbb{E}\left[ \left\| M_{i,j}^{k+1} - G_{i,j}^k \right\|_2^2 \right]$$

$$= \quad (1 - \alpha_D)\mathbb{E}\left[ \mathbb{E}_{\xi}\left[ \left\| (1 - \beta_i)M_{i,j}^k + \beta_i \nabla_i f_j(W^{k+1}; \xi_j^{k+1}) - G_{i,j}^k \right\|_2^2 \right] \right]$$

$$\overset{(13)}{=} \quad (1 - \alpha_D)\mathbb{E}\left[ \left\| (1 - \beta_i)M_{i,j}^k + \beta_i \nabla_i f_j(W^{k+1}) - G_{i,j}^k \right\|_2^2 \right]$$

$$+ (1 - \alpha_D)\beta_i^2 \mathbb{E}\left[ \left\| \nabla_i f_j(W^{k+1}; \xi_j^{k+1}) - \nabla_i f_j(W^{k+1}) \right\|_2^2 \right]$$

$$\overset{(28)}{\le} \quad (1 - \alpha_D)\left(1 + \frac{\alpha_D}{2}\right)\mathbb{E}\left[ \left\| M_{i,j}^k - G_{i,j}^k \right\|_2^2 \right]$$

$$+ (1 - \alpha_D)\left(1 + \frac{2}{\alpha_D}\right)\beta_i^2 \mathbb{E}\left[ \left\| M_{i,j}^k - \nabla_i f_j(W^{k+1}) \right\|_2^2 \right] + (1 - \alpha_D)\beta_i^2 \sigma_i^2,$$

where in the last line we used Assumption 10. Then, Assumption 7 gives

$$\mathbb{E}\left[ \left\| M_{i,j}^{k+1} - G_{i,j}^{k+1} \right\|_2^2 \right]$$

$$\overset{(30),(31)}{\le} \quad \left(1 - \frac{\alpha_D}{2}\right)\mathbb{E}\left[ \left\| M_{i,j}^k - G_{i,j}^k \right\|_2^2 \right] + \frac{2}{\alpha_D}\beta_i^2 \mathbb{E}\left[ \left\| M_{i,j}^k - \nabla_i f_j(W^{k+1}) \right\|_2^2 \right] + (1 - \alpha_D)\beta_i^2 \sigma_i^2$$

$$\overset{(28)}{\le} \quad \left(1 - \frac{\alpha_D}{2}\right)\mathbb{E}\left[ \left\| M_{i,j}^k - G_{i,j}^k \right\|_2^2 \right] + \frac{6\beta_i^2}{\alpha_D}\mathbb{E}\left[ \left\| M_{i,j}^k - \nabla_i f_j(X^k) \right\|_2^2 \right]$$

$$+ \frac{6\beta_i^2}{\alpha_D}\mathbb{E}\left[ \left\| \nabla_i f_j(X^k) - \nabla_i f_j(X^{k+1}) \right\|_2^2 \right]$$

$$+ \frac{6\beta_i^2}{\alpha_D}\mathbb{E}\left[ \left\| \nabla_i f_j(X^{k+1}) - \nabla_i f_j(W^{k+1}) \right\|_2^2 \right] + (1 - \alpha_D)\beta_i^2 \sigma_i^2$$

$$\le \quad \left(1 - \frac{\alpha_D}{2}\right)\mathbb{E}\left[ \left\| M_{i,j}^k - G_{i,j}^k \right\|_2^2 \right] + \frac{6\beta_i^2}{\alpha_D}\mathbb{E}\left[ \left\| M_{i,j}^k - \nabla_i f_j(X^k) \right\|_2^2 \right]$$

$$+ \frac{6\beta_i^2}{\alpha_D \underline{\rho}_i^2}(L_{i,j}^0)^2 \mathbb{E}\left[ \left\| X_i^k - X_i^{k+1} \right\|_{(i)}^2 \right]$$

$$+ \frac{6\beta_i^2}{\alpha_D \underline{\rho}_i^2}(L_{i,j}^0)^2 \mathbb{E}\left[ \left\| X_i^{k+1} - W_i^{k+1} \right\|_{(i)}^2 \right] + (1 - \alpha_D)\beta_i^2 \sigma_i^2.$$

Noting that $\left\| X_i^{k+1} - X_i^k \right\|_{(i)} = \gamma_i^k \left\| (G_i^k)^{\sharp} \right\|_{(i)} \overset{(34)}{=} \gamma_i^k \left\| G_i^k \right\|_{(i)\star}$ finishes the proof. $\qquad\square$

**Lemma 5.** *Let Assumptions 6, 7 and 10 hold. Then, the iterates of Algorithm 3 satisfy*

$$\mathbb{E}\left[ \left\| \nabla_i f_j(X^{k+1}) - M_{i,j}^{k+1} \right\|_2^2 \right]$$

$$\le \quad \left(1 - \frac{\beta_i}{2}\right)\mathbb{E}\left[ \left\| \nabla_i f_j(X^k) - M_{i,j}^k \right\|_2^2 \right] + \frac{2}{\beta_i \underline{\rho}_i^2}(L_{i,j}^0)^2(\gamma_i^k)^2 \mathbb{E}\left[ \left\| G_i^k \right\|_{(i)\star}^2 \right]$$

$$+ \frac{\beta_i^2}{\underline{\rho}_i^2}\left(1 + \frac{2}{\beta_i}\right)(L_{i,j}^0)^2 \mathbb{E}\left[ \left\| X_i^{k+1} - W_i^{k+1} \right\|_{(i)}^2 \right] + \beta_i^2 \sigma_i^2$$

*and*

$$\mathbb{E}\left[ \left\| \nabla_i f(X^{k+1}) - M_i^{k+1} \right\|_2^2 \right]$$

$$\leq \quad \left(1 - \frac{\beta_i}{2}\right) \left\|\nabla_i f(X^k) - M_i^k\right\|_2^2 + \frac{2}{\beta_i \underline{\rho}_i^2} (L_i^0)^2 (\gamma_i^k)^2 \mathbb{E}\left[\left\|G_i^k\right\|_{(i)\star}^2\right]$$

$$+ \frac{\beta_i^2}{\underline{\rho}_i^2}\left(1 + \frac{2}{\beta_i}\right)(L_i^0)^2 \mathbb{E}\left[\left\|X_i^{k+1} - W_i^{k+1}\right\|_{(i)}^2\right] + \frac{\beta_i^2 \sigma_i^2}{n},$$

*where* $M_i^k := \frac{1}{n}\sum_{j=1}^n M_{i,j}^k.$

*Proof.* Using the momentum update rule and letting $\mathbb{E}_\xi[\cdot]$ be the expectation over the stochasticity of the gradients, we get

$$\mathbb{E}_\xi\left[\left\|\nabla_i f_j(X^{k+1}) - M_{i,j}^{k+1}\right\|_2^2\right]$$

$$= \quad \mathbb{E}_\xi\left[\left\|\nabla_i f_j(X^{k+1}) - (1-\beta_i)M_{i,j}^k - \beta_i \nabla_i f_j(W^{k+1}; \xi_j^{k+1})\right\|_2^2\right]$$

$$\overset{(13)}{=} \quad \left\|\nabla_i f_j(X^{k+1}) - (1-\beta_i)M_{i,j}^k - \beta_i \nabla_i f_j(W^{k+1})\right\|_2^2$$

$$+ \beta_i^2 \mathbb{E}_\xi\left[\left\|\nabla_i f_j(W^{k+1}; \xi_j^{k+1}) - \nabla_i f_j(W^{k+1})\right\|_2^2\right]$$

$$\overset{(28)}{\leq} \quad (1-\beta_i)^2\left(1 + \frac{\beta_i}{2}\right)\left\|\nabla_i f_j(X^{k+1}) - M_{i,j}^k\right\|_2^2$$

$$+ \beta_i^2\left(1 + \frac{2}{\beta_i}\right)\left\|\nabla_i f_j(X^{k+1}) - \nabla_i f_j(W^{k+1})\right\|_2^2$$

$$+ \beta_i^2 \mathbb{E}_\xi\left[\left\|\nabla_i f_j(W^{k+1}; \xi_j^{k+1}) - \nabla_i f_j(W^{k+1})\right\|_2^2\right]$$

$$\overset{(30)}{\leq} \quad (1-\beta_i)\left\|\nabla_i f_j(X^{k+1}) - M_{i,j}^k\right\|_2^2$$

$$+ \beta_i^2\left(1 + \frac{2}{\beta_i}\right)\left\|\nabla_i f_j(X^{k+1}) - \nabla_i f_j(W^{k+1})\right\|_2^2 + \beta_i^2 \sigma_i^2,$$

where in the last line we used Assumption 5. Then, Assumption 7 gives

$$\mathbb{E}\left[\left\|\nabla_i f_j(X^{k+1}) - M_{i,j}^{k+1}\right\|_2^2\right]$$

$$= \quad \mathbb{E}\left[\mathbb{E}_\xi\left[\left\|\nabla_i f_j(X^{k+1}) - M_{i,j}^{k+1}\right\|_2^2\right]\right]$$

$$\overset{(28)}{\leq} \quad (1-\beta_i)\left(1 + \frac{\beta_i}{2}\right)\mathbb{E}\left[\left\|\nabla_i f_j(X^k) - M_{i,j}^k\right\|_2^2\right]$$

$$+ (1-\beta_i)\left(1 + \frac{2}{\beta_i}\right)\mathbb{E}\left[\left\|\nabla_i f_j(X^{k+1}) - \nabla_i f_j(X^k)\right\|_2^2\right]$$

$$+ \beta_i^2\left(1 + \frac{2}{\beta_i}\right)\mathbb{E}\left[\left\|\nabla_i f_j(X^{k+1}) - \nabla_i f_j(W^{k+1})\right\|_2^2\right] + \beta_i^2 \sigma_i^2$$

$$\overset{(30),(31)}{\leq} \quad \left(1 - \frac{\beta_i}{2}\right)\mathbb{E}\left[\left\|\nabla_i f_j(X^k) - M_{i,j}^k\right\|_2^2\right] + \frac{2}{\beta_i \underline{\rho}_i^2}(L_{i,j}^0)^2 \mathbb{E}\left[\left\|X_i^{k+1} - X_i^k\right\|_{(i)}^2\right]$$

$$+ \frac{\beta_i^2}{\underline{\rho}_i^2}\left(1 + \frac{2}{\beta_i}\right)(L_{i,j}^0)^2 \mathbb{E}\left[\left\|X_i^{k+1} - W_i^{k+1}\right\|_{(i)}^2\right] + \beta_i^2 \sigma_i^2.$$

To prove the second part of the statement, define $\nabla_i f(X; \xi^k) := \frac{1}{n}\sum_{i=1}^n \nabla_i f_j(X; \xi_j^k)$. Then $M_i^{k+1} = (1-\beta_i)M_i^k + \beta_i \nabla_i f(W^{k+1}; \xi^{k+1})$, so following similar steps as above, we get

$$\mathbb{E}\left[\left\|\nabla_i f(X^{k+1}) - M_i^{k+1}\right\|_2^2\right]$$

$$= \quad \mathbb{E}\left[\mathbb{E}_\xi\left[\left\|\nabla_i f(X^{k+1}) - (1-\beta_i)M_i^k - \beta_i \nabla_i f(W^{k+1}; \xi^{k+1})\right\|_2^2\right]\right]$$

$$\overset{(13)}{=} \quad \mathbb{E}\left[\left\|\nabla_i f(X^{k+1}) - (1-\beta_i)M_i^k - \beta_i \nabla_i f(W^{k+1})\right\|_2^2\right]$$

$$+\beta_i^2 \mathbb{E}\left[\mathbb{E}_\xi\left[\left\|\nabla_i f(W^{k+1};\xi^{k+1}) - \nabla_i f(W^{k+1})\right\|_2^2\right]\right]$$

$$\overset{(28)}{\leq} \quad (1-\beta_i)^2\left(1+\frac{\beta_i}{2}\right)\mathbb{E}\left[\left\|\nabla_i f(X^{k+1}) - M_i^k\right\|_2^2\right]$$

$$+\beta_i^2\left(1+\frac{2}{\beta_i}\right)\mathbb{E}\left[\left\|\nabla_i f(X^{k+1}) - \nabla_i f(W^{k+1})\right\|_2^2\right]$$

$$+\beta_i^2\mathbb{E}\left[\mathbb{E}_\xi\left[\left\|\nabla_i f(W^{k+1};\xi^{k+1}) - \nabla_i f(W^{k+1})\right\|_2^2\right]\right]$$

$$\overset{(30)}{\leq} \quad (1-\beta_i)\mathbb{E}\left[\left\|\nabla_i f(X^{k+1}) - M_i^k\right\|_2^2\right]$$

$$+\beta_i^2\left(1+\frac{2}{\beta_i}\right)\mathbb{E}\left[\left\|\nabla_i f(X^{k+1}) - \nabla_i f(W^{k+1})\right\|_2^2\right] + \frac{\beta_i^2\sigma_i^2}{n}$$

$$\overset{(28)}{\leq} \quad (1-\beta_i)\left(1+\frac{\beta_i}{2}\right)\mathbb{E}\left[\left\|\nabla_i f(X^k) - M_i^k\right\|_2^2\right]$$

$$+(1-\beta_i)\left(1+\frac{2}{\beta_i}\right)\mathbb{E}\left[\left\|\nabla_i f(X^{k+1}) - \nabla_i f(X^k)\right\|_2^2\right]$$

$$+\beta_i^2\left(1+\frac{2}{\beta_i}\right)\mathbb{E}\left[\left\|\nabla_i f(X^{k+1}) - \nabla_i f(W^{k+1})\right\|_2^2\right] + \frac{\beta_i^2\sigma_i^2}{n}$$

$$\overset{(30),(31)}{\leq} \quad \left(1-\frac{\beta_i}{2}\right)\mathbb{E}\left[\left\|\nabla_i f(X^k) - M_i^k\right\|_2^2\right] + \frac{2}{\beta_i\underline{\rho}_i^2}(L_i^0)^2\mathbb{E}\left[\left\|X_i^{k+1} - X_i^k\right\|_{(i)}^2\right]$$

$$+\frac{\beta_i^2}{\underline{\rho}_i^2}\left(1+\frac{2}{\beta_i}\right)(L_i^0)^2\mathbb{E}\left[\left\|X_i^{k+1} - W_i^{k+1}\right\|_{(i)}^2\right] + \frac{\beta_i^2\sigma_i^2}{n}.$$

It remains to use the fact that $\left\|X_i^{k+1} - X_i^k\right\|_{(i)} = \gamma_i^k\left\|\left(G_i^k\right)^\sharp\right\|_{(i)} \overset{(34)}{=} \gamma_i^k\left\|G_i^k\right\|_{(i)\star}$. $\qquad\square$

### E.2.2 GENERALIZED SMOOTH CASE

**Lemma 6.** *Let Assumption 9 hold. Then, the iterates of Algorithm 2 run with $\mathcal{C}_i^k \equiv \mathcal{I}$ (the identity compressor) and $\mathcal{C}_{i,j}^k \in \mathbb{B}_\star(\alpha_D)$ satisfy*

$$\mathbb{E}\left[\left\|\nabla_i f_j(X^{k+1}) - G_{i,j}^{k+1}\right\|_{(i)\star}\Big|\, X^{k+1},G^k\right]$$
$$\leq \quad \sqrt{1-\alpha_D}\left\|\nabla_i f_j(X^k) - G_{i,j}^k\right\|_{(i)\star} + \sqrt{1-\alpha_D}\left(L_{i,j}^0 + L_{i,j}^1\left\|\nabla_i f_j(X^k)\right\|_{(i)\star}\right)t_i^k.$$

*Proof.* The algorithm's update rule and Jensen's inequality give

$$\mathbb{E}\left[\left\|\nabla_i f_j(X^{k+1}) - G_{i,j}^{k+1}\right\|_{(i)\star}\Big|\, X^{k+1},G^k\right]$$

$$= \quad \mathbb{E}\left[\sqrt{\left\|\nabla_i f_j(X^{k+1}) - G_{i,j}^k - \mathcal{C}_{i,j}^k(\nabla_i f_j(X^{k+1}) - G_{i,j}^k)\right\|_{(i)\star}^2}\,\Big|\, X^{k+1},G^k\right]$$

$$\leq \quad \sqrt{\mathbb{E}\left[\left\|\nabla_i f_j(X^{k+1}) - G_{i,j}^k - \mathcal{C}_{i,j}^k(\nabla_i f_j(X^{k+1}) - G_{i,j}^k)\right\|_{(i)\star}^2\,\Big|\, X^{k+1},G^k\right]}$$

$$\leq \quad \sqrt{1-\alpha_D}\left\|\nabla_i f_j(X^{k+1}) - G_{i,j}^k\right\|_{(i)\star}$$

$$\leq \quad \sqrt{1-\alpha_D}\left\|\nabla_i f_j(X^k) - G_{i,j}^k\right\|_{(i)\star} + \sqrt{1-\alpha_D}\left\|\nabla_i f_j(X^{k+1}) - \nabla_i f_j(X^k)\right\|_{(i)\star}$$

$$\leq \quad \sqrt{1-\alpha_D}\left\|\nabla_i f_j(X^k) - G_{i,j}^k\right\|_{(i)\star}$$

$$+\sqrt{1-\alpha_D}\left(L_{i,j}^0 + L_{i,j}^1\left\|\nabla_i f_j(X^k)\right\|_{(i)\star}\right)\left\|X_i^{k+1} - X_i^k\right\|_{(i)}.$$

where $\left\|X_i^{k+1} - X_i^k\right\|_{(i)} = t_i^k$. $\qquad\square$

**Lemma 7.** *Let Assumptions 9 and 10 hold. Then, the iterates of Algorithm 3 run with $\mathcal{C}_i^k \equiv \mathcal{I}$ (the identity compressor) and $\mathcal{C}_{i,j}^k \in \mathbb{B}_2(\alpha_D)$ satisfy*

$$
\mathbb{E}\left[\left\|M_{i,j}^{k+1} - G_{i,j}^{k+1}\right\|_2 \middle| X^{k+1}, M_{i,j}^k, G_{i,j}^k\right]
$$
$$
\leq \sqrt{1-\alpha_D}\left\|M_{i,j}^k - G_{i,j}^k\right\|_2 + \sqrt{1-\alpha_D}\beta_i\left\|M_{i,j}^k - \nabla_i f_j(X^k)\right\|_2
$$
$$
+ \frac{\sqrt{1-\alpha_D}\beta_i}{\underline{\rho}_i}\left(L_{i,j}^0 + L_{i,j}^1\left\|\nabla_i f_j(X^k)\right\|_{(i)\star}\right)t_i^k + \sqrt{1-\alpha_D}\beta_i\sigma_i.
$$

*Proof.* Using the definition of contractive compressors and triangle inequality, we get

$$
\mathbb{E}\left[\left\|M_{i,j}^{k+1} - G_{i,j}^{k+1}\right\|_2 \middle| M_{i,j}^{k+1}, G_{i,j}^k\right]
$$
$$
= \mathbb{E}\left[\sqrt{\left\|M_{i,j}^{k+1} - G_{i,j}^k - \mathcal{C}_{i,j}^k(M_{i,j}^{k+1} - G_{i,j}^k)\right\|_2^2} \middle| M_{i,j}^{k+1}, G_{i,j}^k\right]
$$
$$
\leq \sqrt{\mathbb{E}\left[\left\|M_{i,j}^{k+1} - G_{i,j}^k - \mathcal{C}_{i,j}^k(M_{i,j}^{k+1} - G_{i,j}^k)\right\|_2^2 \middle| M_{i,j}^{k+1}, G_{i,j}^k\right]}
$$
$$
\overset{(1)}{\leq} \sqrt{1-\alpha_D}\left\|M_{i,j}^{k+1} - G_{i,j}^k\right\|_2
$$
$$
= \sqrt{1-\alpha_D}\left\|(1-\beta_i)M_{i,j}^k + \beta_i\nabla_i f_j(X^{k+1};\xi_j^{k+1}) - G_{i,j}^k\right\|_2.
$$

Hence,

$$
\mathbb{E}\left[\left\|M_{i,j}^{k+1} - G_{i,j}^{k+1}\right\|_2 \middle| X^{k+1}, M_{i,j}^k, G_{i,j}^k\right]
$$
$$
= \mathbb{E}\left[\mathbb{E}\left[\left\|M_{i,j}^{k+1} - G_{i,j}^{k+1}\right\|_2 \middle| M_{i,j}^{k+1}, G_{i,j}^k\right] \middle| X^{k+1}, M_{i,j}^k, G_{i,j}^k\right]
$$
$$
\leq \sqrt{1-\alpha_D}\mathbb{E}\left[\left\|(1-\beta_i)M_{i,j}^k + \beta_i\nabla_i f_j(X^{k+1};\xi_j^{k+1}) - G_{i,j}^k\right\|_2 \middle| X^{k+1}, M_{i,j}^k, G_{i,j}^k\right]
$$
$$
\leq \sqrt{1-\alpha_D}\mathbb{E}\left[\left\|(1-\beta_i)M_{i,j}^k + \beta_i\nabla_i f_j(X^{k+1}) - G_{i,j}^k\right\|_2 \middle| X^{k+1}, M_{i,j}^k, G_{i,j}^k\right]
$$
$$
+ \sqrt{1-\alpha_D}\beta_i\mathbb{E}\left[\left\|\nabla_i f_j(X^{k+1};\xi_j^{k+1}) - \nabla_i f_j(X^{k+1})\right\|_2 \middle| X^{k+1}, M_{i,j}^k, G_{i,j}^k\right]
$$
$$
\overset{(10)}{\leq} \sqrt{1-\alpha_D}\left\|M_{i,j}^k - G_{i,j}^k\right\|_2 + \sqrt{1-\alpha_D}\beta_i\left\|M_{i,j}^k - \nabla_i f_j(X^{k+1})\right\|_2 + \sqrt{1-\alpha_D}\beta_i\sigma_i
$$
$$
\leq \sqrt{1-\alpha_D}\left\|M_{i,j}^k - G_{i,j}^k\right\|_2 + \sqrt{1-\alpha_D}\beta_i\left\|M_{i,j}^k - \nabla_i f_j(X^k)\right\|_2
$$
$$
+ \sqrt{1-\alpha_D}\beta_i\left\|\nabla_i f_j(X^k) - \nabla_i f_j(X^{k+1})\right\|_2 + \sqrt{1-\alpha_D}\beta_i\sigma_i
$$
$$
\overset{(9)}{\leq} \sqrt{1-\alpha_D}\left\|M_{i,j}^k - G_{i,j}^k\right\|_2 + \sqrt{1-\alpha_D}\beta_i\left\|M_{i,j}^k - \nabla_i f_j(X^k)\right\|_2
$$
$$
+ \frac{\sqrt{1-\alpha_D}\beta_i}{\underline{\rho}_i}\left(L_{i,j}^0 + L_{i,j}^1\left\|\nabla_i f_j(X^k)\right\|_{(i)\star}\right)\left\|X_i^k - X_i^{k+1}\right\|_{(i)} + \sqrt{1-\alpha_D}\beta_i\sigma_i.
$$

Using the fact that $\left\|X_i^k - X_i^{k+1}\right\| = t_i^k$ finishes the proof. $\qquad\square$

**Lemma 8.** *Let Assumptions 8, 9 and 10 hold. Then, the iterates of Algorithm 3 run with $\mathcal{C}_i^k \equiv \mathcal{I}$ (the identity compressor) and $t_i^k \equiv t_i$ satisfy*

$$
\mathbb{E}\left[\left\|M_i^{k+1} - \nabla_i f(X^{k+1})\right\|_2\right] \leq (1-\beta_i)^{k+1}\mathbb{E}\left[\left\|M_i^0 - \nabla_i f(X^0)\right\|_2\right] + \frac{t_i \bar{L}_i^0}{\beta_i \underline{\rho}_i}
$$
$$
+ \frac{t_i}{\underline{\rho}_i}\frac{1}{n}\sum_{j=1}^n L_{i,j}^1\sum_{l=0}^k (1-\beta_i)^{k+1-l}\mathbb{E}\left[\left\|\nabla_i f_j(X^l)\right\|_{(i)\star}\right] + \sigma_i\sqrt{\frac{\beta_i}{n}}
$$

*and*

$$
\frac{1}{n}\sum_{j=1}^n\mathbb{E}\left[\left\|M_{i,j}^{k+1} - \nabla_i f_j(X^{k+1})\right\|_2\right] \leq (1-\beta_i)\frac{1}{n}\sum_{j=1}^n\mathbb{E}\left[\left\|M_{i,j}^k - \nabla_i f_j(X^k)\right\|_2\right] + t_i\frac{(1-\beta_i)\bar{L}_i^0}{\underline{\rho}_i}
$$

$$+ t_i \frac{1 - \beta_i}{\underline{\rho}_i} \frac{1}{n} \sum_{j=1}^n L_{i,j}^1 \mathbb{E}\left[\left\|\nabla_i f_j(X^k)\right\|_{(i)\star}\right] + \beta_i \sigma_i,$$

where $M_i^k := \frac{1}{n} \sum_{j=1}^n M_{i,j}^k$.

*Proof.* The proof uses techniques similar to those in Cutkosky & Mehta (2020, Theorem 1). First, using the momentum update rule, we can write

$$
\begin{aligned}
M_{i,j}^{k+1} &= (1 - \beta_i) M_{i,j}^k + \beta_i \nabla_i f_j(X^{k+1}; \xi_j^{k+1}) \\
&= (1 - \beta_i)\left(M_{i,j}^k - \nabla_i f_j(X^k)\right) + (1 - \beta_i)\left(\nabla_i f_j(X^k) - \nabla_i f_j(X^{k+1})\right) \\
&\quad + \beta_i\left(\nabla_i f_j(X^{k+1}; \xi_j^{k+1}) - \nabla_i f_j(X^{k+1})\right) + \nabla_i f_j(X^{k+1}),
\end{aligned}
$$

and hence

$$U_{1,i,j}^{k+1} = (1 - \beta_i) U_{1,i,j}^k + (1 - \beta_i) U_{2,i,j}^k + \beta_i U_{3,i,j}^{k+1},$$

where we define $U_{1,i,j}^k := M_{i,j}^k - \nabla_i f_j(X^k)$, $U_{2,i,j}^k := \nabla_i f_j(X^k) - \nabla_i f_j(X^{k+1})$ and $U_{3,i,j}^k := \nabla_i f_j(X^k; \xi_j^k) - \nabla_i f_j(X^k)$. Unrolling the recursion gives

$$U_{1,i,j}^{k+1} = (1 - \beta_i)^{k+1} U_{1,i,j}^0 + \sum_{l=0}^k (1 - \beta_i)^{k+1-l} U_{2,i,j}^l + \beta_i \sum_{l=0}^k (1 - \beta_i)^{k-l} U_{3,i,j}^{l+1}.$$

Hence, using the triangle inequality,

$$
\begin{aligned}
&\mathbb{E}\left[\left\|\frac{1}{n} \sum_{j=1}^n U_{1,i,j}^{k+1}\right\|_2\right] \\
&\leq (1 - \beta_i)^{k+1} \mathbb{E}\left[\left\|\frac{1}{n} \sum_{j=1}^n U_{1,i,j}^0\right\|_2\right] + \mathbb{E}\left[\left\|\sum_{l=0}^k (1 - \beta_i)^{k+1-l} \frac{1}{n} \sum_{j=1}^n U_{2,i,j}^l\right\|_2\right] \\
&\quad + \beta_i \mathbb{E}\left[\left\|\sum_{l=0}^k (1 - \beta_i)^{k-l} \frac{1}{n} \sum_{j=1}^n U_{3,i,j}^{l+1}\right\|_2\right].
\end{aligned}
\tag{11}
$$

Let us now bound the last two terms of the inequality above. First, triangle inequality and Assumption 9 give

$$
\begin{aligned}
&\mathbb{E}\left[\left\|\sum_{l=0}^k (1 - \beta_i)^{k+1-l} \frac{1}{n} \sum_{j=1}^n U_{2,i,j}^l\right\|_2\right] \\
&\leq \frac{1}{n} \sum_{j=1}^n \sum_{l=0}^k (1 - \beta_i)^{k+1-l} \mathbb{E}\left[\left\|U_{2,i,j}^l\right\|_2\right] \\
&= \frac{1}{n} \sum_{j=1}^n \sum_{l=0}^k (1 - \beta_i)^{k+1-l} \mathbb{E}\left[\left\|\nabla_i f_j(X^l) - \nabla_i f_j(X^{l+1})\right\|_2\right] \\
&\overset{(9)}{\leq} \frac{1}{\underline{\rho}_i} \frac{1}{n} \sum_{j=1}^n \sum_{l=0}^k (1 - \beta_i)^{k+1-l} \mathbb{E}\left[\left(L_{i,j}^0 + L_{i,j}^1 \left\|\nabla_i f_j(X^l)\right\|_{(i)\star}\right) \left\|X_i^l - X_i^{l+1}\right\|_{(i)}\right] \\
&= \frac{t_i}{\underline{\rho}_i} \frac{1}{n} \sum_{j=1}^n \sum_{l=0}^k (1 - \beta_i)^{k+1-l} L_{i,j}^0 + \frac{t_i}{\underline{\rho}_i} \frac{1}{n} \sum_{j=1}^n L_{i,j}^1 \sum_{l=0}^k (1 - \beta_i)^{k+1-l} \mathbb{E}\left[\left\|\nabla_i f_j(X^l)\right\|_{(i)\star}\right] \\
&\leq \frac{t_i \bar{L}_i^0}{\beta_i \underline{\rho}_i} + \frac{t_i}{\underline{\rho}_i} \frac{1}{n} \sum_{j=1}^n L_{i,j}^1 \sum_{l=0}^k (1 - \beta_i)^{k+1-l} \mathbb{E}\left[\left\|\nabla_i f_j(X^l)\right\|_{(i)\star}\right],
\end{aligned}
$$

and using Jensen's inequality, the last term can be bounded as

$$
\mathbb{E}\left[\left\|\sum_{l=0}^{k}(1-\beta_i)^{k-l}\frac{1}{n}\sum_{j=1}^{n}U_{3,i,j}^{l+1}\right\|_2\right]
$$

$$
\leq \sqrt{\mathbb{E}\left[\left\|\sum_{l=0}^{k}(1-\beta_i)^{k-l}\frac{1}{n}\sum_{j=1}^{n}U_{3,i,j}^{l+1}\right\|_2^2\right]} \stackrel{(10)}{=} \sqrt{\sum_{l=0}^{k}(1-\beta_i)^{2(k-l)}\frac{1}{n^2}\sum_{j=1}^{n}\mathbb{E}\left[\left\|U_{3,i,j}^{l+1}\right\|_2^2\right]}
$$

$$
\stackrel{(10)}{\leq} \sqrt{\sum_{l=0}^{k}(1-\beta_i)^{2(k-l)}\frac{1}{n^2}\sum_{j=1}^{n}\sigma_i^2} = \frac{\sigma_i}{\sqrt{n}}\sqrt{\sum_{l=0}^{k}(1-\beta_i)^{2l}} \leq \frac{\sigma_i}{\sqrt{n\beta_i(2-\beta_i)}} \leq \frac{\sigma_i}{\sqrt{n\beta_i}}.
$$

Substituting this in (11) yields

$$
\mathbb{E}\left[\left\|\frac{1}{n}\sum_{j=1}^{n}U_{1,i,j}^{k+1}\right\|_2\right] \leq (1-\beta_i)^{k+1}\mathbb{E}\left[\left\|\frac{1}{n}\sum_{j=1}^{n}U_{1,i,j}^{0}\right\|_2\right] + \frac{t_i\bar{L}_i^0}{\beta_i\underline{\rho}_i}
$$

$$
+\frac{t_i}{\underline{\rho}_i}\frac{1}{n}\sum_{j=1}^{n}L_{i,j}^1\sum_{l=0}^{k}(1-\beta_i)^{k+1-l}\mathbb{E}\left[\left\|\nabla_i f_j(X^l)\right\|_{(i)\star}\right] + \beta_i\frac{\sigma_i}{\sqrt{n\beta_i}}.
$$

To prove the second inequality, recall that $U_{1,i,j}^{k+1} = (1-\beta_i)U_{1,i,j}^{k} + (1-\beta_i)U_{2,i,j}^{k} + \beta_i U_{3,i,j}^{k+1}$. Hence, taking norms, averaging, and using the triangle inequality,

$$
\frac{1}{n}\sum_{j=1}^{n}\mathbb{E}\left[\left\|U_{1,i,j}^{k+1}\right\|_2\right] \leq (1-\beta_i)\frac{1}{n}\sum_{j=1}^{n}\mathbb{E}\left[\left\|U_{1,i,j}^{k}\right\|_2\right] + (1-\beta_i)\frac{1}{n}\sum_{j=1}^{n}\mathbb{E}\left[\left\|U_{2,i,j}^{k}\right\|_2\right]
$$

$$
+\beta_i\frac{1}{n}\sum_{j=1}^{n}\mathbb{E}\left[\left\|U_{3,i,j}^{k+1}\right\|_2\right], \tag{12}
$$

where the last two terms can be bounded as

$$
\frac{1}{n}\sum_{j=1}^{n}\mathbb{E}\left[\left\|U_{2,i,j}^{k}\right\|_2\right] = \frac{1}{n}\sum_{j=1}^{n}\mathbb{E}\left[\left\|\nabla_i f_j(X^k) - \nabla_i f_j(X^{k+1})\right\|_2\right]
$$

$$
\stackrel{(9)}{\leq} \frac{1}{\underline{\rho}_i}\frac{1}{n}\sum_{j=1}^{n}\mathbb{E}\left[\left(L_{i,j}^0 + L_{i,j}^1\left\|\nabla_i f_j(X^k)\right\|_{(i)\star}\right)\left\|X_i^k - X_i^{k+1}\right\|_{(i)}\right]
$$

$$
= t_i\frac{\bar{L}_i^0}{\underline{\rho}_i} + \frac{t_i}{\underline{\rho}_i}\frac{1}{n}\sum_{j=1}^{n}L_{i,j}^1\mathbb{E}\left[\left\|\nabla_i f_j(X^k)\right\|_{(i)\star}\right]
$$

and

$$
\frac{1}{n}\sum_{j=1}^{n}\mathbb{E}\left[\left\|U_{3,i,j}^{k+1}\right\|_2\right] = \frac{1}{n}\sum_{j=1}^{n}\mathbb{E}\left[\left\|\nabla_i f_j(X^{k+1};\xi_j^k) - \nabla_i f_j(X^{k+1})\right\|_2\right] \stackrel{(10)}{\leq} \sigma_i.
$$

It remains to substitute this in (12) to obtain

$$
\frac{1}{n}\sum_{j=1}^{n}\mathbb{E}\left[\left\|U_{1,i,j}^{k+1}\right\|_2\right] \leq (1-\beta_i)\frac{1}{n}\sum_{j=1}^{n}\mathbb{E}\left[\left\|U_{1,i,j}^{k}\right\|_2\right] + t_i\frac{(1-\beta_i)\bar{L}_i^0}{\underline{\rho}_i}
$$

$$
+t_i\frac{1-\beta_i}{\underline{\rho}_i}\frac{1}{n}\sum_{j=1}^{n}L_{i,j}^1\mathbb{E}\left[\left\|\nabla_i f_j(X^k)\right\|_{(i)\star}\right] + \beta_i\sigma_i.
$$

$\square$

**Lemma 9.** *Let Assumptions 1 and 8 hold. Then*

$$\sum_{i=1}^{p} \frac{\|\nabla_i f(X)\|_{(i)\star}^2}{2\left(L_i^0 + L_i^1 \|\nabla_i f(X)\|_{(i)\star}\right)} \leq f(X) - f^\star$$

*for any $X = [X_1, \ldots, X_p] \in \mathcal{S}$.*

*Proof.* Let $Y = [Y_1, \ldots, Y_p] \in \mathcal{S}$, where $Y_i = X_i - \frac{\|\nabla_i f(X)\|_{(i)\star}}{L_i^0 + L_i^1 \|\nabla_i f(X)\|_{(i)\star}} H_i$ for some $H_i \in \partial \|\cdot\|_{(i)\star} (\nabla_i f(X))$. Then, Lemma 14 and the definition of subdifferential give

$$
\begin{aligned}
f(Y) &\leq f(X) + \langle \nabla f(X), Y - X \rangle + \sum_{i=1}^{p} \frac{L_i^0 + L_i^1 \|\nabla_i f(X)\|_{(i)\star}}{2} \|X_i - Y_i\|_{(i)}^2 \\
&= f(X) + \sum_{i=1}^{p} \langle \nabla_i f(X), Y_i - X_i \rangle_{(i)} + \sum_{i=1}^{p} \frac{L_i^0 + L_i^1 \|\nabla_i f(X)\|_{(i)\star}}{2} \|X_i - Y_i\|_{(i)}^2 \\
&= f(X) - \sum_{i=1}^{p} \frac{\|\nabla_i f(X)\|_{(i)\star}}{L_i^0 + L_i^1 \|\nabla_i f(X)\|_{(i)\star}} \langle \nabla_i f(X), H_i \rangle_{(i)} \\
&\quad + \sum_{i=1}^{p} \left( \frac{L_i^0 + L_i^1 \|\nabla_i f(X)\|_{(i)\star}}{2} \frac{\|\nabla_i f(X)\|_{(i)\star}^2}{\left(L_i^0 + L_i^1 \|\nabla_i f(X)\|_{(i)\star}\right)^2} \|H_i\|_{(i)}^2 \right) \\
&\overset{(8)}{=} f(X) + \sum_{i=1}^{p} \left( -\frac{\|\nabla_i f(X)\|_{(i)\star}^2}{L_i^0 + L_i^1 \|\nabla_i f(X)\|_{(i)\star}} + \frac{\|\nabla_i f(X)\|_{(i)\star}^2}{2\left(L_i^0 + L_i^1 \|\nabla_i f(X)\|_{(i)\star}\right)} \right) \\
&= f(X) - \sum_{i=1}^{p} \frac{\|\nabla_i f(X)\|_{(i)\star}^2}{2\left(L_i^0 + L_i^1 \|\nabla_i f(X)\|_{(i)\star}\right)},
\end{aligned}
$$

and hence

$$\sum_{i=1}^{p} \frac{\|\nabla_i f(X)\|_{(i)\star}^2}{2\left(L_i^0 + L_i^1 \|\nabla_i f(X)\|_{(i)\star}\right)} \leq f(X) - f(Y) \leq f(X) - f^\star$$

as needed. □

**Lemma 10.** *Let Assumptions 1 and 8 hold. Then, for any $x_i > 0$, $i \in [p]$, we have*

$$\sum_{i=1}^{p} x_i \|\nabla_i f(X)\|_{(i)\star} \leq 4 \max_{i \in [p]} (x_i L_i^1) \left( f(X) - f^\star \right) + \frac{\sum_{i=1}^{p} x_i^2 L_i^0}{\max_{i \in [p]} (x_i L_i^1)}$$

*for all $X \in \mathcal{S}$.*

*Proof.* We follow an approach similar to that in Khirirat et al. (2024, Lemma 2). Applying Lemma 9 and Lemma 12 with $y_i = \|\nabla_i f(X)\|_{(i)\star}$, $z_i = L_i^0 + L_i^1 \|\nabla_i f(X)\|_{(i)\star}$ and any positive $x_i$, we have

$$
\begin{aligned}
2\left( f(X) - f^\star \right) &\geq \sum_{i=1}^{p} \frac{\|\nabla_i f(X)\|_{(i)\star}^2}{L_i^0 + L_i^1 \|\nabla_i f(X)\|_{(i)\star}} \\
&\geq \frac{\left( \sum_{i=1}^{p} x_i \|\nabla_i f(X)\|_{(i)\star} \right)^2}{\sum_{i=1}^{p} x_i^2 L_i^0 + \sum_{i=1}^{p} x_i^2 L_i^1 \|\nabla_i f(X)\|_{(i)\star}} \\
&\geq \frac{\left( \sum_{i=1}^{p} x_i \|\nabla_i f(X)\|_{(i)\star} \right)^2}{\sum_{i=1}^{p} x_i^2 L_i^0 + \max_{i \in [p]} (x_i L_i^1) \sum_{i=1}^{p} x_i \|\nabla_i f(X)\|_{(i)\star}}
\end{aligned}
$$

$$\geq \begin{cases} \frac{\left(\sum_{i=1}^p x_i \|\nabla_i f(X)\|_{(i)\star}\right)^2}{2\sum_{i=1}^p x_i^2 L_i^0} & \text{if } \frac{\sum_{i=1}^p x_i^2 L_i^0}{\max_{i\in[p]}(x_i L_i^1)} \geq \sum_{i=1}^p x_i \|\nabla_i f(X)\|_{(i)\star}, \\ \frac{\sum_{i=1}^p x_i \|\nabla_i f(X)\|_{(i)\star}}{2\max_{i\in[p]}(x_i L_i^1)} & \text{otherwise.} \end{cases}$$

Therefore,

$$\sum_{i=1}^p x_i \|\nabla_i f(X)\|_{(i)\star} \leq \max\left\{ 4\max_{i\in[p]}(x_i L_i^1)\left(f(X)-f^\star\right), \frac{\sum_{i=1}^p x_i^2 L_i^0}{\max_{i\in[p]}(x_i L_i^1)} \right\}$$

$$\leq 4\max_{i\in[p]}(x_i L_i^1)\left(f(X)-f^\star\right) + \frac{\sum_{i=1}^p x_i^2 L_i^0}{\max_{i\in[p]}(x_i L_i^1)}.$$

$\square$

**Lemma 11.** *Let Assumptions 1, 2 and 9 hold. Then, for any $x_i > 0$, $i \in [p]$, we have*

$$\sum_{i=1}^p x_i \|\nabla_i f_j(X)\|_{(i)\star} \leq 4\max_{i\in[p]}(x_i L_{i,j}^1)\left(f_j(X)-f^\star\right) + 4\max_{i\in[p]}(x_i L_{i,j}^1)\left(f^\star - f_j^\star\right)$$

$$+ \frac{\sum_{i=1}^p x_i^2 L_{i,j}^0}{\max_{i\in[p]}(x_i L_{i,j}^1)}$$

*for all $X \in \mathcal{S}$.*

*Proof.* The proof is similar to that of Lemma 10. Applying Lemma 9 and Lemma 12 with $y_i = \|\nabla_i f_j(X)\|_{(i)\star}$, $z_i = L_{i,j}^0 + L_{i,j}^1 \|\nabla_i f_j(X)\|_{(i)\star}$ and any positive $x_i$, we have

$$2\left(f_j(X) - f_j^\star\right) \geq \sum_{i=1}^p \frac{\|\nabla_i f_j(X)\|_{(i)\star}^2}{L_{i,j}^0 + L_{i,j}^1 \|\nabla_i f_j(X)\|_{(i)\star}}$$

$$\geq \frac{\left(\sum_{i=1}^p x_i \|\nabla_i f_j(X)\|_{(i)\star}\right)^2}{\sum_{i=1}^p x_i^2 L_{i,j}^0 + \sum_{i=1}^p x_i^2 L_{i,j}^1 \|\nabla_i f_j(X)\|_{(i)\star}}$$

$$\geq \frac{\left(\sum_{i=1}^p x_i \|\nabla_i f_j(X)\|_{(i)\star}\right)^2}{\sum_{i=1}^p x_i^2 L_{i,j}^0 + \max_{i\in[p]}(x_i L_{i,j}^1)\sum_{i=1}^p x_i \|\nabla_i f_j(X)\|_{(i)\star}}$$

$$\geq \begin{cases} \frac{\left(\sum_{i=1}^p x_i \|\nabla_i f_j(X)\|_{(i)\star}\right)^2}{2\sum_{i=1}^p x_i^2 L_{i,j}^0} & \text{if } \frac{\sum_{i=1}^p x_i^2 L_{i,j}^0}{\max_{i\in[p]}(x_i L_{i,j}^1)} \geq \sum_{i=1}^p x_i \|\nabla_i f_j(X)\|_{(i)\star}, \\ \frac{\sum_{i=1}^p x_i \|\nabla_i f_j(X)\|_{(i)\star}}{2\max_{i\in[p]}(x_i L_{i,j}^1)} & \text{otherwise.} \end{cases}$$

Therefore,

$$\sum_{i=1}^p x_i \|\nabla_i f_j(X)\|_{(i)\star} \leq \max\left\{ 4\max_{i\in[p]}(x_i L_{i,j}^1)\left(f_j(X)-f_j^\star\right), \frac{\sum_{i=1}^p x_i^2 L_{i,j}^0}{\max_{i\in[p]}(x_i L_{i,j}^1)} \right\}$$

$$\leq 4\max_{i\in[p]}(x_i L_{i,j}^1)\left(f_j(X)-f_j^\star\right) + \frac{\sum_{i=1}^p x_i^2 L_{i,j}^0}{\max_{i\in[p]}(x_i L_{i,j}^1)}$$

$$= 4\max_{i\in[p]}(x_i L_{i,j}^1)\left(f_j(X)-f^\star\right) + 4\max_{i\in[p]}(x_i L_{i,j}^1)\left(f^\star - f_j^\star\right)$$

$$+ \frac{\sum_{i=1}^p x_i^2 L_{i,j}^0}{\max_{i\in[p]}(x_i L_{i,j}^1)}.$$

$\square$

### E.3 DETERMINISTIC SETTING

#### E.3.1 LAYER-WISE SMOOTH REGIME

**Theorem 14.** *Let Assumptions 1, 6 and 7 hold. Let $\{X^k\}_{k=0}^{K-1}$, $K \geq 1$, be the iterates of Algorithm 2 run with $\mathcal{C}_i^k \in \mathbb{B}(\alpha_P)$, $\mathcal{C}_{i,j}^k \in \mathbb{B}_\star(\alpha_D)$, and*

$$0 \leq \gamma_i^k \equiv \gamma_i \leq \frac{1}{2L_i^0 + \frac{4}{\alpha_D}\sqrt{12 + \frac{66}{\alpha_P^2}}\tilde{L}_i^0}, \qquad i = 1, \ldots, p.$$

*Then*

$$\frac{1}{K}\sum_{k=0}^{K-1}\sum_{i=1}^{p}\frac{\gamma_i}{\frac{1}{p}\sum_{l=1}^{p}\gamma_l}\mathbb{E}\left[\|\nabla_i f(X^k)\|_{(i)\star}^2\right] \leq \frac{1}{K}\frac{4\Psi^0}{\frac{1}{p}\sum_{l=1}^{p}\gamma_l},$$

*where*

$$\begin{aligned}
\Psi^k &:= & f(X^k) - f^\star + \sum_{i=1}^{p}\frac{6\gamma_i}{\alpha_D}\frac{1}{n}\sum_{j=1}^{n}\left\|\nabla_i f_j(X^k) - G_{i,j}^k\right\|_{(i)\star}^2 \\
& & + \sum_{i=1}^{p}\frac{66\gamma_i}{\alpha_D^2}\left(\frac{2}{\alpha_P} - 1\right)(\tilde{L}_i^0)^2\left\|X_i^k - W_i^k\right\|_{(i)}^2.
\end{aligned}$$

**Remark 15.** *Theorem 3 follows as a corollary of the more general result above by setting $p = 1$ and initializing with $X^0 = W^0$ and $G_j^0 = \nabla f_j(X^0)$ for all $j \in [n]$.*

**Remark 16.** *In the Euclidean case and when $p = 1$, our convergence guarantees recover several existing results. When primal compression is disabled (i.e., $\alpha_P = 1$), they match the rate of Richtárik et al. (2021, Theorem 1), up to constant factors. With primal compression, the rate coincides with that of* EF21-BC *in Fatkhullin et al. (2021, Theorem 21). Additionally, our results match those of* Byz-EF21-BC *(a bidirectionally compressed method with error feedback for Byzantine-robust learning) from Rammal et al. (2024, Theorem 3.1), in the absence of Byzantine workers.*

*Proof of Theorem 14.* Let $A_i, B_i > 0$ be some constants to be determined later, and define

$$\Psi^k := f(X^k) - f^\star + \sum_{i=1}^{p}A_i\frac{1}{n}\sum_{j=1}^{n}\left\|\nabla_i f_j(X^k) - G_{i,j}^k\right\|_{(i)\star}^2 + \sum_{i=1}^{p}B_i\left\|X_i^k - W_i^k\right\|_{(i)}^2.$$

**Step I: Bounding** $\mathbb{E}\left[\left\|\nabla_i f_j(X^{k+1}) - G_{i,j}^{k+1}\right\|_{(i)\star}^2\right]$. The algorithm's update rule gives

$$\begin{aligned}
& \mathbb{E}\left[\left\|\nabla_i f_j(X^{k+1}) - G_{i,j}^{k+1}\right\|_{(i)\star}^2 \,\Big|\, X^{k+1}, W^{k+1}, G_{i,j}^k\right] \\
=\ & \mathbb{E}\left[\left\|\nabla_i f_j(X^{k+1}) - G_{i,j}^k - \mathcal{C}_{i,j}^k(\nabla_i f_j(W^{k+1}) - G_{i,j}^k)\right\|_{(i)\star}^2 \,\Big|\, X^{k+1}, W^{k+1}, G_{i,j}^k\right] \\
\overset{(28)}{\leq}\ & \left(1 + \frac{\alpha_D}{2}\right)\mathbb{E}\left[\left\|\nabla_i f_j(W^{k+1}) - G_{i,j}^k - \mathcal{C}_{i,j}^k(\nabla_i f_j(W^{k+1}) - G_{i,j}^k)\right\|_{(i)\star}^2 \,\Big|\, X^{k+1}, W^{k+1}, G_{i,j}^k\right] \\
& + \left(1 + \frac{2}{\alpha_D}\right)\mathbb{E}\left[\left\|\nabla_i f_j(X^{k+1}) - \nabla_i f_j(W^{k+1})\right\|_{(i)\star}^2 \,\Big|\, X^{k+1}, W^{k+1}, G_{i,j}^k\right] \\
\leq\ & \left(1 + \frac{\alpha_D}{2}\right)(1 - \alpha_D)\mathbb{E}\left[\left\|\nabla_i f_j(W^{k+1}) - G_{i,j}^k\right\|_{(i)\star}^2 \,\Big|\, X^{k+1}, W^{k+1}, G_{i,j}^k\right] \\
& + \left(1 + \frac{2}{\alpha_D}\right)\mathbb{E}\left[\left\|\nabla_i f_j(X^{k+1}) - \nabla_i f_j(W^{k+1})\right\|_{(i)\star}^2 \,\Big|\, X^{k+1}, W^{k+1}, G_{i,j}^k\right] \\
\overset{(30)}{\leq}\ & \left(1 - \frac{\alpha_D}{2}\right)\left\|\nabla_i f_j(W^{k+1}) - G_{i,j}^k\right\|_{(i)\star}^2 + \left(1 + \frac{2}{\alpha_D}\right)\left\|\nabla_i f_j(X^{k+1}) - \nabla_i f_j(W^{k+1})\right\|_{(i)\star}^2 \\
\overset{(28)}{\leq}\ & \left(1 - \frac{\alpha_D}{2}\right)\left(1 + \frac{\alpha_D}{4}\right)\left\|\nabla_i f_j(X^k) - G_{i,j}^k\right\|_{(i)\star}^2 \\
& + \left(1 - \frac{\alpha_D}{2}\right)\left(1 + \frac{4}{\alpha_D}\right)\left\|\nabla_i f_j(W^{k+1}) - \nabla_i f_j(X^k)\right\|_{(i)\star}^2
\end{aligned}$$

$$+ \left( 1 + \frac{2}{\alpha_D} \right) \left\| \nabla_i f_j(X^{k+1}) - \nabla_i f_j(W^{k+1}) \right\|_{(i)\star}^2$$

$$\overset{(30),(31)}{\leq} \left( 1 - \frac{\alpha_D}{4} \right) \left\| \nabla_i f_j(X^k) - G_{i,j}^k \right\|_{(i)\star}^2 + \frac{4}{\alpha_D} \left\| \nabla_i f_j(W^{k+1}) - \nabla_i f_j(X^k) \right\|_{(i)\star}^2$$

$$+ \left( 1 + \frac{2}{\alpha_D} \right) \left\| \nabla_i f_j(X^{k+1}) - \nabla_i f_j(W^{k+1}) \right\|_{(i)\star}^2.$$

Therefore, using smoothness,

$$\mathbb{E} \left[ \left\| \nabla_i f_j(X^{k+1}) - G_{i,j}^{k+1} \right\|_{(i)\star}^2 \Big| X^{k+1}, W^{k+1}, G_{i,j}^k \right]$$

$$\overset{(7)}{\leq} \left( 1 - \frac{\alpha_D}{4} \right) \left\| \nabla_i f_j(X^k) - G_{i,j}^k \right\|_{(i)\star}^2 + \frac{4}{\alpha_D} (L_{i,j}^0)^2 \left\| W_i^{k+1} - X_i^k \right\|_{(i)}^2$$

$$+ \left( 1 + \frac{2}{\alpha_D} \right) (L_{i,j}^0)^2 \left\| X_i^{k+1} - W_i^{k+1} \right\|_{(i)}^2$$

$$\overset{(28)}{\leq} \left( 1 - \frac{\alpha_D}{4} \right) \left\| \nabla_i f_j(X^k) - G_{i,j}^k \right\|_{(i)\star}^2 + \frac{8}{\alpha_D} (L_{i,j}^0)^2 \left\| X_i^{k+1} - X_i^k \right\|_{(i)}^2$$

$$+ \frac{8}{\alpha_D} (L_{i,j}^0)^2 \left\| X_i^{k+1} - W_i^{k+1} \right\|_{(i)}^2 + \left( 1 + \frac{2}{\alpha_D} \right) (L_{i,j}^0)^2 \left\| X_i^{k+1} - W_i^{k+1} \right\|_{(i)}^2$$

$$\leq \left( 1 - \frac{\alpha_D}{4} \right) \left\| \nabla_i f_j(X^k) - G_{i,j}^k \right\|_{(i)\star}^2 + \frac{8}{\alpha_D} (L_{i,j}^0)^2 \gamma_i^2 \left\| G_i^k \right\|_{(i)\star}^2$$

$$+ \frac{11}{\alpha_D} (L_{i,j}^0)^2 \left\| X_i^{k+1} - W_i^{k+1} \right\|_{(i)}^2.$$

Taking expectation, we obtain the recursion

$$\mathbb{E} \left[ \left\| \nabla_i f_j(X^{k+1}) - G_{i,j}^{k+1} \right\|_{(i)\star}^2 \right]$$

$$\leq \left( 1 - \frac{\alpha_D}{4} \right) \mathbb{E} \left[ \left\| \nabla_i f_j(X^k) - G_{i,j}^k \right\|_{(i)\star}^2 \right] + \frac{8}{\alpha_D} (L_{i,j}^0)^2 \gamma_i^2 \mathbb{E} \left[ \left\| G_i^k \right\|_{(i)\star}^2 \right]$$

$$+ \frac{11}{\alpha_D} (L_{i,j}^0)^2 \mathbb{E} \left[ \left\| X_i^{k+1} - W_i^{k+1} \right\|_{(i)}^2 \right]. \tag{13}$$

**Step II: Bounding** $\mathbb{E} \left[ \left\| X_i^{k+1} - W_i^{k+1} \right\|_{(i)}^2 \right]$. By Lemma 3

$$\mathbb{E} \left[ \left\| X_i^{k+1} - W_i^{k+1} \right\|_{(i)}^2 \right] \leq \left( 1 - \frac{\alpha_P}{2} \right) \mathbb{E} \left[ \left\| X_i^k - W_i^k \right\|_{(i)}^2 \right] + \frac{2}{\alpha_P} \gamma_i^2 \mathbb{E} \left[ \left\| G_i^k \right\|_{(i)\star}^2 \right]. \tag{14}$$

**Step III: Bounding** $\Psi^{k+1}$. By Lemma 1 and Jensen's inequality

$$\Psi^{k+1}$$

$$= f(X^{k+1}) - f^\star + \sum_{i=1}^p A_i \frac{1}{n} \sum_{j=1}^n \left\| \nabla_i f_j(X^{k+1}) - G_{i,j}^{k+1} \right\|_{(i)\star}^2 + \sum_{i=1}^p B_i \left\| X_i^{k+1} - W_i^{k+1} \right\|_{(i)}^2$$

$$\leq f(X^k) - f^\star + \sum_{i=1}^p \frac{3\gamma_i}{2} \left\| \nabla_i f(X^k) - G_i^k \right\|_{(i)\star}^2 - \sum_{i=1}^p \frac{\gamma_i}{4} \left\| \nabla_i f(X^k) \right\|_{(i)\star}^2$$

$$- \sum_{i=1}^p \left( \frac{1}{4\gamma_i} - \frac{L_i^0}{2} \right) \gamma_i^2 \left\| G_i^k \right\|_{(i)\star}^2$$

$$+ \sum_{i=1}^p A_i \frac{1}{n} \sum_{j=1}^n \left\| \nabla_i f_j(X^{k+1}) - G_{i,j}^{k+1} \right\|_{(i)\star}^2 + \sum_{i=1}^p B_i \left\| X_i^{k+1} - W_i^{k+1} \right\|_{(i)}^2$$

$$\leq f(X^k) - f^\star + \sum_{i=1}^p \frac{3\gamma_i}{2} \frac{1}{n} \sum_{j=1}^n \left\| \nabla_i f_j(X^k) - G_{i,j}^k \right\|_{(i)\star}^2 - \sum_{i=1}^p \frac{\gamma_i}{4} \left\| \nabla_i f(X^k) \right\|_{(i)\star}^2$$

$$- \sum_{i=1}^{p} \left( \frac{1}{4\gamma_i} - \frac{L_i^0}{2} \right) \gamma_i^2 \left\| G_i^k \right\|_{(i)\star}^2$$

$$+ \sum_{i=1}^{p} A_i \frac{1}{n} \sum_{j=1}^{n} \left\| \nabla_i f_j(X^{k+1}) - G_{i,j}^{k+1} \right\|_{(i)\star}^2 + \sum_{i=1}^{p} B_i \left\| X_i^{k+1} - W_i^{k+1} \right\|_{(i)}^2 .$$

Taking expectation and using (13) gives

$$\mathbb{E} \left[ \Psi^{k+1} \right]$$

$$\leq \mathbb{E} \left[ f(X^k) - f^\star \right] + \sum_{i=1}^{p} \frac{3\gamma_i}{2} \frac{1}{n} \sum_{j=1}^{n} \mathbb{E} \left[ \left\| \nabla_i f_j(X^k) - G_{i,j}^k \right\|_{(i)\star}^2 \right] - \sum_{i=1}^{p} \frac{\gamma_i}{4} \mathbb{E} \left[ \left\| \nabla_i f(X^k) \right\|_{(i)\star}^2 \right]$$

$$- \sum_{i=1}^{p} \left( \frac{1}{4\gamma_i} - \frac{L_i^0}{2} \right) \gamma_i^2 \mathbb{E} \left[ \left\| G_i^k \right\|_{(i)\star}^2 \right] + \sum_{i=1}^{p} A_i \frac{1}{n} \sum_{j=1}^{n} \left( 1 - \frac{\alpha_D}{4} \right) \mathbb{E} \left[ \left\| \nabla_i f_j(X^k) - G_{i,j}^k \right\|_{(i)\star}^2 \right]$$

$$+ \sum_{i=1}^{p} A_i \frac{1}{n} \sum_{j=1}^{n} \frac{8}{\alpha_D} (L_{i,j}^0)^2 \gamma_i^2 \mathbb{E} \left[ \left\| G_i^k \right\|_{(i)\star}^2 \right] + \sum_{i=1}^{p} A_i \frac{1}{n} \sum_{j=1}^{n} \frac{11}{\alpha_D} (L_{i,j}^0)^2 \mathbb{E} \left[ \left\| X_i^{k+1} - W_i^{k+1} \right\|_{(i)}^2 \right]$$

$$+ \sum_{i=1}^{p} B_i \mathbb{E} \left[ \left\| X_i^{k+1} - W_i^{k+1} \right\|_{(i)}^2 \right]$$

$$= \mathbb{E} \left[ f(X^k) - f^\star \right] + \sum_{i=1}^{p} \left( \frac{3\gamma_i}{2} + A_i \left( 1 - \frac{\alpha_D}{4} \right) \right) \frac{1}{n} \sum_{j=1}^{n} \mathbb{E} \left[ \left\| \nabla_i f_j(X^k) - G_{i,j}^k \right\|_{(i)\star}^2 \right]$$

$$- \sum_{i=1}^{p} \frac{\gamma_i}{4} \mathbb{E} \left[ \left\| \nabla_i f(X^k) \right\|_{(i)\star}^2 \right] - \sum_{i=1}^{p} \left( \frac{1}{4\gamma_i} - \frac{L_i^0}{2} - A_i \frac{8}{\alpha_D} (\tilde{L}_i^0)^2 \right) \gamma_i^2 \mathbb{E} \left[ \left\| G_i^k \right\|_{(i)\star}^2 \right]$$

$$+ \sum_{i=1}^{p} \left( A_i \frac{11}{\alpha_D} (\tilde{L}_i^0)^2 + B_i \right) \mathbb{E} \left[ \left\| X_i^{k+1} - W_i^{k+1} \right\|_{(i)}^2 \right] .$$

Next, applying (14), we get

$$\mathbb{E} \left[ \Psi^{k+1} \right] \leq \mathbb{E} \left[ f(X^k) - f^\star \right] + \sum_{i=1}^{p} \left( \frac{3\gamma_i}{2} + A_i \left( 1 - \frac{\alpha_D}{4} \right) \right) \frac{1}{n} \sum_{j=1}^{n} \mathbb{E} \left[ \left\| \nabla_i f_j(X^k) - G_{i,j}^k \right\|_{(i)\star}^2 \right]$$

$$- \sum_{i=1}^{p} \frac{\gamma_i}{4} \mathbb{E} \left[ \left\| \nabla_i f(X^k) \right\|_{(i)\star}^2 \right] - \sum_{i=1}^{p} \left( \frac{1}{4\gamma_i} - \frac{L_i^0}{2} - A_i \frac{8}{\alpha_D} (\tilde{L}_i^0)^2 \right) \gamma_i^2 \mathbb{E} \left[ \left\| G_i^k \right\|_{(i)\star}^2 \right]$$

$$+ \sum_{i=1}^{p} \left( A_i \frac{11}{\alpha_D} (\tilde{L}_i^0)^2 + B_i \right) \frac{2}{\alpha_P} \gamma_i^2 \mathbb{E} \left[ \left\| G_i^k \right\|_{(i)\star}^2 \right]$$

$$+ \sum_{i=1}^{p} \left( A_i \frac{11}{\alpha_D} (\tilde{L}_i^0)^2 + B_i \right) \left( 1 - \frac{\alpha_P}{2} \right) \mathbb{E} \left[ \left\| X_i^k - W_i^k \right\|_{(i)}^2 \right] .$$

Taking $A_i = \frac{6\gamma_i}{\alpha_D}$ and $B_i = A_i \frac{11}{\alpha_D} \left( \frac{2}{\alpha_P} - 1 \right) (\tilde{L}_i^0)^2 = \frac{66\gamma_i}{\alpha_D^2} \left( \frac{2}{\alpha_P} - 1 \right) (\tilde{L}_i^0)^2$ yields

$$\frac{3\gamma_i}{2} + A_i \left( 1 - \frac{\alpha_D}{4} \right) = A_i,$$

$$\left( A_i \frac{11}{\alpha_D} (\tilde{L}_i^0)^2 + B_i \right) \left( 1 - \frac{\alpha_P}{2} \right) = B_i,$$

and consequently,

$$\mathbb{E} \left[ \Psi^{k+1} \right] \leq \mathbb{E} \left[ f(X^k) - f^\star \right] + \sum_{i=1}^{p} A_i \frac{1}{n} \sum_{j=1}^{n} \mathbb{E} \left[ \left\| \nabla_i f_j(X^k) - G_{i,j}^k \right\|_{(i)\star}^2 \right]$$

$$- \sum_{i=1}^{p} \frac{\gamma_i}{4} \mathbb{E} \left[ \left\| \nabla_i f(X^k) \right\|_{(i)\star}^2 \right] - \sum_{i=1}^{p} \left( \frac{1}{4\gamma_i} - \frac{L_i^0}{2} - \frac{8A_i}{\alpha_D} (\tilde{L}_i^0)^2 \right) \gamma_i^2 \mathbb{E} \left[ \left\| G_i^k \right\|_{(i)\star}^2 \right]$$

$$+ \sum_{i=1}^{p} \frac{B_i}{1 - \frac{\alpha_P}{2}} \frac{2}{\alpha_P} \gamma_i^2 \mathbb{E}\left[\|G_i^k\|_{(i)\star}^2\right] + \sum_{i=1}^{p} B_i \mathbb{E}\left[\|X_i^k - W_i^k\|_{(i)}^2\right]$$

$$= \mathbb{E}\left[\Psi^k\right] - \sum_{i=1}^{p} \frac{\gamma_i}{4} \mathbb{E}\left[\|\nabla_i f(X^k)\|_{(i)\star}^2\right]$$

$$- \sum_{i=1}^{p} \left(\frac{1}{4\gamma_i} - \frac{L_i^0}{2} - \frac{8A_i}{\alpha_D}(\tilde{L}_i^0)^2 - \frac{4B_i}{\alpha_P(2-\alpha_P)}\right) \gamma_i^2 \mathbb{E}\left[\|G_i^k\|_{(i)\star}^2\right].$$

Now, note that

$$\frac{1}{4\gamma_i} - \frac{L_i^0}{2} - \frac{8A_i}{\alpha_D}(\tilde{L}_i^0)^2 - \frac{4B_i}{\alpha_P(2-\alpha_P)} = \frac{1}{4\gamma_i} - \frac{L_i^0}{2} - \underbrace{\left(\frac{48}{\alpha_D^2}(\tilde{L}_i^0)^2 + \frac{264}{\alpha_P^2\alpha_D^2}(\tilde{L}_i^0)^2\right)}_{:=\zeta_i} \gamma_i \geq 0$$

for $\gamma_i \leq \frac{1}{2L_i^0 + 2\sqrt{\zeta_i}}$. For such a choice of the stepsizes, we have

$$\mathbb{E}\left[\Psi^{k+1}\right] \leq \mathbb{E}\left[\Psi^k\right] - \sum_{i=1}^{p} \frac{\gamma_i}{4} \mathbb{E}\left[\|\nabla_i f(X^k)\|_{(i)\star}^2\right],$$

and hence

$$\sum_{k=0}^{K-1} \sum_{i=1}^{p} \gamma_i \mathbb{E}\left[\|\nabla_i f(X^k)\|_{(i)\star}^2\right] \leq 4 \sum_{k=0}^{K-1} \left(\mathbb{E}\left[\Psi^k\right] - \mathbb{E}\left[\Psi^{k+1}\right]\right) \leq 4\Psi^0.$$

Lastly, dividing by $\frac{K}{p} \sum_{l=1}^{p} \gamma_l$, we obtain

$$\frac{1}{K} \sum_{k=0}^{K-1} \sum_{i=1}^{p} \frac{\gamma_i}{\frac{1}{p} \sum_{l=1}^{p} \gamma_l} \mathbb{E}\left[\|\nabla_i f(X^k)\|_{(i)\star}^2\right] \leq \frac{1}{K} \frac{4\Psi^0}{\frac{1}{p} \sum_{l=1}^{p} \gamma_l}.$$

$\square$

### E.3.2 LAYER-WISE $(L^0, L^1)$–SMOOTH REGIME

We now consider a deterministic variant of EF21-Muon (Algorithm 2) without primal compression, which iterates

$$X_i^{k+1} = \text{LMO}_{\mathcal{B}(X_i^k, t_i^k)}\left(G_i^k\right),$$
$$G_{i,j}^{k+1} = G_{i,j}^k + \mathcal{C}_{i,j}^k(\nabla_i f_j(X^{k+1}) - G_{i,j}^k),$$
$$G_i^{k+1} = \frac{1}{n} \sum_{j=1}^{n} G_{i,j}^{k+1} = G_i^k + \frac{1}{n} \sum_{j=1}^{n} \mathcal{C}_{i,j}^k(\nabla_i f_j(X^{k+1}) - G_{i,j}^k).$$

This corresponds to using identity compressors on the server side.

**Theorem 17.** *Let Assumptions 1, 2, 8 and 9 hold and let $\{X^k\}_{k=0}^{K-1}$, $K \geq 1$, be the iterates of Algorithm 2 run with $\mathcal{C}_i^k \equiv \mathcal{I}$ (the identity compressor), $\mathcal{C}_{i,j}^k \in \mathbb{B}_\star(\alpha_D)$, and*

$$t_i^k \equiv t_i = \frac{\eta_i}{\sqrt{K+1}}, \qquad i = 1, \ldots, p,$$

*for some $\eta_i > 0$. Then,*

$$\min_{k=0,\ldots,K} \sum_{i=1}^{p} \frac{\eta_i}{\frac{1}{p} \sum_{l=1}^{p} \eta_i} \mathbb{E}\left[\|\nabla_i f(X^k)\|_{(i)\star}\right]$$

$$\leq \frac{\exp\left(4 \max_{i \in [p], j \in [n]}(\eta_i^2 C_i L_{i,j}^1)\right)}{\sqrt{K+1}\left(\frac{1}{p} \sum_{l=1}^{p} \eta_i\right)} \Psi^0$$

$$+\frac{1}{\sqrt{K+1}\left(\frac{1}{p}\sum_{l=1}^{p}\eta_i\right)}\left(\frac{1}{n}\sum_{j=1}^{n}4\max_{i\in[p]}(\eta_i^2 C_i L_{i,j}^1)\left(f^\star - f_j^\star\right) + \frac{1}{n}\sum_{j=1}^{n}\sum_{i=1}^{p}\frac{\eta_i^2 C_i L_{i,j}^0}{L_{i,j}^1} + \sum_{i=1}^{p}\eta_i^2 D_i\right).$$

where $C_i := \frac{L_i^1}{2} + \frac{2\sqrt{1-\alpha_D}L_{i,\max}^1}{1-\sqrt{1-\alpha_D}}$, $D_i := \frac{L_i^0}{2} + \frac{2\sqrt{1-\alpha_D}\bar{L}_i^0}{1-\sqrt{1-\alpha_D}}$ and

$$\Psi^k := f(X^k) - f^\star + \sum_{i=1}^{p}\frac{2t_i}{1-\sqrt{1-\alpha_D}}\frac{1}{n}\sum_{j=1}^{n}\left\|\nabla_i f_j(X^k) - G_{i,j}^k\right\|_{(i)\star}.$$

**Remark 18.** *Theorem 4 follows as a corollary of the result in Theorem 17 by setting $p = 1$ and initializing with $G_j^0 = \nabla f_j(X^0)$ for all $j \in [n]$.*

*Proof.* Let $A_i > 0$ be some constants to be determined later, and define

$$\Psi^k := f(X^k) - f^\star + \sum_{i=1}^{p}A_i\frac{1}{n}\sum_{j=1}^{n}\left\|\nabla_i f_j(X^k) - G_{i,j}^k\right\|_{(i)\star}.$$

By Lemma 2 and Jensen's inequality

$$
\begin{aligned}
\Psi^{k+1} &= f(X^{k+1}) - f^\star + \sum_{i=1}^{p}A_i\frac{1}{n}\sum_{j=1}^{n}\left\|\nabla_i f_j(X^{k+1}) - G_{i,j}^{k+1}\right\|_{(i)\star} \\
&\leq f(X^k) - f^\star + \sum_{i=1}^{p}2t_i\left\|\nabla_i f(X^k) - G_i^k\right\|_{(i)\star} - \sum_{i=1}^{p}t_i\left\|\nabla_i f(X^k)\right\|_{(i)\star} \\
&\quad + \sum_{i=1}^{p}\frac{L_i^0 + L_i^1\left\|\nabla_i f(X^k)\right\|_{(i)\star}}{2}t_i^2 + \sum_{i=1}^{p}A_i\frac{1}{n}\sum_{j=1}^{n}\left\|\nabla_i f_j(X^{k+1}) - G_{i,j}^{k+1}\right\|_{(i)\star} \\
&\leq f(X^k) - f^\star + \sum_{i=1}^{p}2t_i\frac{1}{n}\sum_{j=1}^{n}\left\|\nabla_i f_j(X^k) - G_{i,j}^k\right\|_{(i)\star} - \sum_{i=1}^{p}t_i\left\|\nabla_i f(X^k)\right\|_{(i)\star} \\
&\quad + \sum_{i=1}^{p}\frac{L_i^0 + L_i^1\left\|\nabla_i f(X^k)\right\|_{(i)\star}}{2}t_i^2 + \sum_{i=1}^{p}A_i\frac{1}{n}\sum_{j=1}^{n}\left\|\nabla_i f_j(X^{k+1}) - G_{i,j}^{k+1}\right\|_{(i)\star}.
\end{aligned}
$$

Taking expectation conditioned on $[X^{k+1}, X^k, G^k]$ and using Lemma 6 gives

$$
\begin{aligned}
&\mathbb{E}\left[\Psi^{k+1}\,\middle|\,X^{k+1}, X^k, G^k\right] \\
&\leq f(X^k) - f^\star + \sum_{i=1}^{p}2t_i\frac{1}{n}\sum_{j=1}^{n}\left\|\nabla_i f_j(X^k) - G_{i,j}^k\right\|_{(i)\star} - \sum_{i=1}^{p}t_i\left\|\nabla_i f(X^k)\right\|_{(i)\star} \\
&\quad + \sum_{i=1}^{p}\frac{L_i^0 + L_i^1\left\|\nabla_i f(X^k)\right\|_{(i)\star}}{2}t_i^2 \\
&\quad + \sum_{i=1}^{p}A_i\frac{1}{n}\sum_{j=1}^{n}\mathbb{E}\left[\left\|\nabla_i f_j(X^{k+1}) - G_{i,j}^{k+1}\right\|_{(i)\star}\,\middle|\,X^{k+1}, X^k, G^k\right] \\
&\overset{(6)}{\leq} f(X^k) - f^\star + \sum_{i=1}^{p}2t_i\frac{1}{n}\sum_{j=1}^{n}\left\|\nabla_i f_j(X^k) - G_{i,j}^k\right\|_{(i)\star} - \sum_{i=1}^{p}t_i\left\|\nabla_i f(X^k)\right\|_{(i)\star} \\
&\quad + \sum_{i=1}^{p}\frac{L_i^0 + L_i^1\left\|\nabla_i f(X^k)\right\|_{(i)\star}}{2}t_i^2 \\
&\quad + \sum_{i=1}^{p}A_i\sqrt{1-\alpha_D}\frac{1}{n}\sum_{j=1}^{n}\left(\left\|\nabla_i f_j(X^k) - G_{i,j}^k\right\|_{(i)\star} + \left(L_{i,j}^0 + L_{i,j}^1\left\|\nabla_i f_j(X^k)\right\|_{(i)\star}\right)t_i\right) \\
&= f(X^k) - f^\star + \sum_{i=1}^{p}\left(2t_i + A_i\sqrt{1-\alpha_D}\right)\frac{1}{n}\sum_{j=1}^{n}\left\|\nabla_i f_j(X^k) - G_{i,j}^k\right\|_{(i)\star} - \sum_{i=1}^{p}t_i\left\|\nabla_i f(X^k)\right\|_{(i)\star}
\end{aligned}
$$

$$+ \sum_{i=1}^{p} \frac{L_i^1 t_i^2}{2} \left\| \nabla_i f(X^k) \right\|_{(i)\star} + \sqrt{1-\alpha_D} \sum_{i=1}^{p} A_i t_i \left( \frac{1}{n} \sum_{j=1}^{n} L_{i,j}^1 \left\| \nabla_i f_j(X^k) \right\|_{(i)\star} \right)$$

$$+ \sum_{i=1}^{p} \frac{t_i^2 L_i^0}{2} + \sqrt{1-\alpha_D} \sum_{i=1}^{p} A_i t_i \bar{L}_i^0.$$

Now, letting $A_i = \frac{2t_i}{1-\sqrt{1-\alpha_D}}$, we have

$$2t_i + A_i \sqrt{1-\alpha_D} = 2t_i + \frac{2t_i}{1-\sqrt{1-\alpha_D}} \sqrt{1-\alpha_D} = A_i,$$

and consequently,

$$\mathbb{E}\left[ \Psi^{k+1} \middle| X^{k+1}, X^k, G^k \right]$$

$$\leq \quad f(X^k) - f^\star + \sum_{i=1}^{p} A_i \frac{1}{n} \sum_{j=1}^{n} \left\| \nabla_i f_j(X^k) - G_{i,j}^k \right\|_{(i)\star} - \sum_{i=1}^{p} t_i \left\| \nabla_i f(X^k) \right\|_{(i)\star}$$

$$+ \sum_{i=1}^{p} \frac{L_i^1 t_i^2}{2} \left\| \nabla_i f(X^k) \right\|_{(i)\star} + \sqrt{1-\alpha_D} \sum_{i=1}^{p} A_i t_i \left( \frac{1}{n} \sum_{j=1}^{n} L_{i,j}^1 \left\| \nabla_i f_j(X^k) \right\|_{(i)\star} \right)$$

$$+ \sum_{i=1}^{p} \frac{t_i^2 L_i^0}{2} + \sqrt{1-\alpha_D} \sum_{i=1}^{p} A_i t_i \bar{L}_i^0$$

$$\leq \quad \Psi^k - \sum_{i=1}^{p} t_i \left\| \nabla_i f(X^k) \right\|_{(i)\star} + \sum_{i=1}^{p} \frac{L_i^1 t_i^2}{2} \left( \frac{1}{n} \sum_{j=1}^{n} \left\| \nabla_i f_j(X^k) \right\|_{(i)\star} \right)$$

$$+ \sum_{i=1}^{p} \frac{2\sqrt{1-\alpha_D} L_{i,\max}^1}{1-\sqrt{1-\alpha_D}} t_i^2 \left( \frac{1}{n} \sum_{j=1}^{n} \left\| \nabla_i f_j(X^k) \right\|_{(i)\star} \right) + \sum_{i=1}^{p} \left( \frac{t_i^2 L_i^0}{2} + \frac{2\sqrt{1-\alpha_D}}{1-\sqrt{1-\alpha_D}} t_i^2 \bar{L}_i^0 \right)$$

$$= \quad \Psi^k - \sum_{i=1}^{p} t_i \left\| \nabla_i f(X^k) \right\|_{(i)\star}$$

$$+ \sum_{i=1}^{p} \left( \underbrace{\left( \frac{L_i^1}{2} + \frac{2\sqrt{1-\alpha_D} L_{i,\max}^1}{1-\sqrt{1-\alpha_D}} \right)}_{:=C_i} \frac{1}{n} \sum_{j=1}^{n} \left\| \nabla_i f_j(X^k) \right\|_{(i)\star} + \underbrace{\frac{L_i^0}{2} + \frac{2\sqrt{1-\alpha_D}\bar{L}_i^0}{1-\sqrt{1-\alpha_D}}}_{:=D_i} \right) t_i^2.$$

Taking $t_i = \frac{\eta_i}{\sqrt{K+1}}$ for some $\eta_i > 0$ and using Lemma 11 with $x_i = \eta_i^2 C_i$, we get

$$\mathbb{E}\left[ \Psi^{k+1} \middle| X^{k+1}, X^k, G^k \right]$$

$$\leq \quad \Psi^k - \sum_{i=1}^{p} t_i \left\| \nabla_i f(X^k) \right\|_{(i)\star} + \frac{1}{K+1} \frac{1}{n} \sum_{j=1}^{n} \sum_{i=1}^{p} \eta_i^2 C_i \left\| \nabla_i f_j(X^k) \right\|_{(i)\star} + \sum_{i=1}^{p} D_i t_i^2$$

$$\overset{(11)}{\leq} \quad \Psi^k - \sum_{i=1}^{p} t_i \left\| \nabla_i f(X^k) \right\|_{(i)\star} + \sum_{i=1}^{p} D_i t_i^2 + \frac{1}{K+1} \frac{1}{n} \sum_{j=1}^{n} 4 \max_{i\in[p]} (\eta_i^2 C_i L_{i,j}^1) \left( f_j(X^k) - f^\star \right)$$

$$+ \frac{1}{K+1} \frac{1}{n} \sum_{j=1}^{n} 4 \max_{i\in[p]} (\eta_i^2 C_i L_{i,j}^1) \left( f^\star - f_j^\star \right) + \frac{1}{K+1} \frac{1}{n} \sum_{j=1}^{n} \frac{\sum_{i=1}^{p} \eta_i^4 C_i^2 L_{i,j}^0}{\max_{i\in[p]} (\eta_i^2 C_i L_{i,j}^1)}$$

$$\leq \quad \Psi^k - \frac{1}{\sqrt{K+1}} \sum_{i=1}^{p} \eta_i \left\| \nabla_i f(X^k) \right\|_{(i)\star} + \frac{4}{K+1} \max_{i\in[p],j\in[n]} (\eta_i^2 C_i L_{i,j}^1) \frac{1}{n} \sum_{j=1}^{n} \left( f_j(X^k) - f^\star \right)$$

$$+ \frac{1}{K+1} \left( \frac{1}{n} \sum_{j=1}^{n} 4 \max_{i\in[p]} (\eta_i^2 C_i L_{i,j}^1) \left( f^\star - f_j^\star \right) + \frac{1}{n} \sum_{j=1}^{n} \sum_{i=1}^{p} \frac{\eta_i^2 C_i L_{i,j}^0}{L_{i,j}^1} + \sum_{i=1}^{p} \eta_i^2 D_i \right).$$

Now, since $\frac{1}{n} \sum_{j=1}^{n} \left( f_j(X^k) - f^\star \right) = f(X^k) - f^\star \leq \Psi^k$, we obtain

$$\mathbb{E}\left[ \Psi^{k+1} \middle| X^{k+1}, X^k, G^k \right]$$

$$\leq \quad \left( 1 + \frac{4}{K+1} \max_{i \in [p], j \in [n]} (\eta_i^2 C_i L_{i,j}^1) \right) \Psi^k - \frac{1}{\sqrt{K+1}} \sum_{i=1}^{p} \eta_i \left\| \nabla_i f(X^k) \right\|_{(i)\star}$$

$$+ \frac{1}{K+1} \left( \frac{1}{n} \sum_{j=1}^{n} 4 \max_{i \in [p]} (\eta_i^2 C_i L_{i,j}^1) \left( f^\star - f_j^\star \right) + \frac{1}{n} \sum_{j=1}^{n} \sum_{i=1}^{p} \frac{\eta_i^2 C_i L_{i,j}^0}{L_{i,j}^1} + \sum_{i=1}^{p} \eta_i^2 D_i \right).$$

Taking expectation,

$$\mathbb{E}\left[ \Psi^{k+1} \right]$$

$$\leq \quad \left( 1 + \underbrace{\frac{4}{K+1} \max_{i \in [p], j \in [n]} (\eta_i^2 C_i L_{i,j}^1)}_{:=a_1} \right) \mathbb{E}\left[ \Psi^k \right] - \frac{1}{\sqrt{K+1}} \sum_{i=1}^{p} \eta_i \mathbb{E}\left[ \left\| \nabla_i f(X^k) \right\|_{(i)\star} \right]$$

$$\underbrace{+ \frac{1}{K+1} \left( \frac{1}{n} \sum_{j=1}^{n} 4 \max_{i \in [p]} (\eta_i^2 C_i L_{i,j}^1) \left( f^\star - f_j^\star \right) + \frac{1}{n} \sum_{j=1}^{n} \sum_{i=1}^{p} \frac{\eta_i^2 C_i L_{i,j}^0}{L_{i,j}^1} + \sum_{i=1}^{p} \eta_i^2 D_i \right)}_{:=a_2},$$

and hence, applying Lemma 15 with $A^k = \mathbb{E}\left[ \Psi^k \right]$ and $B_i^k = \frac{\eta_i}{\sqrt{K+1}} \mathbb{E}\left[ \left\| \nabla_i f(X^k) \right\|_{(i)\star} \right]$,

$$\min_{k=0,\ldots,K} \sum_{i=1}^{p} \frac{\eta_i}{\sqrt{K+1}} \mathbb{E}\left[ \left\| \nabla_i f(X^k) \right\|_{(i)\star} \right]$$

$$\leq \quad \frac{\exp\left( \frac{4}{K+1} \max_{i \in [p], j \in [n]} (\eta_i^2 C_i L_{i,j}^1)(K+1) \right)}{(K+1)} \Psi^0$$

$$+ \frac{1}{K+1} \left( \frac{1}{n} \sum_{j=1}^{n} 4 \max_{i \in [p]} (\eta_i^2 C_i L_{i,j}^1) \left( f^\star - f_j^\star \right) + \frac{1}{n} \sum_{j=1}^{n} \sum_{i=1}^{p} \frac{\eta_i^2 C_i L_{i,j}^0}{L_{i,j}^1} + \sum_{i=1}^{p} \eta_i^2 D_i \right).$$

Dividing by $\frac{\frac{1}{p} \sum_{l=1}^{p} \eta_i}{\sqrt{K+1}}$ finishes the proof. $\qquad\square$

### E.4 STOCHASTIC SETTING

#### E.4.1 LAYER-WISE SMOOTH REGIME

**Theorem 19.** *Let Assumptions 1, 6, 7 and 10 hold. Let $\{X^k\}_{k=0}^{K-1}$, $K \geq 1$, be the iterates of Algorithm 3 run with $\mathcal{C}_i^k \in \mathbb{B}(\alpha_P)$, $\mathcal{C}_{i,j}^k \in \mathbb{B}_2(\alpha_D)$, any $\beta_i \in (0,1]$, and*

$$0 \leq \gamma_i^k \equiv \gamma_i \leq \frac{1}{2L_i^0 + 2\sqrt{\zeta_i}}, \qquad i = 1, \ldots, p,$$

*where $\zeta_i := \frac{\bar{\rho}_i^2}{\rho_i^2} \left( \frac{12}{\beta_i^2} (L_i^0)^2 + \frac{24(\beta_i+2)}{\alpha_P^2} (L_i^0)^2 + \frac{36(\beta_i^2+4)}{\alpha_D^2} (\tilde{L}_i^0)^2 + \frac{144\beta_i^2(2\beta_i+5)}{\alpha_P^2 \alpha_D^2} (\tilde{L}_i^0)^2 \right)$. Then*

$$\frac{1}{K} \sum_{k=0}^{K-1} \sum_{i=1}^{p} \frac{\gamma_i}{\frac{1}{p} \sum_{l=1}^{p} \gamma_l} \mathbb{E}\left[ \left\| \nabla_i f(X^k) \right\|_{(i)\star}^2 \right]$$

$$\leq \quad \frac{1}{K} \frac{4\Psi^0}{\frac{1}{p} \sum_{l=1}^{p} \gamma_l} + 24 \sum_{i=1}^{p} \left( \frac{1}{n} + \frac{(1-\alpha_D)\beta_i}{\alpha_D} + \frac{12\beta_i^2}{\alpha_D^2} \right) \frac{\sigma_i^2 \bar{\rho}_i^2 \beta_i \gamma_i}{\frac{1}{p} \sum_{l=1}^{p} \gamma_l}, \qquad (15)$$

*where*

$$\Psi^0 := f(X^0) - f^\star + \sum_{i=1}^{p} \frac{6\bar{\rho}_i^2}{\beta_i} \gamma_i \mathbb{E}\left[ \left\| \nabla_i f(X^0) - M_i^0 \right\|_2^2 \right]$$

$$+ \sum_{i=1}^{p} \frac{72\bar{\rho}_i^2 \beta_i}{\alpha_D^2} \gamma_i \frac{1}{n} \sum_{j=1}^{n} \mathbb{E}\left[\left\|\nabla_i f_j(X^0) - M_{i,j}^0\right\|_2^2\right] + \sum_{i=1}^{p} \frac{6\bar{\rho}_i^2}{\alpha_D} \gamma_i \frac{1}{n} \sum_{j=1}^{n} \mathbb{E}\left[\left\|M_{i,j}^0 - G_{i,j}^0\right\|_2^2\right]$$

and $M_i^k := \frac{1}{n} \sum_{j=1}^{n} M_{i,j}^k$.

**Corollary 1.** *Let the assumptions of Theorem 19 hold and let $\{X^k\}_{k=0}^{K-1}$, $K \geq 1$, be the iterates of Algorithm 3 initialized with $M_{i,j}^0 = G_{i,j}^0 = \nabla_i f_j(X^0; \xi_j^0)$, $j \in [n]$. Then, the result in Theorem 19 guarantees that*

$$\frac{1}{K} \sum_{k=0}^{K-1} \sum_{i=1}^{p} \frac{\gamma_i}{\frac{1}{p} \sum_{l=1}^{p} \gamma_l} \mathbb{E}\left[\left\|\nabla_i f(X^k)\right\|_{(i)\star}^2\right]$$

$$\leq \quad \frac{4\left(f(X^0) - f^\star\right)}{K \frac{1}{p} \sum_{l=1}^{p} \gamma_l} + \frac{24}{K} \sum_{i=1}^{p} \left(\frac{1}{\sqrt{n}\beta_i} + \frac{12\beta_i}{\alpha_D^2}\right) \frac{\sigma_i \bar{\rho}_i^2 \gamma_i}{\frac{1}{p} \sum_{l=1}^{p} \gamma_l}$$

$$+ 24 \sum_{i=1}^{p} \left(\frac{1}{n} + \frac{(1-\alpha_D)\beta_i}{\alpha_D} + \frac{12\beta_i^2}{\alpha_D^2}\right) \frac{\sigma_i^2 \bar{\rho}_i^2 \beta_i \gamma_i}{\frac{1}{p} \sum_{l=1}^{p} \gamma_l}.$$

**Remark 20.** *Theorem 5 follows as a corollary of the result above by setting $p = 1$.*

**Corollary 2.** *Let the assumptions of Theorem 19 hold and let $\{X^k\}_{k=0}^{K-1}$, $K \geq 1$, be the iterates of Algorithm 1 (Algorithm 3 with $p = 1$) run with $\mathcal{C}_i^k \in \mathbb{B}(\alpha_P)$, $\mathcal{C}_{i,j}^k \in \mathbb{B}_2(\alpha_D)$. Choosing the stepsize*

$$\gamma_1 = \frac{1}{\sqrt{2\zeta_1} + 2L_1^0} = \mathcal{O}\left(\left(\frac{\bar{\rho}_1^2 L_1^0}{\underline{\rho}_1^2 \beta_1} + \frac{\bar{\rho}_1^2 \tilde{L}_1^0}{\underline{\rho}_1^2 \alpha_P \alpha_D}\right)^{-1}\right) \tag{16}$$

*and momentum*

$$\beta_1 = \min\left\{1, \left(\frac{\Psi^0 L_1^0 n}{\bar{\rho}_1^2 \sigma_1^2 K}\right)^{1/2}, \left(\frac{\Psi^0 L_1^0 \alpha_D}{\underline{\rho}_1^2 \sigma_1^2 K}\right)^{1/3}, \left(\frac{\Psi^0 L_1^0 \alpha_D^2}{\underline{\rho}_1^2 \sigma_1^2 K}\right)^{1/4}\right\}, \tag{17}$$

*the result in Theorem 19 guarantees that*

$$\frac{1}{K} \sum_{k=0}^{K-1} \mathbb{E}\left[\left\|\nabla f(X^k)\right\|_\star^2\right]$$

$$= \mathcal{O}\left(\frac{\Psi^0 \bar{\rho}_1^2 \tilde{L}_1^0}{\underline{\rho}_1^2 \alpha_P \alpha_D K} + \left(\frac{\Psi^0 \bar{\rho}_1^4 \sigma_1^2 L_1^0}{\underline{\rho}_1^2 n K}\right)^{1/2} + \left(\frac{\Psi^0 \bar{\rho}_1^3 \sigma_1 L_1^0}{\underline{\rho}_1^2 \sqrt{\alpha_D} K}\right)^{2/3} + \left(\frac{\Psi^0 \underline{\rho}_1^{8/3} \sigma_1^{2/3} L_1^0}{\bar{\rho}_1^2 \alpha_D^{2/3} K}\right)^{3/4}\right).$$

**Remark 21.** *In the Euclidean case ($\bar{\rho}_i^2 = \underline{\rho}_i^2 = 1$), without primal compression ($\alpha_P = 1$), and for $p = 1$, the result in Theorem 19 simplifies to*

$$\frac{1}{K} \sum_{k=0}^{K-1} \mathbb{E}\left[\left\|\nabla f(X^k)\right\|^2\right] = \mathcal{O}\left(\frac{\Psi^0}{K\gamma} + \left(\frac{1}{n} + \frac{\beta}{\alpha_D} + \frac{\beta^2}{\alpha_D^2}\right)\beta\sigma^2\right),$$

*for $\gamma = \mathcal{O}\left(\frac{\beta}{L_1^0} + \frac{\alpha_D}{\tilde{L}_1^0}\right)$, which recovers the rate of* EF21-SDGM *in Fatkhullin et al. (2023, Theorem 3) (up to a constant).*

**Remark 22.** *In the absence of stochasticity and momentum, i.e., when $\sigma_i^2 = 0$ and $\beta_i = 1$, and under the initialization $W^0 = X^0$, $M_j^0 = G_j^0 = \nabla f_j(X^0)$, Algorithm 3 reduces to Algorithm 2. In this setting, Theorem 19 guarantees that*

$$\frac{1}{K} \sum_{k=0}^{K-1} \sum_{i=1}^{p} \frac{\gamma_i}{\frac{1}{p} \sum_{l=1}^{p} \gamma_l} \mathbb{E}\left[\left\|\nabla_i f(X^k)\right\|_{(i)\star}^2\right] \leq \frac{1}{K} \frac{4\left(f(X^0) - f^\star\right)}{\frac{1}{p} \sum_{l=1}^{p} \gamma_l},$$

*for*

$$0 \leq \gamma_i^k \equiv \gamma_i \leq \frac{1}{2L_i^0 + 2\sqrt{\zeta_i}}, \qquad i = 1, \ldots, p,$$

*where $\zeta_i := \frac{\bar{\rho}_i^2}{\underline{\rho}_i^2}\left(12(L_i^0)^2 + \frac{72}{\alpha_P^2}(L_i^0)^2 + \frac{180}{\alpha_D^2}(\tilde{L}_i^0)^2 + \frac{1008}{\alpha_P^2 \alpha_D^2}(\tilde{L}_i^0)^2\right)$. This recovers the guarantee in Theorem 14, up to a constant factor.*

**Remark 23.** *Alternatively, one may use compressors $\mathcal{C}_i^k \in \mathbb{B}_2(\alpha_P)$ in Theorem 19. The proof is essentially the same, with the only modification being the replacement of Lemma 3 by the recursion*

$$
\begin{aligned}
&\mathbb{E}_{\mathcal{C}} \left[ \left\| X_i^{k+1} - W_i^{k+1} \right\|_2^2 \right] \\
&= \quad \mathbb{E}_{\mathcal{C}} \left[ \left\| W_i^k + \mathcal{C}_i^k(X_i^{k+1} - W_i^k) - X_i^{k+1} \right\|_2^2 \right] \\
&\leq \quad (1 - \alpha_P) \left\| X_i^{k+1} - W_i^k \right\|_2^2 \\
&\overset{(28)}{\leq} \quad (1 - \alpha_P) \left( 1 + \frac{\alpha_P}{2} \right) \left\| X_i^k - W_i^k \right\|_2^2 + (1 - \alpha_P) \left( 1 + \frac{2}{\alpha_P} \right) \left\| X_i^{k+1} - X_i^k \right\|_2^2 \\
&\overset{(30),(31)}{\leq} \quad \left( 1 - \frac{\alpha_P}{2} \right) \left\| X_i^k - W_i^k \right\|_2^2 + \frac{2\bar{\rho}_i^2}{\alpha_P} \left\| X_i^{k+1} - X_i^k \right\|_{(i)}^2 \\
&= \quad \left( 1 - \frac{\alpha_P}{2} \right) \left\| X_i^k - W_i^k \right\|_2^2 + \frac{2\bar{\rho}_i^2}{\alpha_P} (\gamma_i^k)^2 \left\| G_i^k \right\|_{(i)\star}^2 .
\end{aligned}
$$

*The resulting convergence guarantee matches that of Theorem 19 up to a modification of the constant $\zeta_i$, which now becomes*

$$
\zeta_i = \frac{\bar{\rho}_i^2}{\underline{\rho}_i^2} \left( \frac{12}{\beta_i^2} (L_i^0)^2 + \frac{24\bar{\rho}_i^2 (\beta_i + 2)}{\alpha_P^2} (L_i^0)^2 + \frac{36 (\beta_i^2 + 4)}{\alpha_D^2} (\tilde{L}_i^0)^2 + \frac{144\bar{\rho}_i^2 \beta_i^2 (2\beta_i + 5)}{\alpha_P^2 \alpha_D^2} (\tilde{L}_i^0)^2 \right),
$$

*where the additional norm equivalence factors highlighted in* red *arise due to the use of Euclidean compressors.*

*Proof of Theorem 19.* Lemma 1 and Young's and Jensen's inequalities give

$$
\begin{aligned}
f(X^{k+1}) \quad &\overset{(1)}{\leq} \quad f(X^k) + \frac{3}{2} \sum_{i=1}^p \gamma_i \left\| \nabla_i f(X^k) - G_i^k \right\|_{(i)\star}^2 - \frac{1}{4} \sum_{i=1}^p \gamma_i \left\| \nabla_i f(X^k) \right\|_{(i)\star}^2 \\
&\qquad - \sum_{i=1}^p \left( \frac{1}{4\gamma_i} - \frac{L_i^0}{2} \right) \gamma_i^2 \left\| G_i^k \right\|_{(i)\star}^2 \\
&\overset{(28)}{\leq} \quad f(X^k) + 3 \sum_{i=1}^p \bar{\rho}_i^2 \gamma_i \left( \left\| \nabla_i f(X^k) - M_i^k \right\|_2^2 + \frac{1}{n} \sum_{j=1}^n \left\| M_{i,j}^k - G_{i,j}^k \right\|_2^2 \right) \\
&\qquad - \frac{1}{4} \sum_{i=1}^p \gamma_i \left\| \nabla_i f(X^k) \right\|_{(i)\star}^2 - \sum_{i=1}^p \left( \frac{1}{4\gamma_i} - \frac{L_i^0}{2} \right) \gamma_i^2 \left\| G_i^k \right\|_{(i)\star}^2 .
\end{aligned}
$$

Recall that by Lemmas 3, 4 and 5, we have

$$
\mathbb{E} \left[ \left\| X_i^{k+1} - W_i^{k+1} \right\|_{(i)}^2 \right] \overset{(3)}{\leq} \left( 1 - \frac{\alpha_P}{2} \right) \mathbb{E} \left[ \left\| X_i^k - W_i^k \right\|_{(i)}^2 \right] + \frac{2}{\alpha_P} \gamma_i^2 \mathbb{E} \left[ \left\| G_i^k \right\|_{(i)\star}^2 \right],
$$

$$
\begin{aligned}
\mathbb{E} \left[ \left\| M_{i,j}^{k+1} - G_{i,j}^{k+1} \right\|_2^2 \right] &\overset{(4)}{\leq} \left( 1 - \frac{\alpha_D}{2} \right) \mathbb{E} \left[ \left\| M_{i,j}^k - G_{i,j}^k \right\|_2^2 \right] + \frac{6\beta_i^2}{\alpha_D} \mathbb{E} \left[ \left\| M_{i,j}^k - \nabla_i f_j(X^k) \right\|_2^2 \right] \\
&\qquad + \frac{6\beta_i^2 (L_{i,j}^0)^2}{\alpha_D \underline{\rho}_i^2} \gamma_i^2 \mathbb{E} \left[ \left\| G_i^k \right\|_\star^2 \right] + \frac{6\beta_i^2 (L_{i,j}^0)^2}{\alpha_D \underline{\rho}_i^2} \mathbb{E} \left[ \left\| X_i^{k+1} - W_i^{k+1} \right\|_{(i)}^2 \right] \\
&\qquad + (1 - \alpha_D) \beta_i^2 \sigma_i^2,
\end{aligned}
$$

$$
\begin{aligned}
\mathbb{E} \left[ \left\| \nabla_i f_j(X^{k+1}) - M_{i,j}^{k+1} \right\|_2^2 \right] &\overset{(5)}{\leq} \left( 1 - \frac{\beta_i}{2} \right) \mathbb{E} \left[ \left\| \nabla_i f_j(X^k) - M_{i,j}^k \right\|_2^2 \right] + \frac{2(L_{i,j}^0)^2}{\beta_i \underline{\rho}_i^2} \gamma_i^2 \mathbb{E} \left[ \left\| G_i^k \right\|_{(i)\star}^2 \right] \\
&\qquad + \frac{\beta_i^2}{\underline{\rho}_i^2} \left( 1 + \frac{2}{\beta_i} \right) (L_{i,j}^0)^2 \mathbb{E} \left[ \left\| X_i^{k+1} - W_i^{k+1} \right\|_{(i)}^2 \right] + \beta_i^2 \sigma_i^2,
\end{aligned}
$$

$$
\mathbb{E} \left[ \left\| \nabla_i f(X^{k+1}) - M_i^{k+1} \right\|_2^2 \right] \overset{(5)}{\leq} \left( 1 - \frac{\beta_i}{2} \right) \mathbb{E} \left[ \left\| \nabla_i f(X^k) - M_i^k \right\|_2^2 \right] + \frac{2(L_i^0)^2}{\beta_i \underline{\rho}_i^2} \gamma_i^2 \mathbb{E} \left[ \left\| G_i^k \right\|_{(i)\star}^2 \right]
$$

$$+ \frac{\beta_i^2}{\underline{\rho}_i^2}\left(1 + \frac{2}{\beta_i}\right)(L_i^0)^2 \mathbb{E}\left[\left\|X_i^{k+1} - W_i^{k+1}\right\|_{(i)}^2\right] + \frac{\beta_i^2 \sigma_i^2}{n},$$

where $M_i^k := \frac{1}{n}\sum_{j=1}^n M_{i,j}^k$. To simplify the notation, let us define $\delta^k := \mathbb{E}\left[f(X^k) - f^\star\right]$, $P_i^k := \gamma_i \mathbb{E}\left[\left\|\nabla_i f(X^k) - M_i^k\right\|_2^2\right]$, $\tilde{P}_i^k := \gamma_i \frac{1}{n}\sum_{j=1}^n \mathbb{E}\left[\left\|\nabla_i f_j(X^k) - M_{i,j}^k\right\|_2^2\right]$, $\tilde{S}_i^k := \gamma_i \frac{1}{n}\sum_{j=1}^n \mathbb{E}\left[\left\|M_{i,j}^k - G_{i,j}^k\right\|_2^2\right]$ and $R_i^k := \gamma_i \mathbb{E}\left[\left\|X_i^k - W_i^k\right\|_{(i)}^2\right]$. Then, the above inequalities yield

$$\begin{aligned}
\delta^{k+1} &\leq \delta^k + 3\sum_{i=1}^p \bar{\rho}_i^2 P_i^k + 3\sum_{i=1}^p \bar{\rho}_i^2 \tilde{S}_i^k - \frac{1}{4}\sum_{i=1}^p \gamma_i \mathbb{E}\left[\left\|\nabla_i f(X^k)\right\|_{(i)\star}^2\right] \\
&\quad - \sum_{i=1}^p \left(\frac{1}{4\gamma_i} - \frac{L_i^0}{2}\right)\gamma_i^2 \mathbb{E}\left[\left\|G_i^k\right\|_{(i)\star}^2\right], \tag{18}
\end{aligned}$$

$$R_i^{k+1} \leq \left(1 - \frac{\alpha_P}{2}\right)R_i^k + \frac{2}{\alpha_P}\gamma_i^3 \mathbb{E}\left[\left\|G_i^k\right\|_{(i)\star}^2\right], \tag{19}$$

$$\begin{aligned}
\tilde{S}_i^{k+1} &\leq \left(1 - \frac{\alpha_D}{2}\right)\tilde{S}_i^k + \frac{6\beta_i^2}{\alpha_D}\tilde{P}_i^k + \frac{6\beta_i^2(\tilde{L}_i^0)^2}{\alpha_D \underline{\rho}_i^2}\gamma_i^3 \mathbb{E}\left[\left\|G_i^k\right\|_{(i)\star}^2\right] \\
&\quad + \frac{6\beta_i^2(\tilde{L}_i^0)^2}{\alpha_D \underline{\rho}_i^2}R_i^{k+1} + (1 - \alpha_D)\sigma_i^2 \beta_i^2 \gamma_i, \tag{20}
\end{aligned}$$

$$\begin{aligned}
\tilde{P}_i^{k+1} &\leq \left(1 - \frac{\beta_i}{2}\right)\tilde{P}_i^k + \frac{2(\tilde{L}_i^0)^2}{\beta_i \underline{\rho}_i^2}\gamma_i^3 \mathbb{E}\left[\left\|G_i^k\right\|_{(i)\star}^2\right] \\
&\quad + \frac{\beta_i^2}{\underline{\rho}_i^2}\left(1 + \frac{2}{\beta_i}\right)(\tilde{L}_i^0)^2 R_i^{k+1} + \sigma_i^2 \beta_i^2 \gamma_i, \tag{21}
\end{aligned}$$

$$\begin{aligned}
P_i^{k+1} &\leq \left(1 - \frac{\beta_i}{2}\right)P_i^k + \frac{2(L_i^0)^2}{\beta_i \underline{\rho}_i^2}\gamma_i^3 \mathbb{E}\left[\left\|G_i^k\right\|_{(i)\star}^2\right] \\
&\quad + \frac{\beta_i^2}{\underline{\rho}_i^2}\left(1 + \frac{2}{\beta_i}\right)(L_i^0)^2 R_i^{k+1} + \frac{\sigma_i^2 \beta_i^2 \gamma_i}{n}. \tag{22}
\end{aligned}$$

Now, let $A_i, B_i, C_i, D_i > 0$ be some constants to be determined later, and define

$$\Psi^k := \delta^k + \sum_{i=1}^p A_i P_i^k + \sum_{i=1}^p B_i \tilde{P}_i^k + \sum_{i=1}^p C_i \tilde{S}_i^k + \sum_{i=1}^p D_i R_i^k.$$

Then, applying (18), (20), (21), and (22), we have

$$\Psi^{k+1}$$

$$= \delta^{k+1} + \sum_{i=1}^p A_i P_i^{k+1} + \sum_{i=1}^p B_i \tilde{P}_i^{k+1} + \sum_{i=1}^p C_i \tilde{S}_i^{k+1} + \sum_{i=1}^p D_i R_i^{k+1}$$

$$\leq \delta^k + 3\sum_{i=1}^p \bar{\rho}_i^2 P_i^k + 3\sum_{i=1}^p \bar{\rho}_i^2 \tilde{S}_i^k - \frac{1}{4}\sum_{i=1}^p \gamma_i \mathbb{E}\left[\left\|\nabla_i f(X^k)\right\|_{(i)\star}^2\right]$$

$$\quad - \sum_{i=1}^p \left(\frac{1}{4\gamma_i} - \frac{L_i^0}{2}\right)\gamma_i^2 \mathbb{E}\left[\left\|G_i^k\right\|_{(i)\star}^2\right]$$

$$\quad + \sum_{i=1}^p A_i\left(\left(1 - \frac{\beta_i}{2}\right)P_i^k + \frac{2(L_i^0)^2}{\beta_i \underline{\rho}_i^2}\gamma_i^3 \mathbb{E}\left[\left\|G_i^k\right\|_{(i)\star}^2\right] + \frac{\beta_i^2}{\underline{\rho}_i^2}\left(1 + \frac{2}{\beta_i}\right)(L_i^0)^2 R_i^{k+1} + \frac{\sigma_i^2 \beta_i^2 \gamma_i}{n}\right)$$

$$\quad + \sum_{i=1}^p B_i\left(\left(1 - \frac{\beta_i}{2}\right)\tilde{P}_i^k + \frac{2(\tilde{L}_i^0)^2}{\beta_i \underline{\rho}_i^2}\gamma_i^3 \mathbb{E}\left[\left\|G_i^k\right\|_{(i)\star}^2\right] + \frac{\beta_i^2}{\underline{\rho}_i^2}\left(1 + \frac{2}{\beta_i}\right)(\tilde{L}_i^0)^2 R_i^{k+1} + \sigma_i^2 \beta_i^2 \gamma_i\right)$$

$$\quad + \sum_{i=1}^p C_i\left(\left(1 - \frac{\alpha_D}{2}\right)\tilde{S}_i^k + \frac{6\beta_i^2}{\alpha_D}\tilde{P}_i^k + \frac{6\beta_i^2(\tilde{L}_i^0)^2}{\alpha_D \underline{\rho}_i^2}\gamma_i^3 \mathbb{E}\left[\left\|G_i^k\right\|_{(i)\star}^2\right]\right)$$

$$+ \sum_{i=1}^{p} C_i \left( \frac{6\beta_i^2 (\tilde{L}_i^0)^2}{\alpha_D \underline{\rho}_i^2} R_i^{k+1} + (1-\alpha_D)\sigma_i^2 \beta_i^2 \gamma_i \right) + \sum_{i=1}^{p} D_i R_i^{k+1}$$

$$= \delta^k + \sum_{i=1}^{p} \left( 3\bar{\rho}_i^2 + A_i \left( 1 - \frac{\beta_i}{2} \right) \right) P_i^k + \sum_{i=1}^{p} \left( B_i \left( 1 - \frac{\beta_i}{2} \right) + C_i \frac{6\beta_i^2}{\alpha_D} \right) \tilde{P}_i^k$$

$$+ \sum_{i=1}^{p} \left( 3\bar{\rho}_i^2 + C_i \left( 1 - \frac{\alpha_D}{2} \right) \right) \tilde{S}_i^k - \frac{1}{4} \sum_{i=1}^{p} \gamma_i \mathbb{E} \left[ \left\| \nabla_i f(X^k) \right\|_{(i)\star}^2 \right]$$

$$+ \sum_{i=1}^{p} \left( A_i \frac{\beta_i^2}{\underline{\rho}_i^2} \left( 1 + \frac{2}{\beta_i} \right) (L_i^0)^2 + B_i \frac{\beta_i^2}{\underline{\rho}_i^2} \left( 1 + \frac{2}{\beta_i} \right) (\tilde{L}_i^0)^2 + C_i \frac{6\beta_i^2 (\tilde{L}_i^0)^2}{\alpha_D \underline{\rho}_i^2} + D_i \right) R_i^{k+1}$$

$$+ \sum_{i=1}^{p} \left( A_i \frac{2(L_i^0)^2}{\beta_i \underline{\rho}_i^2} + B_i \frac{2(\tilde{L}_i^0)^2}{\beta_i \underline{\rho}_i^2} + C_i \frac{6\beta_i^2 (\tilde{L}_i^0)^2}{\alpha_D \underline{\rho}_i^2} \right) \gamma_i^3 \mathbb{E} \left[ \left\| G_i^k \right\|_{(i)\star}^2 \right]$$

$$- \sum_{i=1}^{p} \left( \frac{1}{4\gamma_i} - \frac{L_i^0}{2} \right) \gamma_i^2 \mathbb{E} \left[ \left\| G_i^k \right\|_{(i)\star}^2 \right] + \sum_{i=1}^{p} \left( \frac{A_i}{n} + B_i + C_i(1-\alpha_D) \right) \sigma_i^2 \beta_i^2 \gamma_i.$$

Then, using (19) gives

$$\Psi^{k+1}$$

$$\leq \delta^k + \sum_{i=1}^{p} \left( 3\bar{\rho}_i^2 + A_i \left( 1 - \frac{\beta_i}{2} \right) \right) P_i^k + \sum_{i=1}^{p} \left( B_i \left( 1 - \frac{\beta_i}{2} \right) + C_i \frac{6\beta_i^2}{\alpha_D} \right) \tilde{P}_i^k$$

$$+ \sum_{i=1}^{p} \left( 3\bar{\rho}_i^2 + C_i \left( 1 - \frac{\alpha_D}{2} \right) \right) \tilde{S}_i^k - \frac{1}{4} \sum_{i=1}^{p} \gamma_i \mathbb{E} \left[ \left\| \nabla_i f(X^k) \right\|_{(i)\star}^2 \right]$$

$$+ \sum_{i=1}^{p} \left( A_i \frac{\beta_i^2}{\underline{\rho}_i^2} \left( 1 + \frac{2}{\beta_i} \right) (L_i^0)^2 + B_i \frac{\beta_i^2}{\underline{\rho}_i^2} \left( 1 + \frac{2}{\beta_i} \right) (\tilde{L}_i^0)^2 + C_i \frac{6\beta_i^2 (\tilde{L}_i^0)^2}{\alpha_D \underline{\rho}_i^2} + D_i \right) \left( 1 - \frac{\alpha_P}{2} \right) R_i^k$$

$$+ \sum_{i=1}^{p} \left( A_i \frac{\beta_i^2}{\underline{\rho}_i^2} \left( 1 + \frac{2}{\beta_i} \right) (L_i^0)^2 + B_i \frac{\beta_i^2}{\underline{\rho}_i^2} \left( 1 + \frac{2}{\beta_i} \right) (\tilde{L}_i^0)^2 + C_i \frac{6\beta_i^2 (\tilde{L}_i^0)^2}{\alpha_D \underline{\rho}_i^2} + D_i \right) \frac{2}{\alpha_P} \gamma_i^3 \mathbb{E} \left[ \left\| G_i^k \right\|_{(i)\star}^2 \right]$$

$$+ \sum_{i=1}^{p} \left( A_i \frac{2(L_i^0)^2}{\beta_i \underline{\rho}_i^2} + B_i \frac{2(\tilde{L}_i^0)^2}{\beta_i \underline{\rho}_i^2} + C_i \frac{6\beta_i^2 (\tilde{L}_i^0)^2}{\alpha_D \underline{\rho}_i^2} \right) \gamma_i^3 \mathbb{E} \left[ \left\| G_i^k \right\|_{(i)\star}^2 \right]$$

$$- \sum_{i=1}^{p} \left( \frac{1}{4\gamma_i} - \frac{L_i^0}{2} \right) \gamma_i^2 \mathbb{E} \left[ \left\| G_i^k \right\|_{(i)\star}^2 \right] + \sum_{i=1}^{p} \left( \frac{A_i}{n} + B_i + C_i(1-\alpha_D) \right) \sigma_i^2 \beta_i^2 \gamma_i.$$

Taking $A_i = \frac{6\bar{\rho}_i^2}{\beta_i}$, $B_i = \frac{72\bar{\rho}_i^2 \beta_i}{\alpha_D^2}$, $C_i = \frac{6\bar{\rho}_i^2}{\alpha_D}$ and

$$D_i = \left( A_i \frac{\beta_i^2}{\underline{\rho}_i^2} \left( 1 + \frac{2}{\beta_i} \right) (L_i^0)^2 + B_i \frac{\beta_i^2}{\underline{\rho}_i^2} \left( 1 + \frac{2}{\beta_i} \right) (\tilde{L}_i^0)^2 + C_i \frac{6\beta_i^2 (\tilde{L}_i^0)^2}{\alpha_D \underline{\rho}_i^2} \right) \left( \frac{2}{\alpha_P} - 1 \right)$$

$$= \frac{6\bar{\rho}_i^2}{\underline{\rho}_i^2} \left( (\beta_i + 2)(L_i^0)^2 + \frac{6\beta_i^2 (2\beta_i + 5)}{\alpha_D^2} (\tilde{L}_i^0)^2 \right) \left( \frac{2}{\alpha_P} - 1 \right),$$

we obtain

$$3\bar{\rho}_i^2 + A_i \left( 1 - \frac{\beta_i}{2} \right) = 3\bar{\rho}_i^2 + \frac{6\bar{\rho}_i^2}{\beta_i} \left( 1 - \frac{\beta_i}{2} \right) = A_i,$$

$$B_i \left( 1 - \frac{\beta_i}{2} \right) + C_i \frac{6\beta_i^2}{\alpha_D} = \frac{72\bar{\rho}_i^2 \beta_i}{\alpha_D^2} \left( 1 - \frac{\beta_i}{2} \right) + \frac{6\bar{\rho}_i^2}{\alpha_D} \frac{6\beta_i^2}{\alpha_D} = B_i,$$

$$3\bar{\rho}_i^2 + C_i \left( 1 - \frac{\alpha_D}{2} \right) = 3\bar{\rho}_i^2 + \frac{6\bar{\rho}_i^2}{\alpha_D} \left( 1 - \frac{\alpha_D}{2} \right) = C_i,$$

and

$$\left( A_i \frac{\beta_i^2}{\underline{\rho}_i^2} \left( 1 + \frac{2}{\beta_i} \right) (L_i^0)^2 + B_i \frac{\beta_i^2}{\underline{\rho}_i^2} \left( 1 + \frac{2}{\beta_i} \right) (\tilde{L}_i^0)^2 + C_i \frac{6\beta_i^2 (\tilde{L}_i^0)^2}{\alpha_D \underline{\rho}_i^2} + D_i \right) \left( 1 - \frac{\alpha_P}{2} \right)$$

$$= \left( \frac{D_i}{\frac{2}{\alpha_P} - 1} + D_i \right) \left( 1 - \frac{\alpha_P}{2} \right) = D_i.$$

Consequently,

$$\Psi^{k+1}$$
$$\leq \delta^k + \sum_{i=1}^{p} A_i P_i^k + \sum_{i=1}^{p} B_i \tilde{P}_i^k + \sum_{i=1}^{p} C_i \tilde{S}_i^k + \sum_{i=1}^{p} D_i R_i^k - \frac{1}{4} \sum_{i=1}^{p} \gamma_i \mathbb{E} \left[ \left\| \nabla_i f(X^k) \right\|_{(i)\star}^2 \right]$$
$$+ \sum_{i=1}^{p} \left( \frac{D_i}{\frac{2}{\alpha_P} - 1} + D_i \right) \frac{2}{\alpha_P} \gamma_i^3 \mathbb{E} \left[ \left\| G_i^k \right\|_{(i)\star}^2 \right] - \sum_{i=1}^{p} \left( \frac{1}{4\gamma_i} - \frac{L_i^0}{2} \right) \gamma_i^2 \mathbb{E} \left[ \left\| G_i^k \right\|_{(i)\star}^2 \right]$$
$$+ \sum_{i=1}^{p} \left( \frac{6\bar{\rho}_i^2}{\beta_i} \frac{2(L_i^0)^2}{\beta_i \rho_i^2} + \frac{72\bar{\rho}_i^2 \beta_i}{\alpha_D^2} \frac{2(\tilde{L}_i^0)^2}{\beta_i \rho_i^2} + \frac{6\bar{\rho}_i^2}{\alpha_D} \frac{6\beta_i^2(\tilde{L}_i^0)^2}{\alpha_D \rho_i^2} \right) \gamma_i^3 \mathbb{E} \left[ \left\| G_i^k \right\|_{(i)\star}^2 \right]$$
$$+ \sum_{i=1}^{p} \left( \frac{1}{n} \frac{6\bar{\rho}_i^2}{\beta_i} + \frac{72\bar{\rho}_i^2 \beta_i}{\alpha_D^2} + \frac{6\bar{\rho}_i^2}{\alpha_D}(1 - \alpha_D) \right) \sigma_i^2 \beta_i^2 \gamma_i$$
$$= \Psi^k - \frac{1}{4} \sum_{i=1}^{p} \gamma_i \mathbb{E} \left[ \left\| \nabla_i f(X^k) \right\|_{(i)\star}^2 \right] - \sum_{i=1}^{p} \left( \frac{1}{4\gamma_i} - \frac{L_i^0}{2} \right) \gamma_i^2 \mathbb{E} \left[ \left\| G_i^k \right\|_{(i)\star}^2 \right]$$
$$+ \sum_{i=1}^{p} \left( \frac{12\bar{\rho}_i^2}{\beta_i^2 \rho_i^2}(L_i^0)^2 + \frac{144\bar{\rho}_i^2}{\alpha_D^2 \rho_i^2}(\tilde{L}_i^0)^2 + \frac{36\beta_i^2 \bar{\rho}_i^2}{\alpha_D^2 \rho_i^2}(\tilde{L}_i^0)^2 + \frac{4D_i}{\alpha_P(2 - \alpha_P)} \right) \gamma_i^3 \mathbb{E} \left[ \left\| G_i^k \right\|_{(i)\star}^2 \right]$$
$$+ 6 \sum_{i=1}^{p} \left( \frac{1}{n} + \frac{12\beta_i^2}{\alpha_D^2} + \frac{(1 - \alpha_D)\beta_i}{\alpha_D} \right) \sigma_i^2 \bar{\rho}_i^2 \beta_i \gamma_i.$$

Now, note that

$$\frac{1}{4\gamma_i} - \frac{L_i^0}{2} - \gamma_i \left( \frac{12\bar{\rho}_i^2}{\beta_i^2 \rho_i^2}(L_i^0)^2 + \frac{144\bar{\rho}_i^2}{\alpha_D^2 \rho_i^2}(\tilde{L}_i^0)^2 + \frac{36\beta_i^2 \bar{\rho}_i^2}{\alpha_D^2 \rho_i^2}(\tilde{L}_i^0)^2 + \frac{4D_i}{\alpha_P(2 - \alpha_P)} \right)$$
$$= \frac{1}{4\gamma_i} - \frac{L_i^0}{2}$$
$$- \gamma_i \underbrace{\frac{\bar{\rho}_i^2}{\rho_i^2} \left( \frac{12}{\beta_i^2}(L_i^0)^2 + \frac{24(\beta_i + 2)}{\alpha_P^2}(L_i^0)^2 + \frac{36(\beta_i^2 + 4)}{\alpha_D^2}(\tilde{L}_i^0)^2 + \frac{144\beta_i^2(2\beta_i + 5)}{\alpha_P^2 \alpha_D^2}(\tilde{L}_i^0)^2 \right)}_{:= \zeta_i} \geq 0$$

for $\gamma_i \leq \frac{1}{2\sqrt{\zeta_i} + 2L_i^0}$. For such a choice of the stepsizes, we have

$$\Psi^{k+1} \leq \Psi^k - \frac{1}{4} \sum_{i=1}^{p} \gamma_i \mathbb{E} \left[ \left\| \nabla_i f(X^k) \right\|_{(i)\star}^2 \right] + \sum_{i=1}^{p} \underbrace{6 \left( \frac{1}{n} + \frac{12\beta_i^2}{\alpha_D^2} + \frac{(1 - \alpha_D)\beta_i}{\alpha_D} \right) \sigma_i^2 \bar{\rho}_i^2 \beta_i}_{:= \xi_i} \gamma_i.$$

Summing over the first $K$ iterations gives

$$\sum_{k=0}^{K-1} \sum_{i=1}^{p} \gamma_i \mathbb{E} \left[ \left\| \nabla_i f(X^k) \right\|_{(i)\star}^2 \right] \leq 4 \sum_{k=0}^{K-1} \left( \Psi^k - \Psi^{k+1} \right) + 4 \sum_{k=0}^{K-1} \sum_{i=1}^{p} \xi_i \gamma_i \leq 4\Psi^0 + 4K \sum_{i=1}^{p} \xi_i \gamma_i,$$

and lastly, dividing by $\frac{K}{p} \sum_{l=1}^{p} \gamma_l$, we obtain

$$\frac{1}{K} \sum_{k=0}^{K-1} \sum_{i=1}^{p} \frac{\gamma_i}{\frac{1}{p} \sum_{l=1}^{p} \gamma_l} \mathbb{E} \left[ \left\| \nabla_i f(X^k) \right\|_{(i)\star}^2 \right] \leq \frac{4\Psi^0 p}{K \sum_{l=1}^{p} \gamma_l} + \frac{4 \sum_{i=1}^{p} \xi_i \gamma_i}{\frac{1}{p} \sum_{i=1}^{p} \gamma_i}.$$

Substituting $X_i^0 = W_i^0$ proves the theorem statement. $\qquad \square$

*Proof of Corollary 1.* Substituting the initialization, we have

$$\mathbb{E}\left[\left\|\nabla_i f(X^0) - M_i^0\right\|_2\right] = \mathbb{E}\left[\left\|\frac{1}{n}\sum_{j=1}^n \left(\nabla_i f_j(X^0) - \nabla_i f_j(X^0; \xi_j^0)\right)\right\|_2\right]$$

$$\leq \sqrt{\mathbb{E}\left[\left\|\frac{1}{n}\sum_{j=1}^n \left(\nabla_i f_j(X^0) - \nabla_i f_j(X^0; \xi_j^0)\right)\right\|_2^2\right]} \overset{(10)}{\leq} \frac{\sigma_i}{\sqrt{n}},$$

$$\frac{1}{n}\sum_{j=1}^n \mathbb{E}\left[\left\|\nabla_i f_j(X^0) - M_{i,j}^0\right\|_2\right] = \frac{1}{n}\sum_{j=1}^n \mathbb{E}\left[\left\|\nabla_i f_j(X^0) - \nabla_i f_j(X^0; \xi_j^0)\right\|_2\right] \overset{(10)}{\leq} \sigma_i,$$

and hence

$$\Psi^0 := f(X^0) - f^\star + \sum_{i=1}^p \frac{6\bar{\rho}_i^2}{\beta_i}\gamma_i \mathbb{E}\left[\left\|\nabla_i f(X^0) - M_i^0\right\|_2^2\right]$$

$$+ \sum_{i=1}^p \frac{72\bar{\rho}_i^2\beta_i}{\alpha_D^2}\gamma_i \frac{1}{n}\sum_{j=1}^n \mathbb{E}\left[\left\|\nabla_i f_j(X^0) - M_{i,j}^0\right\|_2^2\right] + \sum_{i=1}^p \frac{6\bar{\rho}_i^2}{\alpha_D}\gamma_i \frac{1}{n}\sum_{j=1}^n \mathbb{E}\left[\left\|M_{i,j}^0 - G_{i,j}^0\right\|_2^2\right]$$

$$\leq f(X^0) - f^\star + \sum_{i=1}^p \frac{6\bar{\rho}_i^2}{\sqrt{n}\beta_i}\gamma_i\sigma_i + \sum_{i=1}^p \frac{72\bar{\rho}_i^2\beta_i}{\alpha_D^2}\gamma_i\sigma_i.$$

Substituting this in the rate, we get

$$\frac{1}{K}\sum_{k=0}^{K-1}\sum_{i=1}^p \frac{\gamma_i}{\frac{1}{p}\sum_{l=1}^p \gamma_l}\mathbb{E}\left[\left\|\nabla_i f(X^k)\right\|_{(i)\star}^2\right]$$

$$\leq \frac{4\left(f(X^0) - f^\star\right)}{K\frac{1}{p}\sum_{l=1}^p \gamma_l} + \frac{24}{K}\sum_{i=1}^p \left(\frac{1}{\sqrt{n}\beta_i} + \frac{12\beta_i}{\alpha_D^2}\right)\frac{\sigma_i\bar{\rho}_i^2\gamma_i}{\frac{1}{p}\sum_{l=1}^p \gamma_l}$$

$$+ 24\sum_{i=1}^p \left(\frac{1}{n} + \frac{(1-\alpha_D)\beta_i}{\alpha_D} + \frac{12\beta_i^2}{\alpha_D^2}\right)\frac{\sigma_i^2\bar{\rho}_i^2\beta_i\gamma_i}{\frac{1}{p}\sum_{l=1}^p \gamma_l}.$$

$\square$

*Proof of Corollary 2.* Substituting the choice of $\gamma$ from (16) in (15), we have

$$\frac{1}{K}\sum_{k=0}^{K-1}\mathbb{E}\left[\left\|\nabla f(X^k)\right\|_\star^2\right] \leq \frac{4\Psi^0}{K\gamma_1} + 24\left(\frac{1}{n} + \frac{(1-\alpha_D)\beta_1}{\alpha_D} + \frac{12\beta_1^2}{\alpha_D^2}\right)\sigma_1^2\bar{\rho}_1^2\beta_1$$

$$= \mathcal{O}\left(\frac{\Psi^0\bar{\rho}_1^2\tilde{L}_1^0}{\underline{\rho}_1^2\alpha_P\alpha_D K} + \frac{\Psi^0\bar{\rho}_1^2 L_1^0}{\underline{\rho}_1^2\beta_1 K} + \frac{\bar{\rho}_1^2\beta_1\sigma_1^2}{n} + \frac{\bar{\rho}_1^2\beta_1^2\sigma_1^2}{\alpha_D} + \frac{\bar{\rho}_1^2\beta_1^3\sigma_1^2}{\alpha_D^2}\right).$$

Then, choosing $\beta_1$ as in (17) guarantees that $\frac{\bar{\rho}_1^2\beta_1\sigma_1^2}{n}, \frac{\bar{\rho}_1^2\beta_1^2\sigma_1^2}{\alpha_D}, \frac{\bar{\rho}_1^2\beta_1^3\sigma_1^2}{\alpha_D^2} \leq \frac{\Psi^0\bar{\rho}_1^2 L_1^0}{\underline{\rho}_1^2\beta_1 K}$. Substituting this into the upper bound gives

$$\frac{1}{K}\sum_{k=0}^{K-1}\mathbb{E}\left[\left\|\nabla f(X^k)\right\|_\star^2\right]$$

$$\leq \mathcal{O}\left(\frac{\Psi^0\bar{\rho}_1^2\tilde{L}_1^0}{\underline{\rho}_1^2\alpha_P\alpha_D K} + \left(\frac{\Psi^0\bar{\rho}_1^4\sigma_1^2 L_1^0}{\underline{\rho}_1^2 n K}\right)^{1/2} + \left(\frac{\Psi^0\bar{\rho}_1^3\sigma_1 L_1^0}{\underline{\rho}_1^2\sqrt{\alpha_D}K}\right)^{2/3} + \left(\frac{\Psi^0\underline{\rho}_1^{8/3}\sigma_1^{2/3}L_1^0}{\bar{\rho}_1^2\alpha_D^{2/3}K}\right)^{3/4}\right)$$

as needed. $\square$

### E.4.2 LAYER-WISE $(L^0, L^1)$–SMOOTH REGIME

As in Section E.3.2, in the generalized smooth setting we consider EF21-Muon without primal compression.

**Theorem 24.** *Let Assumptions 1, 2, 8, 9 and 10 hold. Let $\{X^k\}_{k=0}^{K-1}$, $K \geq 1$, be the iterates of Algorithm 3 run with $\mathcal{C}_i^k \equiv \mathcal{I}$ (the identity compressor), $\mathcal{C}_{i,j}^k \in \mathbb{B}_2(\alpha_D)$, $\beta_i \equiv \beta = \frac{1}{(K+1)^{1/2}}$ and*

$$0 \leq t_i^k \equiv t_i = \frac{\eta_i}{(K+1)^{3/4}}, \qquad i = 1, \ldots, p,$$

*where $\eta_i^2 \leq \min\left\{ \frac{(K+1)^{1/2}}{6(L_i^1)^2}, \frac{(1-\sqrt{1-\alpha_D})\underline{\rho}_i(K+1)^{1/2}}{24\sqrt{1-\alpha_D}\bar{\rho}_i(L_{i,\max}^1)^2}, \frac{\beta_{\min}\underline{\rho}_i(K+1)^{1/2}}{24\bar{\rho}_i(L_{i,\max}^1)^2}, 1 \right\}$. Then*

$$\min_{k=0,\ldots,K} \sum_{i=1}^p \frac{\eta_i}{\frac{1}{p}\sum_{l=1}^p \eta_l} \mathbb{E}\left[ \|\nabla_i f(X^k)\|_{(i)\star} \right]$$

$$\leq \quad \frac{3\Psi^0}{(K+1)^{1/4}\frac{1}{p}\sum_{l=1}^p \eta_l} + \frac{6}{(K+1)^{1/2}} \sum_{i=1}^p \frac{\eta_i \bar{\rho}_i}{\frac{1}{p}\sum_{l=1}^p \eta_l} \mathbb{E}\left[ \|\nabla_i f(X^0) - M_i^0\|_2 \right]$$

$$+ \left( \frac{8}{(K+1)^{1/4}} + \frac{8\sqrt{1-\alpha_D}}{(1-\sqrt{1-\alpha_D})(K+1)^{3/4}} \right) \frac{1}{n} \sum_{j=1}^n \frac{\max_{i\in[p]} \eta_i^2 \frac{\bar{\rho}_i}{\underline{\rho}_i}(L_{i,j}^1)^2}{\frac{1}{p}\sum_{l=1}^p \eta_l} \left( f^\star - f_j^\star \right)$$

$$+ \sum_{i=1}^p \frac{\eta_i^2}{\frac{1}{p}\sum_{l=1}^p \eta_l} \left( \frac{L_i^0}{(K+1)^{3/4}} + \frac{4\bar{\rho}_i \bar{L}_i^0}{\underline{\rho}_i(K+1)^{1/4}} + \frac{4\bar{\rho}_i\sqrt{1-\alpha_D}\bar{L}_i^0}{\underline{\rho}_i(1-\sqrt{1-\alpha_D})(K+1)^{3/4}} \right)$$

$$+ \sum_{i=1}^p \frac{\eta_i \bar{\rho}_i \sigma_i}{\frac{1}{p}\sum_{l=1}^p \eta_l} \left( \frac{4\sqrt{1-\alpha_D}}{(1-\sqrt{1-\alpha_D})(K+1)^{1/2}} + \frac{2}{\sqrt{n}(K+1)^{1/4}} \right),$$

*where $M_i^0 := \frac{1}{n}\sum_{j=1}^n M_{i,j}^0$ and*

$$\Psi^0 \quad := \quad f(X^0) - f^\star + \sum_{i=1}^p \frac{2t_i \bar{\rho}_i}{1-\sqrt{1-\alpha_D}} \frac{1}{n} \sum_{j=1}^n \mathbb{E}\left[ \|M_{i,j}^0 - G_{i,j}^0\|_2 \right]$$

$$+ \sum_{i=1}^p \frac{2t_i \bar{\rho}_i \sqrt{1-\alpha_D}}{1-\sqrt{1-\alpha_D}} \frac{1}{n} \sum_{j=1}^n \mathbb{E}\left[ \|\nabla_i f_j(X^0) - M_{i,j}^0\|_2 \right].$$

**Corollary 3.** *Let the assumptions of Theorem 24 hold and let $\{X^k\}_{k=0}^{K-1}$, $K \geq 1$, be the iterates of Algorithm 3 initialized with $M_{i,j}^0 = \nabla_i f_j(X^0; \xi_j^0)$, $G_{i,j}^0 = \mathcal{C}_{i,j}^0(\nabla_i f_j(X^0; \xi_j^0))$, $j \in [n]$, and run with $\mathcal{C}_i^k \equiv \mathcal{I}$ (the identity compressor), $\mathcal{C}_{i,j}^k \in \mathbb{B}_2(\alpha_D)$, $\beta_i \equiv \beta = \frac{1}{(K+1)^{1/2}}$ and*

$$0 \leq t_i^k \equiv t_i = \frac{\eta_i}{(K+1)^{3/4}}, \qquad i = 1, \ldots, p,$$

*where $\eta_i^2 \leq \min\left\{ \frac{(K+1)^{1/2}}{6(L_i^1)^2}, \frac{(1-\sqrt{1-\alpha_D})\underline{\rho}_i(K+1)^{1/2}}{24\sqrt{1-\alpha_D}\bar{\rho}_i(L_{i,\max}^1)^2}, \frac{\beta_{\min}\underline{\rho}_i(K+1)^{1/2}}{24\bar{\rho}_i(L_{i,\max}^1)^2}, 1 \right\}$. Then, the result in Theorem 19 guarantees that*

$$\min_{k=0,\ldots,K} \sum_{i=1}^p \frac{\eta_i}{\frac{1}{p}\sum_{l=1}^p \eta_l} \mathbb{E}\left[ \|\nabla_i f(X^k)\|_{(i)\star} \right]$$

$$\leq \quad \frac{3}{(K+1)^{1/4}\frac{1}{p}\sum_{l=1}^p \eta_l} \left( f(X^0) - f^\star + \sum_{i=1}^p \frac{4\sqrt{1-\alpha_D}\eta_i \bar{\rho}_i \sigma_i}{(K+1)^{3/4}(1-\sqrt{1-\alpha_D})} \right)$$

$$+ \frac{6}{(K+1)^{1/2}} \sum_{i=1}^p \frac{\bar{\rho}_i \eta_i \sigma_i}{\sqrt{n}\frac{1}{p}\sum_{l=1}^p \eta_l}$$

$$+ \left( \frac{8}{(K+1)^{1/4}} + \frac{8\sqrt{1-\alpha_D}}{(K+1)^{3/4}(1-\sqrt{1-\alpha_D})} \right) \frac{1}{n} \sum_{j=1}^n \frac{\max_{i\in[p]} \eta_i^2 \frac{\bar{\rho}_i}{\underline{\rho}_i}(L_{i,j}^1)^2}{\frac{1}{p}\sum_{l=1}^p \eta_l} \left( f^\star - f_j^\star \right)$$

$$+ \sum_{i=1}^{p} \frac{\eta_i^2}{\frac{1}{p}\sum_{l=1}^{p}\eta_l} \left( \frac{L_i^0}{(K+1)^{3/4}} + \frac{4\bar{\rho}_i \bar{L}_i^0}{\underline{\rho}_i(K+1)^{1/4}} + \frac{4\bar{\rho}_i\sqrt{1-\alpha_D}\bar{L}_i^0}{\underline{\rho}_i(K+1)^{3/4}(1-\sqrt{1-\alpha_D})} \right)$$

$$+ \sum_{i=1}^{p} \frac{\eta_i\bar{\rho}_i\sigma_i}{\frac{1}{p}\sum_{l=1}^{p}\eta_l} \left( \frac{4\sqrt{1-\alpha_D}}{(K+1)^{1/2}(1-\sqrt{1-\alpha_D})} + \frac{2}{\sqrt{n}(K+1)^{1/4}} \right).$$

**Remark 25.** *Theorem 6 follows from Corollary 3 by setting $p=1$:*

$$\min_{k=0,\dots,K} \mathbb{E}\left[\left\|\nabla f(X^k)\right\|_\star\right]$$

$$\leq \frac{3\left(f(X^0)-f^\star\right)}{\eta(K+1)^{1/4}} + \frac{12\sqrt{1-\alpha_D}\bar{\rho}\sigma}{(1-\sqrt{1-\alpha_D})(K+1)} + \frac{6\bar{\rho}\sigma}{\sqrt{n}(K+1)^{1/2}}$$

$$+ \frac{\eta\bar{\rho}}{\underline{\rho}} \left( \frac{8}{(K+1)^{1/4}} + \frac{8\sqrt{1-\alpha_D}}{(1-\sqrt{1-\alpha_D})(K+1)^{3/4}} \right) \frac{1}{n}\sum_{j=1}^{n}(L_j^1)^2\left(f^\star - f_j^\star\right)$$

$$+ \frac{\eta L^0}{(K+1)^{3/4}} + \frac{\eta\bar{\rho}}{\underline{\rho}} \left( \frac{4}{(K+1)^{1/4}} + \frac{4\sqrt{1-\alpha_D}}{(1-\sqrt{1-\alpha_D})(K+1)^{3/4}} \right)\bar{L}^0$$

$$+ \frac{4\bar{\rho}\sigma\sqrt{1-\alpha_D}}{(1-\sqrt{1-\alpha_D})(K+1)^{1/2}} + \frac{2\bar{\rho}\sigma}{\sqrt{n}(K+1)^{1/4}}$$

$$\leq \frac{3\left(f(X^0)-f^\star\right)}{\eta(K+1)^{1/4}} + \frac{16\sqrt{1-\alpha_D}\bar{\rho}\sigma}{(1-\sqrt{1-\alpha_D})(K+1)^{1/2}} + \frac{\eta L^0}{(K+1)^{3/4}} + \frac{8\bar{\rho}\sigma}{\sqrt{n}(K+1)^{1/4}}$$

$$+ \frac{\eta\bar{\rho}}{\underline{\rho}} \left( \frac{8}{(K+1)^{1/4}} + \frac{8\sqrt{1-\alpha_D}}{(1-\sqrt{1-\alpha_D})(K+1)^{3/4}} \right) \left( \frac{1}{n}\sum_{j=1}^{n}(L_j^1)^2\left(f^\star - f_j^\star\right) + \bar{L}^0 \right).$$

*Proof of Theorem 24.* By Lemma 2 and Jensen's inequality

$$\begin{aligned} f(X^{k+1}) &\leq f(X^k) + \sum_{i=1}^{p} 2t_i \left\|\nabla_i f(X^k) - G_i^k\right\|_{(i)\star} - \sum_{i=1}^{p} t_i \left\|\nabla_i f(X^k)\right\|_{(i)\star} \\ &\quad + \sum_{i=1}^{p} \frac{L_i^0 + L_i^1 \left\|\nabla_i f(X^k)\right\|_{(i)\star}}{2} t_i^2 \\ &\leq f(X^k) + \sum_{i=1}^{p} \left( 2t_i \left\|\nabla_i f(X^k) - M_i^k\right\|_{(i)\star} + 2t_i \left\|M_i^k - G_i^k\right\|_{(i)\star} \right) \\ &\quad - \sum_{i=1}^{p} t_i \left\|\nabla_i f(X^k)\right\|_{(i)\star} + \sum_{i=1}^{p} \left( \frac{L_i^0}{2}t_i^2 + \frac{L_i^1}{2} \left\|\nabla_i f(X^k)\right\|_{(i)\star} t_i^2 \right) \\ &\leq f(X^k) + \sum_{i=1}^{p} \left( 2\bar{\rho}_i t_i \left\|\nabla_i f(X^k) - M_i^k\right\|_2 + 2\bar{\rho}_i t_i \frac{1}{n}\sum_{j=1}^{n}\mathbb{E}\left[\left\|M_{i,j}^k - G_{i,j}^k\right\|_2\right] \right) \\ &\quad - \sum_{i=1}^{p} t_i \left\|\nabla_i f(X^k)\right\|_{(i)\star} + \sum_{i=1}^{p} \left( \frac{L_i^0}{2}t_i^2 + \frac{L_i^1}{2} \left\|\nabla_i f(X^k)\right\|_{(i)\star} t_i^2 \right). \end{aligned}$$

To simplify the notation, let $\delta^k := \mathbb{E}\left[f(X^k)-f^\star\right]$, $P_i^k := \mathbb{E}\left[\left\|\nabla_i f(X^k) - M_i^k\right\|_2\right]$, $\tilde{P}_i^k := \frac{1}{n}\sum_{j=1}^{n}\mathbb{E}\left[\left\|\nabla_i f_j(X^k) - M_{i,j}^k\right\|_2\right]$ and $\tilde{S}_i^k := \frac{1}{n}\sum_{j=1}^{n}\mathbb{E}\left[\left\|M_{i,j}^k - G_{i,j}^k\right\|_2\right]$. Then, Lemmas 7, 8, and the descent inequality above yield

$$\tilde{S}_i^{k+1} \overset{(7)}{\leq} \sqrt{1-\alpha_D}\tilde{S}_i^k + \sqrt{1-\alpha_D}\beta_i\tilde{P}_i^k + \frac{t_i\sqrt{1-\alpha_D}\beta_i}{\underline{\rho}_i} \left( \frac{1}{n}\sum_{j=1}^{n}L_{i,j}^1\mathbb{E}\left[\left\|\nabla_i f_j(X^k)\right\|_{(i)\star}\right] \right)$$

$$+ \frac{t_i\sqrt{1-\alpha_D}\beta_i\bar{L}_i^0}{\underline{\rho}_i} + \sqrt{1-\alpha_D}\beta_i\sigma_i, \tag{23}$$

$$P_i^k \quad \overset{(8)}{\leq} \quad (1-\beta_i)^k P_i^0 + \frac{t_i}{\underline{\rho}_i} \frac{1}{n} \sum_{j=1}^{n} L_{i,j}^1 \sum_{l=0}^{k-1} (1-\beta_i)^{k-l} \mathbb{E}\left[\left\|\nabla_i f_j(X^l)\right\|_{(i)\star}\right]$$

$$+ \frac{t_i \bar{L}_i^0}{\underline{\rho}_i \beta_i} + \sigma_i \sqrt{\frac{\beta_i}{n}}, \tag{24}$$

$$\tilde{P}_i^{k+1} \quad \overset{(8)}{\leq} \quad (1-\beta_i)\tilde{P}_i^k + \frac{t_i(1-\beta_i)}{\underline{\rho}_i} \frac{1}{n} \sum_{j=1}^{n} L_{i,j}^1 \mathbb{E}\left[\left\|\nabla_i f_j(X^k)\right\|_{(i)\star}\right]$$

$$+ \frac{t_i(1-\beta_i)\bar{L}_i^0}{\underline{\rho}_i} + \beta_i \sigma_i, \tag{25}$$

$$\delta^{k+1} \quad \leq \quad \delta^k - \sum_{i=1}^{p} t_i \mathbb{E}\left[\left\|\nabla_i f(X^k)\right\|_{(i)\star}\right] \tag{26}$$

$$+ \sum_{i=1}^{p} \left(2t_i \bar{\rho}_i P_i^k + 2t_i \bar{\rho}_i \tilde{S}_i^k + \frac{t_i^2 L_i^0}{2} + \frac{t_i^2 L_i^1}{2}\mathbb{E}\left[\left\|\nabla_i f(X^k)\right\|_{(i)\star}\right]\right).$$

Let $A_i, B_i > 0$ be some constants to be determined later, and define

$$\Psi^k := \delta^k + \sum_{i=1}^{p} A_i \tilde{S}_i^k + \sum_{i=1}^{p} B_i \tilde{P}_i^k.$$

Then, using (23), (25) and (26)

$$\Psi^{k+1}$$

$$= \delta^{k+1} + \sum_{i=1}^{p} A_i \tilde{S}_i^{k+1} + \sum_{i=1}^{p} B_i \tilde{P}_i^{k+1}$$

$$\leq \delta^k - \sum_{i=1}^{p} t_i \mathbb{E}\left[\left\|\nabla_i f(X^k)\right\|_{(i)\star}\right] + \sum_{i=1}^{p} \left(2t_i \bar{\rho}_i P_i^k + 2t_i \bar{\rho}_i \tilde{S}_i^k + \frac{t_i^2 L_i^0}{2} + \frac{t_i^2 L_i^1}{2}\mathbb{E}\left[\left\|\nabla_i f(X^k)\right\|_{(i)\star}\right]\right)$$

$$+ \sum_{i=1}^{p} A_i \left(\sqrt{1-\alpha_D}\tilde{S}_i^k + \sqrt{1-\alpha_D}\beta_i \tilde{P}_i^k + \frac{t_i \sqrt{1-\alpha_D}\beta_i}{\underline{\rho}_i}\left(\frac{1}{n}\sum_{j=1}^{n} L_{i,j}^1 \mathbb{E}\left[\left\|\nabla_i f_j(X^k)\right\|_{(i)\star}\right]\right)\right)$$

$$+ \sum_{i=1}^{p} A_i \left(\frac{t_i \sqrt{1-\alpha_D}\beta_i \bar{L}_i^0}{\underline{\rho}_i} + \sqrt{1-\alpha_D}\beta_i \sigma_i\right)$$

$$+ \sum_{i=1}^{p} B_i \left((1-\beta_i)\tilde{P}_i^k + \frac{t_i(1-\beta_i)}{\underline{\rho}_i}\left(\frac{1}{n}\sum_{j=1}^{n} L_{i,j}^1 \mathbb{E}\left[\left\|\nabla_i f_j(X^k)\right\|_{(i)\star}\right]\right) + \frac{t_i(1-\beta_i)\bar{L}_i^0}{\underline{\rho}_i} + \beta_i \sigma_i\right)$$

$$= \delta^k - \sum_{i=1}^{p} t_i \mathbb{E}\left[\left\|\nabla_i f(X^k)\right\|_{(i)\star}\right] + \sum_{i=1}^{p} \left(2t_i \bar{\rho}_i + A_i \sqrt{1-\alpha_D}\right)\tilde{S}_i^k$$

$$+ \sum_{i=1}^{p} \left(A_i \sqrt{1-\alpha_D}\beta_i + B_i(1-\beta_i)\right)\tilde{P}_i^k + \sum_{i=1}^{p} 2t_i \bar{\rho}_i P_i^k + \sum_{i=1}^{p} \frac{t_i^2 L_i^1}{2}\mathbb{E}\left[\left\|\nabla_i f(X^k)\right\|_{(i)\star}\right]$$

$$+ \sum_{i=1}^{p} \frac{t_i}{\underline{\rho}_i}\left(A_i \sqrt{1-\alpha_D}\beta_i + B_i(1-\beta_i)\right)\left(\frac{1}{n}\sum_{j=1}^{n} L_{i,j}^1 \mathbb{E}\left[\left\|\nabla_i f_j(X^k)\right\|_{(i)\star}\right]\right)$$

$$+ \sum_{i=1}^{p} \frac{t_i^2 L_i^0}{2} + \sum_{i=1}^{p} A_i \frac{t_i \sqrt{1-\alpha_D}\beta_i \bar{L}_i^0}{\underline{\rho}_i} + \sum_{i=1}^{p} B_i \frac{t_i(1-\beta_i)\bar{L}_i^0}{\underline{\rho}_i}$$

$$+ \sum_{i=1}^{p} A_i \sqrt{1-\alpha_D}\beta_i \sigma_i + \sum_{i=1}^{p} B_i \beta_i \sigma_i.$$

Taking $A_i = \frac{2t_i\bar{\rho}_i}{1-\sqrt{1-\alpha_D}}$ and $B_i = A_i\sqrt{1-\alpha_D} = \frac{2t_i\bar{\rho}\sqrt{1-\alpha_D}}{1-\sqrt{1-\alpha_D}}$, we obtain

$$2t_i\bar{\rho} + A_i\sqrt{1-\alpha_D} = 2t_i\bar{\rho} + \frac{2t_i\bar{\rho}}{1-\sqrt{1-\alpha_D}}\sqrt{1-\alpha_D} = A_i,$$

$$A_i\sqrt{1-\alpha_D}\beta_i + B_i(1-\beta_i) = A_i\sqrt{1-\alpha_D}\beta_i + A_i\sqrt{1-\alpha_D}(1-\beta_i) = B_i.$$

Consequently,

$$
\begin{aligned}
\Psi^{k+1} \\
\leq\quad & \delta^k - \sum_{i=1}^{p} t_i \mathbb{E}\left[\left\|\nabla_i f(X^k)\right\|_{(i)\star}\right] + \sum_{i=1}^{p} A_i \tilde{S}_i^k + \sum_{i=1}^{p} B_i \tilde{P}_i^k + \sum_{i=1}^{p} 2t_i\bar{\rho}_i P_i^k \\
& + \sum_{i=1}^{p} \frac{t_i^2 L_i^1}{2}\mathbb{E}\left[\left\|\nabla_i f(X^k)\right\|_{(i)\star}\right] + \sum_{i=1}^{p} \frac{2t_i^2\bar{\rho}_i\sqrt{1-\alpha_D}}{\underline{\rho}_i(1-\sqrt{1-\alpha_D})}\left(\frac{1}{n}\sum_{j=1}^{n} L_{i,j}^1 \mathbb{E}\left[\left\|\nabla_i f_j(X^k)\right\|_{(i)\star}\right]\right) \\
& + \sum_{i=1}^{p} \frac{t_i^2 L_i^0}{2} + \sum_{i=1}^{p} \frac{2t_i^2\bar{\rho}_i\sqrt{1-\alpha_D}\bar{L}_i^0}{\underline{\rho}_i(1-\sqrt{1-\alpha_D})} + \sum_{i=1}^{p} \frac{4t_i\bar{\rho}_i\sqrt{1-\alpha_D}\beta_i\sigma_i}{1-\sqrt{1-\alpha_D}} \\
\overset{(24)}{\leq}\quad & \delta^k - \sum_{i=1}^{p} t_i\mathbb{E}\left[\left\|\nabla_i f(X^k)\right\|_{(i)\star}\right] + \sum_{i=1}^{p} A_i \tilde{S}_i^k + \sum_{i=1}^{p} B_i \tilde{P}_i^k \\
& + \sum_{i=1}^{p} 2t_i\bar{\rho}_i\left((1-\beta_i)^k P_i^0 + \frac{t_i}{\underline{\rho}_i}\frac{1}{n}\sum_{j=1}^{n} L_{i,j}^1 \sum_{l=0}^{k-1}(1-\beta_i)^{k-l}\mathbb{E}\left[\left\|\nabla_i f_j(X^l)\right\|_{(i)\star}\right] + \frac{t_i\bar{L}_i^0}{\underline{\rho}_i\beta_i} + \sigma_i\sqrt{\frac{\beta_i}{n}}\right) \\
& + \sum_{i=1}^{p} \frac{t_i^2 L_i^1}{2}\mathbb{E}\left[\left\|\nabla_i f(X^k)\right\|_{(i)\star}\right] + \sum_{i=1}^{p} \frac{2t_i^2\bar{\rho}_i\sqrt{1-\alpha_D}}{\underline{\rho}_i(1-\sqrt{1-\alpha_D})}\left(\frac{1}{n}\sum_{j=1}^{n} L_{i,j}^1 \mathbb{E}\left[\left\|\nabla_i f_j(X^k)\right\|_{(i)\star}\right]\right) \\
& + \sum_{i=1}^{p} \frac{t_i^2 L_i^0}{2} + \sum_{i=1}^{p} \frac{2t_i^2\bar{\rho}_i\sqrt{1-\alpha_D}\bar{L}_i^0}{\underline{\rho}_i(1-\sqrt{1-\alpha_D})} + \sum_{i=1}^{p} \frac{4t_i\bar{\rho}_i\sqrt{1-\alpha_D}\beta_i\sigma_i}{1-\sqrt{1-\alpha_D}} \\
=\quad & \Psi^k - \sum_{i=1}^{p} t_i\mathbb{E}\left[\left\|\nabla_i f(X^k)\right\|_{(i)\star}\right] + \sum_{i=1}^{p} 2t_i\bar{\rho}_i(1-\beta_i)^k P_i^0 + \frac{1}{2}\sum_{i=1}^{p} t_i^2 L_i^1 \mathbb{E}\left[\left\|\nabla_i f(X^k)\right\|_{(i)\star}\right] \\
& + 2\sum_{i=1}^{p} \frac{t_i^2\bar{\rho}_i}{\underline{\rho}_i}\sum_{l=0}^{k-1}(1-\beta_i)^{k-l}\left(\frac{1}{n}\sum_{j=1}^{n} L_{i,j}^1 \mathbb{E}\left[\left\|\nabla_i f_j(X^l)\right\|_{(i)\star}\right]\right) \\
& + \frac{2\sqrt{1-\alpha_D}}{1-\sqrt{1-\alpha_D}}\sum_{i=1}^{p} \frac{t_i^2\bar{\rho}_i}{\underline{\rho}_i}\left(\frac{1}{n}\sum_{j=1}^{n} L_{i,j}^1 \mathbb{E}\left[\left\|\nabla_i f_j(X^k)\right\|_{(i)\star}\right]\right) + \sum_{i=1}^{p} \frac{t_i^2 L_i^0}{2} \\
& + \sum_{i=1}^{p} \frac{2t_i^2\bar{\rho}_i\bar{L}_i^0}{\underline{\rho}_i\beta_i} + \sum_{i=1}^{p} \frac{2t_i^2\bar{\rho}_i\sqrt{1-\alpha_D}\bar{L}_i^0}{\underline{\rho}_i(1-\sqrt{1-\alpha_D})} + \sum_{i=1}^{p} \frac{4t_i\bar{\rho}_i\sqrt{1-\alpha_D}\beta_i\sigma_i}{1-\sqrt{1-\alpha_D}} + \sum_{i=1}^{p} 2t_i\bar{\rho}_i\sigma_i\sqrt{\frac{\beta_i}{n}}. \quad (27)
\end{aligned}
$$

Let us bound the terms involving the norms of the gradients. Using Lemma 10, we get

$$
\sum_{i=1}^{p} t_i^2 L_i^1 \mathbb{E}\left[\left\|\nabla_i f(X^k)\right\|_{(i)\star}\right] \overset{(10)}{\leq} 4\max_{i\in[p]}(t_i^2(L_i^1)^2)\mathbb{E}\left[f(X^k)-f^\star\right] + \frac{\sum_{i=1}^{p}(t_i^2 L_i^1)^2 L_i^0}{\max_{i\in[p]}(t_i^2(L_i^1)^2)}
$$

$$
\leq 4\max_{i\in[p]}(t_i^2(L_i^1)^2)\delta^k + \sum_{i=1}^{p} t_i^2 L_i^0.
$$

Similarly, Lemma 11 gives

$$
\sum_{i=1}^{p} \frac{t_i^2\bar{\rho}_i}{\underline{\rho}_i}\sum_{l=0}^{k-1}(1-\beta_i)^{k-l}\frac{1}{n}\sum_{j=1}^{n} L_{i,j}^1 \mathbb{E}\left[\left\|\nabla_i f_j(X^l)\right\|_{(i)\star}\right]
$$

$$= \frac{1}{n}\sum_{j=1}^{n}\sum_{l=0}^{k-1}\sum_{i=1}^{p}\frac{t_i^2\bar{\rho}_i}{\underline{\rho}_i}(1-\beta_i)^{k-l}L_{i,j}^1\mathbb{E}\left[\left\|\nabla_i f_j(X^l)\right\|_{(i)\star}\right]$$

$$\overset{(11)}{\leq} \frac{1}{n}\sum_{j=1}^{n}\sum_{l=0}^{k-1}\left(4\max_{i\in[p]}\left(\frac{t_i^2\bar{\rho}_i}{\underline{\rho}_i}(1-\beta_i)^{k-l}(L_{i,j}^1)^2\right)\mathbb{E}\left[f_j(X^l)-f^\star\right]\right)$$

$$+\frac{1}{n}\sum_{j=1}^{n}\sum_{l=0}^{k-1}\left(4\max_{i\in[p]}\left(\frac{t_i^2\bar{\rho}_i}{\underline{\rho}_i}(1-\beta_i)^{k-l}(L_{i,j}^1)^2\right)\left(f^\star-f_j^\star\right)\right)$$

$$+\frac{1}{n}\sum_{j=1}^{n}\sum_{l=0}^{k-1}\left(\frac{\sum_{i=1}^{p}\left(\frac{t_i^2\bar{\rho}_i}{\underline{\rho}_i}(1-\beta_i)^{k-l}L_{i,j}^1\right)^2 L_{i,j}^0}{\max_{i\in[p]}\left(\frac{t_i^2\bar{\rho}_i}{\underline{\rho}_i}(1-\beta_i)^{k-l}(L_{i,j}^1)^2\right)}\right)$$

$$\leq \sum_{l=0}^{k-1}4\max_{i\in[p],j\in[n]}\left(\frac{t_i^2\bar{\rho}_i}{\underline{\rho}_i}(1-\beta_i)^{k-l}(L_{i,j}^1)^2\right)\delta^l$$

$$+\frac{1}{n}\sum_{j=1}^{n}\sum_{l=0}^{k-1}\left(4\max_{i\in[p]}\left(\frac{t_i^2\bar{\rho}_i}{\underline{\rho}_i}(1-\beta_i)^{k-l}(L_{i,j}^1)^2\right)\left(f^\star-f_j^\star\right)+\sum_{i=1}^{p}\frac{t_i^2\bar{\rho}_i}{\underline{\rho}_i}(1-\beta_i)^{k-l}L_{i,j}^0\right)$$

and

$$\sum_{i=1}^{p}\frac{t_i^2\bar{\rho}_i}{\underline{\rho}_i}\frac{1}{n}\sum_{j=1}^{n}L_{i,j}^1\mathbb{E}\left[\left\|\nabla_i f_j(X^k)\right\|_{(i)\star}\right]$$

$$= \frac{1}{n}\sum_{j=1}^{n}\sum_{i=1}^{p}\frac{t_i^2\bar{\rho}_i L_{i,j}^1}{\underline{\rho}_i}\mathbb{E}\left[\left\|\nabla_i f_j(X^k)\right\|_{(i)\star}\right]$$

$$\overset{(11)}{\leq} \frac{1}{n}\sum_{j=1}^{n}\left(4\max_{i\in[p]}\frac{t_i^2\bar{\rho}_i(L_{i,j}^1)^2}{\underline{\rho}_i}\mathbb{E}\left[f_j(X^k)-f^\star\right]+4\max_{i\in[p]}\frac{t_i^2\bar{\rho}_i(L_{i,j}^1)^2}{\underline{\rho}_i}\left(f^\star-f_j^\star\right)\right)$$

$$+\frac{1}{n}\sum_{j=1}^{n}\left(\frac{\sum_{i=1}^{p}\left(\frac{t_i^2\bar{\rho}_i}{\underline{\rho}_i}L_{i,j}^1\right)^2 L_{i,j}^0}{\max_{i\in[p]}\left(\frac{t_i^2\bar{\rho}_i}{\underline{\rho}_i}(L_{i,j}^1)^2\right)}\right)$$

$$\leq 4\max_{i\in[p],j\in[n]}\left(\frac{t_i^2\bar{\rho}_i}{\underline{\rho}_i}(L_{i,j}^1)^2\right)\delta^k+\frac{1}{n}\sum_{j=1}^{n}\left(4\max_{i\in[p]}\frac{t_i^2\bar{\rho}_i(L_{i,j}^1)^2}{\underline{\rho}_i}\left(f^\star-f_j^\star\right)+\sum_{i=1}^{p}\frac{t_i^2\bar{\rho}_i L_{i,j}^0}{\underline{\rho}_i}\right).$$

Substituting these bounds in (27), we obtain

$$\Psi^{k+1}$$

$$\leq \Psi^k-\sum_{i=1}^{p}t_i\mathbb{E}\left[\left\|\nabla_i f(X^k)\right\|_{(i)\star}\right]+\sum_{i=1}^{p}2t_i\bar{\rho}_i(1-\beta_i)^k P_i^0+2\max_{i\in[p]}(t_i^2(L_i^1)^2)\delta^k+\frac{1}{2}\sum_{i=1}^{p}t_i^2 L_i^0$$

$$+8\sum_{l=0}^{k-1}\max_{i\in[p],j\in[n]}\left(\frac{t_i^2\bar{\rho}_i}{\underline{\rho}_i}(1-\beta_i)^{k-l}(L_{i,j}^1)^2\right)\delta^l$$

$$+2\frac{1}{n}\sum_{j=1}^{n}\sum_{l=0}^{k-1}\left(4\max_{i\in[p]}\left(\frac{t_i^2\bar{\rho}_i}{\underline{\rho}_i}(1-\beta_i)^{k-l}(L_{i,j}^1)^2\right)\left(f^\star-f_j^\star\right)+\sum_{i=1}^{p}\frac{t_i^2\bar{\rho}_i}{\underline{\rho}_i}(1-\beta_i)^{k-l}L_{i,j}^0\right)$$

$$+\frac{8\sqrt{1-\alpha_D}}{1-\sqrt{1-\alpha_D}}\max_{i\in[p],j\in[n]}\left(\frac{t_i^2\bar{\rho}_i}{\underline{\rho}_i}(L_{i,j}^1)^2\right)\delta^k$$

$$+\frac{2\sqrt{1-\alpha_D}}{1-\sqrt{1-\alpha_D}}\frac{1}{n}\sum_{j=1}^{n}\left(4\max_{i\in[p]}\frac{t_i^2\bar{\rho}_i(L_{i,j}^1)^2}{\underline{\rho}_i}\left(f^\star-f_j^\star\right)+\sum_{i=1}^{p}\frac{t_i^2\bar{\rho}_i L_{i,j}^0}{\underline{\rho}_i}\right)+\sum_{i=1}^{p}\frac{t_i^2 L_i^0}{2}$$

$$+\sum_{i=1}^{p}\frac{2t_i^2\bar{\rho}_i\bar{L}_i^0}{\underline{\rho}_i\beta_i}+\sum_{i=1}^{p}\frac{2t_i^2\bar{\rho}_i\sqrt{1-\alpha_D}\bar{L}_i^0}{\underline{\rho}_i(1-\sqrt{1-\alpha_D})}+\sum_{i=1}^{p}\frac{4t_i\bar{\rho}_i\sqrt{1-\alpha_D}\beta_i\sigma_i}{1-\sqrt{1-\alpha_D}}+\sum_{i=1}^{p}2t_i\bar{\rho}_i\sigma_i\sqrt{\frac{\beta_i}{n}}.$$

Since $\delta^k \leq \Psi^k$, it follows that

$\Psi^{k+1}$

$$\leq \left(1 + 2\max_{i\in[p]}(t_i^2(L_i^1)^2) + \frac{8\sqrt{1-\alpha_D}}{1-\sqrt{1-\alpha_D}}\max_{i\in[p],j\in[n]}\left(\frac{t_i^2\bar{\rho}_i}{\underline{\rho}_i}(L_{i,j}^1)^2\right)\right)\Psi^k$$

$$-\sum_{i=1}^p t_i\mathbb{E}\left[\left\|\nabla_i f(X^k)\right\|_{(i)\star}\right] + \sum_{i=1}^p 2t_i\bar{\rho}_i(1-\beta_i)^k P_i^0$$

$$+8\max_{i\in[p],j\in[n]}\left(\frac{t_i^2\bar{\rho}_i(L_{i,j}^1)^2}{\underline{\rho}_i}\right)\sum_{l=0}^{k-1}\max_{i\in[p]}((1-\beta_i)^{k-l}\Psi^l)$$

$$+8\frac{1}{n}\sum_{j=1}^n\left(\max_{i\in[p]}\frac{t_i^2\bar{\rho}_i(L_{i,j}^1)^2}{\underline{\rho}_i}\sum_{l=0}^{k-1}\max_{i\in[p]}\left((1-\beta_i)^{k-l}\right)\left(f^\star - f_j^\star\right)\right)$$

$$+2\sum_{i=1}^p\frac{t_i^2\bar{\rho}_i\bar{L}_i^0}{\underline{\rho}_i}\sum_{l=0}^{k-1}(1-\beta_i)^{k-l} + \frac{8\sqrt{1-\alpha_D}}{1-\sqrt{1-\alpha_D}}\frac{1}{n}\sum_{j=1}^n\left(\max_{i\in[p]}\frac{t_i^2\bar{\rho}_i(L_{i,j}^1)^2}{\underline{\rho}_i}\left(f^\star - f_j^\star\right)\right)$$

$$+\frac{2\sqrt{1-\alpha_D}}{1-\sqrt{1-\alpha_D}}\sum_{i=1}^p\frac{t_i^2\bar{\rho}_i\bar{L}_i^0}{\underline{\rho}_i} + \sum_{i=1}^p t_i^2 L_i^0$$

$$+\sum_{i=1}^p\frac{2t_i^2\bar{\rho}_i\bar{L}_i^0}{\underline{\rho}_i\beta_i} + \sum_{i=1}^p\frac{2t_i^2\bar{\rho}_i\sqrt{1-\alpha_D}\bar{L}_i^0}{\underline{\rho}_i(1-\sqrt{1-\alpha_D})} + \sum_{i=1}^p\frac{4t_i\bar{\rho}_i\sqrt{1-\alpha_D}\beta_i\sigma_i}{1-\sqrt{1-\alpha_D}} + \sum_{i=1}^p 2t_i\bar{\rho}_i\sigma_i\sqrt{\frac{\beta_i}{n}}$$

$$\leq \left(1 + \underbrace{2\max_{i\in[p]}(t_i^2(L_i^1)^2) + \frac{8\sqrt{1-\alpha_D}}{1-\sqrt{1-\alpha_D}}\max_{i\in[p],j\in[n]}\left(\frac{t_i^2\bar{\rho}_i(L_{i,j}^1)^2}{\underline{\rho}_i}\right)}_{:=C_1}\right)\Psi^k$$

$$-\sum_{i=1}^p t_i\mathbb{E}\left[\left\|\nabla_i f(X^k)\right\|_{(i)\star}\right] + \sum_{i=1}^p 2t_i\bar{\rho}_i(1-\beta_i)^k P_i^0$$

$$+8\underbrace{\max_{i\in[p],j\in[n]}\left(\frac{t_i^2\bar{\rho}_i(L_{i,j}^1)^2}{\underline{\rho}_i}\right)}_{:=C_2}\sum_{l=0}^{k-1}((1-\beta_{\min})^{k-l}\Psi^l)$$

$$+8\underbrace{\left(\frac{1}{\beta_{\min}} + \frac{\sqrt{1-\alpha_D}}{1-\sqrt{1-\alpha_D}}\right)\frac{1}{n}\sum_{j=1}^n\left(\max_{i\in[p]}\frac{t_i^2\bar{\rho}_i(L_{i,j}^1)^2}{\underline{\rho}_i}\left(f^\star - f_j^\star\right)\right)}_{:=C_3}$$

$$+\sum_{i=1}^p t_i^2\underbrace{\left(L_i^0 + \frac{4\bar{\rho}_i\bar{L}_i^0}{\underline{\rho}_i\beta_i} + \frac{4\bar{\rho}_i\sqrt{1-\alpha_D}\bar{L}_i^0}{\underline{\rho}_i(1-\sqrt{1-\alpha_D})}\right)}_{:=C_{4,i}} + \sum_{i=1}^p t_i\underbrace{\bar{\rho}_i\sigma_i\left(\frac{4\sqrt{1-\alpha_D}\beta_i}{1-\sqrt{1-\alpha_D}} + 2\sqrt{\frac{\beta_i}{n}}\right)}_{:=C_{5,i}}$$

$$= (1+C_1)\Psi^k - \sum_{i=1}^p t_i\mathbb{E}\left[\left\|\nabla_i f(X^k)\right\|_{(i)\star}\right] + \sum_{i=1}^p 2t_i\bar{\rho}_i(1-\beta_i)^k P_i^0$$

$$+C_2\sum_{l=0}^{k-1}((1-\beta_{\min})^{k-l}\Psi^l) + C_3 + \sum_{i=1}^p t_i^2 C_{4,i} + \sum_{i=1}^p t_i C_{5,i}.$$

Now, define a weighting sequence $w^k := \frac{w^{k-1}}{1+C_1+\frac{C_2}{\beta_{\min}}}$, where $w^{-1} = 1$. Then, multiplying the above inequality by $w^k$ and summing over the first $K+1$ iterations, we obtain

$$\sum_{k=0}^K w^k\Psi^{k+1}$$

$$
\begin{aligned}
\leq \quad & \sum_{k=0}^{K} w^k \left(1+C_1\right) \Psi^k + \sum_{k=0}^{K} w^k C_2 \sum_{l=0}^{k-1} \left((1-\beta_{\min})^{k-l}\Psi^l\right) - \sum_{k=0}^{K} w^k \sum_{i=1}^{p} t_i \mathbb{E}\left[\left\|\nabla_i f(X^k)\right\|_{(i)\star}\right] \\
& + \sum_{k=0}^{K} w^k \sum_{i=1}^{p} 2 t_i \bar{\rho}_i (1-\beta_i)^k P_i^0 + \sum_{k=0}^{K} w^k C_3 + \sum_{k=0}^{K} w^k \sum_{i=1}^{p} t_i^2 C_{4,i} + \sum_{k=0}^{K} w^k \sum_{i=1}^{p} t_i C_{5,i} \\
= \quad & (1+C_1) \sum_{k=0}^{K} w^k \Psi^k + C_2 \sum_{k=0}^{K} w^k \sum_{l=0}^{k-1}\left((1-\beta_{\min})^{k-l}\Psi^l\right) - \sum_{k=0}^{K} w^k \sum_{i=1}^{p} t_i \mathbb{E}\left[\left\|\nabla_i f(X^k)\right\|_{(i)\star}\right] \\
& + \sum_{k=0}^{K} w^k \sum_{i=1}^{p} 2 t_i \bar{\rho}_i (1-\beta_i)^k P_i^0 + W^K C_3 + W^K \sum_{i=1}^{p} t_i^2 C_{4,i} + W^K \sum_{i=1}^{p} t_i C_{5,i}.
\end{aligned}
$$

where $W^K := \sum_{k=0}^{K} w^k$. Since, by definition, $w^k \leq w^{k-1} \leq w^{-1} = 1$, we have

$$
\sum_{k=0}^{K} w^k \Psi^{k+1}
$$

$$
\begin{aligned}
\leq \quad & (1+C_1) \sum_{k=0}^{K} w^k \Psi^k + C_2 \sum_{k=0}^{K} \sum_{l=0}^{k-1} (w^l (1-\beta_{\min})^{k-l}\Psi^l) - \sum_{k=0}^{K} w^k \sum_{i=1}^{p} t_i \mathbb{E}\left[\left\|\nabla_i f(X^k)\right\|_{(i)\star}\right] \\
& + \sum_{k=0}^{K} \sum_{i=1}^{p} 2 t_i \bar{\rho}_i (1-\beta_i)^k P_i^0 + W^K C_3 + W^K \sum_{i=1}^{p} t_i^2 C_{4,i} + W^K \sum_{i=1}^{p} t_i C_{5,i} \\
\leq \quad & (1+C_1) \sum_{k=0}^{K} w^k \Psi^k + C_2 \sum_{l=0}^{\infty} (1-\beta_{\min})^l \sum_{k=0}^{K} w^k \Psi^k - \sum_{k=0}^{K} w^k \sum_{i=1}^{p} t_i \mathbb{E}\left[\left\|\nabla_i f(X^k)\right\|_{(i)\star}\right] \\
& + 2 \sum_{i=1}^{p} \frac{t_i \bar{\rho}_i}{\beta_i} P_i^0 + W^K C_3 + W^K \sum_{i=1}^{p} t_i^2 C_{4,i} + W^K \sum_{i=1}^{p} t_i C_{5,i} \\
= \quad & \left(1+C_1+\frac{C_2}{\beta_{\min}}\right) \sum_{k=0}^{K} w^k \Psi^k - \sum_{k=0}^{K} w^k \sum_{i=1}^{p} t_i \mathbb{E}\left[\left\|\nabla_i f(X^k)\right\|_{(i)\star}\right] \\
& + 2 \sum_{i=1}^{p} \frac{t_i \bar{\rho}_i}{\beta_i} P_i^0 + W^K C_3 + W^K \sum_{i=1}^{p} t_i^2 C_{4,i} + W^K \sum_{i=1}^{p} t_i C_{5,i} \\
= \quad & \sum_{k=0}^{K} w^{k-1} \Psi^k - \sum_{k=0}^{K} w^k \sum_{i=1}^{p} t_i \mathbb{E}\left[\left\|\nabla_i f(X^k)\right\|_{(i)\star}\right] + 2 \sum_{i=1}^{p} \frac{t_i \bar{\rho}_i}{\beta_i} P_i^0 \\
& + W^K C_3 + W^K \sum_{i=1}^{p} t_i^2 C_{4,i} + W^K \sum_{i=1}^{p} t_i C_{5,i}.
\end{aligned}
$$

Rearranging the terms and dividing by $W^K$ gives

$$
\begin{aligned}
& \min_{k=0,\ldots,K} \sum_{i=1}^{p} t_i \mathbb{E}\left[\left\|\nabla_i f(X^k)\right\|_{(i)\star}\right] \\
\leq \quad & \sum_{k=0}^{K} \sum_{i=1}^{p} \frac{w^k}{W^K} t_i \mathbb{E}\left[\left\|\nabla_i f(X^k)\right\|_{(i)\star}\right] \\
\leq \quad & \frac{1}{W^K} \sum_{k=0}^{K} \left(w^{k-1}\Psi^k - w^k \Psi^{k+1}\right) + \frac{2}{W^K} \sum_{i=1}^{p} \frac{t_i \bar{\rho}_i}{\beta_i} P_i^0 + C_3 + \sum_{i=1}^{p} t_i^2 C_{4,i} + \sum_{i=1}^{p} t_i C_{5,i} \\
\leq \quad & \frac{\Psi^0}{W^K} + \frac{2}{W^K} \sum_{i=1}^{p} \frac{t_i \bar{\rho}_i}{\beta_i} P_i^0 + C_3 + \sum_{i=1}^{p} t_i^2 C_{4,i} + \sum_{i=1}^{p} t_i C_{5,i}.
\end{aligned}
$$

Now, note that

$$W^K = \sum_{k=0}^{K} w^k \geq (K+1)w^K = \frac{(K+1)w^{-1}}{(1+C_1+\frac{C_2}{\beta_{\min}})^{K+1}} \geq \frac{K+1}{\exp\left((K+1)(C_1+\frac{C_2}{\beta_{\min}})\right)}.$$

Taking $t_i = \frac{\eta_i}{(K+1)^{3/4}}$, where $\eta_i^2 \leq \min\left\{\frac{(K+1)^{1/2}}{6(L_i^1)^2}, \frac{(1-\sqrt{1-\alpha_D})\underline{\rho}_i(K+1)^{1/2}}{24\sqrt{1-\alpha_D}\bar{\rho}_i(L_{i,\max}^1)^2}, \frac{\beta_{\min}\underline{\rho}_i(K+1)^{1/2}}{24\bar{\rho}_i(L_{i,\max}^1)^2}, 1\right\}$ to ensure that

$$2(K+1)\max_{i\in[p]}(t_i^2(L_i^1)^2) \leq \frac{1}{3},$$

$$(K+1)\frac{8\sqrt{1-\alpha_D}}{1-\sqrt{1-\alpha_D}}\max_{i\in[p],j\in[n]}\left(\frac{t_i^2\bar{\rho}_i(L_{i,j}^1)^2}{\underline{\rho}_i}\right) \leq \frac{1}{3},$$

$$(K+1)\frac{8}{\beta_{\min}}\max_{i\in[p],j\in[n]}\left(\frac{t_i^2\bar{\rho}_i(L_{i,j}^1)^2}{\underline{\rho}_i}\right) \leq \frac{1}{3},$$

we have $(K+1)(C_1+\frac{C_2}{\beta_{\min}}) \leq 1$, and so $W^K \geq \frac{K+1}{\exp(1)} \geq \frac{K+1}{3}$. Therefore,

$$\min_{k=0,\ldots,K}\sum_{i=1}^{p} t_i\mathbb{E}\left[\left\|\nabla_i f(X^k)\right\|_{(i)\star}\right]$$

$$\leq \frac{3\Psi^0}{K+1} + \frac{6}{K+1}\sum_{i=1}^{p}\frac{t_i\bar{\rho}_i}{\beta_i}P_i^0 + C_3 + \sum_{i=1}^{p}t_i^2 C_{4,i} + \sum_{i=1}^{p}t_i C_{5,i}$$

$$= \frac{3\Psi^0}{K+1} + \frac{6}{K+1}\sum_{i=1}^{p}\frac{\eta_i\bar{\rho}_i}{\beta_i(K+1)^{3/4}}P_i^0$$

$$+ \frac{8}{(K+1)^{3/2}}\left(\frac{1}{\beta_{\min}} + \frac{\sqrt{1-\alpha_D}}{1-\sqrt{1-\alpha_D}}\right)\frac{1}{n}\sum_{j=1}^{n}\left(\max_{i\in[p]}\frac{\eta_i^2\bar{\rho}_i(L_{i,j}^1)^2}{\underline{\rho}_i}\left(f^\star - f_j^\star\right)\right)$$

$$+ \sum_{i=1}^{p}\frac{\eta_i^2}{(K+1)^{3/2}}\left(L_i^0 + \frac{4\bar{\rho}_i\bar{L}_i^0}{\underline{\rho}_i\beta_i} + \frac{4\bar{\rho}_i\sqrt{1-\alpha_D}\bar{L}_i^0}{\underline{\rho}_i(1-\sqrt{1-\alpha_D})}\right)$$

$$+ \sum_{i=1}^{p}\frac{\eta_i}{(K+1)^{3/4}}\bar{\rho}_i\sigma_i\left(\frac{4\sqrt{1-\alpha_D}\beta_i}{1-\sqrt{1-\alpha_D}} + 2\sqrt{\frac{\beta_i}{n}}\right).$$

Lastly, dividing by $\frac{1}{p}\sum_{l=1}^{p}t_l = \frac{1}{(K+1)^{3/4}}\frac{1}{p}\sum_{l=1}^{p}\eta_l$ gives

$$\min_{k=0,\ldots,K}\sum_{i=1}^{p}\frac{\eta_i}{\frac{1}{p}\sum_{l=1}^{p}\eta_l}\mathbb{E}\left[\left\|\nabla_i f(X^k)\right\|_{(i)\star}\right]$$

$$\leq \frac{3\Psi^0}{(K+1)^{1/4}\frac{1}{p}\sum_{l=1}^{p}\eta_l} + \frac{6}{K+1}\sum_{i=1}^{p}\frac{\eta_i}{\frac{1}{p}\sum_{l=1}^{p}\eta_l}\frac{\bar{\rho}_i}{\beta_i}P_i^0$$

$$+ \frac{8}{(K+1)^{3/4}}\left(\frac{1}{\beta_{\min}} + \frac{\sqrt{1-\alpha_D}}{1-\sqrt{1-\alpha_D}}\right)\frac{1}{n}\sum_{j=1}^{n}\frac{\max_{i\in[p]}\frac{\eta_i^2\bar{\rho}_i(L_{i,j}^1)^2}{\underline{\rho}_i}}{\frac{1}{p}\sum_{l=1}^{p}\eta_l}\left(f^\star - f_j^\star\right)$$

$$+ \sum_{i=1}^{p}\frac{\eta_i^2}{(K+1)^{3/4}\frac{1}{p}\sum_{l=1}^{p}\eta_l}\left(L_i^0 + \frac{4\bar{\rho}_i\bar{L}_i^0}{\underline{\rho}_i\beta_i} + \frac{4\bar{\rho}_i\sqrt{1-\alpha_D}\bar{L}_i^0}{\underline{\rho}_i(1-\sqrt{1-\alpha_D})}\right)$$

$$+ \sum_{i=1}^{p}\frac{\eta_i\bar{\rho}_i\sigma_i}{\frac{1}{p}\sum_{l=1}^{p}\eta_l}\left(\frac{4\sqrt{1-\alpha_D}\beta_i}{1-\sqrt{1-\alpha_D}} + 2\sqrt{\frac{\beta_i}{n}}\right)$$

$$= \frac{3\Psi^0}{(K+1)^{1/4}\frac{1}{p}\sum_{l=1}^{p}\eta_l} + \frac{6}{(K+1)^{1/2}}\sum_{i=1}^{p}\frac{\eta_i\bar{\rho}_i}{\frac{1}{p}\sum_{l=1}^{p}\eta_l}P_i^0$$

$$+ \left( \frac{8}{(K+1)^{1/4}} + \frac{8\sqrt{1-\alpha_D}}{(1-\sqrt{1-\alpha_D})(K+1)^{3/4}} \right) \frac{1}{n} \sum_{j=1}^{n} \frac{\max_{i \in [p]} \frac{\eta_i^2 \bar{\rho}_i (L_{i,j}^1)^2}{\underline{\rho}_i}}{\frac{1}{p} \sum_{l=1}^{p} \eta_l} \left( f^\star - f_j^\star \right)$$

$$+ \sum_{i=1}^{p} \frac{\eta_i^2}{\frac{1}{p} \sum_{l=1}^{p} \eta_l} \left( \frac{L_i^0}{(K+1)^{3/4}} + \frac{4\bar{\rho}_i \bar{L}_i^0}{\underline{\rho}_i (K+1)^{1/4}} + \frac{4\bar{\rho}_i \sqrt{1-\alpha_D} \bar{L}_i^0}{\underline{\rho}_i (1-\sqrt{1-\alpha_D})(K+1)^{3/4}} \right)$$

$$+ \sum_{i=1}^{p} \frac{\eta_i \bar{\rho}_i \sigma_i}{\frac{1}{p} \sum_{l=1}^{p} \eta_l} \left( \frac{4\sqrt{1-\alpha_D}}{(1-\sqrt{1-\alpha_D})(K+1)^{1/2}} + \frac{2}{\sqrt{n}(K+1)^{1/4}} \right),$$

where in the last equality we set $\beta_i = \frac{1}{(K+1)^{1/2}}$. $\qquad\qquad\square$

*Proof of Corollary 3.* Substituting the initialization, we have

$$P_i^0 \;\; := \;\; \mathbb{E}\left[ \left\| \nabla_i f(X^0) - M_i^0 \right\|_2 \right] = \mathbb{E}\left[ \left\| \frac{1}{n} \sum_{j=1}^{n} \left( \nabla_i f_j(X^0) - \nabla_i f_j(X^0; \xi_j^0) \right) \right\|_2 \right]$$

$$\leq \;\; \sqrt{ \mathbb{E}\left[ \left\| \frac{1}{n} \sum_{j=1}^{n} \left( \nabla_i f_j(X^0) - \nabla_i f_j(X^0; \xi_j^0) \right) \right\|_2^2 \right] } \;\; \overset{(10)}{\leq} \;\; \frac{\sigma_i}{\sqrt{n}},$$

$$\tilde{P}_i^0 \;\; := \;\; \frac{1}{n} \sum_{j=1}^{n} \mathbb{E}\left[ \left\| \nabla_i f_j(X^0) - M_{i,j}^0 \right\|_2 \right] = \frac{1}{n} \sum_{j=1}^{n} \mathbb{E}\left[ \left\| \nabla_i f_j(X^0) - \nabla_i f_j(X^0; \xi_j^0) \right\|_2 \right] \leq \sigma_i,$$

$$\tilde{S}_i^0 \;\; := \;\; \frac{1}{n} \sum_{j=1}^{n} \mathbb{E}\left[ \left\| M_{i,j}^0 - G_{i,j}^0 \right\|_2 \right] = \frac{1}{n} \sum_{j=1}^{n} \mathbb{E}\left[ \left\| \nabla_i f_j(X^0; \xi_j^0) - \mathcal{C}_{i,j}^0(\nabla_i f_j(X^0; \xi_j^0)) \right\|_2 \right]$$

$$\overset{(1)}{\leq} \;\; \sqrt{1-\alpha_D} \frac{1}{n} \sum_{j=1}^{n} \mathbb{E}\left[ \left\| \nabla_i f_j(X^0; \xi_j^0) \right\|_2 \right]$$

$$\leq \;\; \sqrt{1-\alpha_D} \frac{1}{n} \sum_{j=1}^{n} \mathbb{E}\left[ \left\| \nabla_i f_j(X^0; \xi_j^0) - \nabla_i f_j(X^0) \right\|_2 \right] \overset{(10)}{\leq} \sqrt{1-\alpha_D} \sigma_i,$$

and hence

$$\Psi^0 \;\; := \;\; f(X^0) - f^\star + \sum_{i=1}^{p} \frac{2t_i \bar{\rho}_i}{1-\sqrt{1-\alpha_D}} \frac{1}{n} \sum_{j=1}^{n} \mathbb{E}\left[ \left\| M_{i,j}^0 - G_{i,j}^0 \right\|_2 \right]$$

$$+ \sum_{i=1}^{p} \frac{2t_i \bar{\rho}_i \sqrt{1-\alpha_D}}{1-\sqrt{1-\alpha_D}} \frac{1}{n} \sum_{j=1}^{n} \mathbb{E}\left[ \left\| \nabla_i f_j(X^0) - M_{i,j}^0 \right\|_2 \right]$$

$$\leq \;\; f(X^0) - f^\star + \sum_{i=1}^{p} \frac{2t_i \bar{\rho}_i}{1-\sqrt{1-\alpha_D}} \sqrt{1-\alpha_D} \sigma_i + \sum_{i=1}^{p} \frac{2t_i \bar{\rho}_i \sqrt{1-\alpha_D}}{1-\sqrt{1-\alpha_D}} \sigma_i$$

$$= \;\; f(X^0) - f^\star + \sum_{i=1}^{p} \frac{4\sqrt{1-\alpha_D} t_i \bar{\rho}_i \sigma_i}{1-\sqrt{1-\alpha_D}}.$$

Substituting this in the rate, we get

$$\min_{k=0,\dots,K} \sum_{i=1}^{p} \frac{\eta_i}{\frac{1}{p} \sum_{l=1}^{p} \eta_l} \mathbb{E}\left[ \left\| \nabla_i f(X^k) \right\|_{(i)\star} \right]$$

$$\leq \;\; \frac{3}{(K+1)^{1/4} \frac{1}{p} \sum_{l=1}^{p} \eta_l} \left( f(X^0) - f^\star + \sum_{i=1}^{p} \frac{4\sqrt{1-\alpha_D} \eta_i \bar{\rho}_i \sigma_i}{(K+1)^{3/4}(1-\sqrt{1-\alpha_D})} \right)$$

$$+ \frac{6}{(K+1)^{1/2}} \sum_{i=1}^{p} \frac{\bar{\rho}_i \eta_i \sigma_i}{\sqrt{n} \frac{1}{p} \sum_{l=1}^{p} \eta_l}$$

$$+ \left( \frac{8}{(K+1)^{1/4}} + \frac{8\sqrt{1-\alpha_D}}{(K+1)^{3/4}(1-\sqrt{1-\alpha_D})} \right) \frac{1}{n} \sum_{j=1}^{n} \frac{\max_{i \in [p]} \eta_i^2 \frac{\bar{\rho}_i}{\underline{\rho}_i} (L_{i,j}^1)^2}{\frac{1}{p} \sum_{l=1}^{p} \eta_l} \left( f^\star - f_j^\star \right)$$

$$+ \sum_{i=1}^{p} \frac{\eta_i^2}{\frac{1}{p} \sum_{l=1}^{p} \eta_l} \left( \frac{L_i^0}{(K+1)^{3/4}} + \frac{4\bar{\rho}_i \bar{L}_i^0}{\underline{\rho}_i (K+1)^{1/4}} + \frac{4\bar{\rho}_i \sqrt{1-\alpha_D} \bar{L}_i^0}{\underline{\rho}_i (K+1)^{3/4}(1-\sqrt{1-\alpha_D})} \right)$$

$$+ \sum_{i=1}^{p} \frac{\eta_i \bar{\rho}_i \sigma_i}{\frac{1}{p} \sum_{l=1}^{p} \eta_l} \left( \frac{4\sqrt{1-\alpha_D}}{(K+1)^{1/2}(1-\sqrt{1-\alpha_D})} + \frac{2}{\sqrt{n}(K+1)^{1/4}} \right).$$

$\square$

## F    USEFUL FACTS AND LEMMAS

For all $X, Y \in \mathcal{S}$, $Z \in \mathcal{S}^\star$ (where $\mathcal{S}^\star$ is the dual space of $\mathcal{S}$), $t > 0$ and $\alpha \in (0, 1]$, we have:

$$\|X + Y\|^2 \leq (1 + t) \|X\|^2 + (1 + t^{-1}) \|Y\|^2, \tag{28}$$

$$\langle X, Z \rangle \leq \frac{\|X\|^2}{2t} + \frac{t \|Z\|_\star^2}{2}, \tag{29}$$

$$(1 - \alpha) \left(1 + \frac{\alpha}{2}\right) \leq 1 - \frac{\alpha}{2}, \tag{30}$$

$$(1 - \alpha) \left(1 + \frac{2}{\alpha}\right) \leq \frac{2}{\alpha}, \tag{31}$$

$$\langle G, \mathrm{LMO}_{\mathcal{B}(X,t)}(G) \rangle = -t \|G\|_\star \tag{32}$$

$$\langle X, X^\sharp \rangle = \left\|X^\sharp\right\|^2, \tag{33}$$

$$\|X\|_\star = \left\|X^\sharp\right\|. \tag{34}$$

**Lemma 12** (Riabinin et al. (2025b), Lemma 3). *Suppose that* $x_1, \ldots, x_p, y_1, \ldots, y_p \in \mathbb{R}$, $\max_{i \in [p]} |x_i| > 0$ *and* $z_1, \ldots, z_p > 0$. *Then*

$$\sum_{i=1}^p \frac{y_i^2}{z_i} \geq \frac{\left(\sum_{i=1}^p x_i y_i\right)^2}{\sum_{i=1}^p z_i x_i^2}.$$

**Lemma 13** (Variance decomposition). *For any random vector* $X \in \mathcal{S}$ *and any non-random* $c \in \mathcal{S}$, *we have*

$$\mathbb{E}\left[\|X - c\|_2^2\right] = \mathbb{E}\left[\|X - \mathbb{E}[X]\|_2^2\right] + \|\mathbb{E}[X] - c\|_2^2.$$

**Lemma 14** (Riabinin et al. (2025b), Lemma 1). *Let Assumption 8 hold. Then, for any* $X, Y \in \mathcal{S}$,

$$|f(Y) - f(X) - \langle \nabla f(X), Y - X \rangle| \leq \sum_{i=1}^p \frac{L_i^0 + L_i^1 \|\nabla_i f(X)\|_{(i)\star}}{2} \|X_i - Y_i\|_{(i)}^2.$$

**Lemma 15.** *Let* $\{A^k\}_{k \geq 0}$, $\{B_i^k\}_{k \geq 0}$, $i \in [p]$ *be non-negative sequences such that*

$$A^{k+1} \leq (1 + a_1) A^k - \sum_{i=1}^p B_i^k + a_2,$$

*where* $a_1, a_2 \geq 0$. *Then*

$$\min_{k=0,\ldots,K} \sum_{i=1}^p B_i^k \leq \frac{\exp(a_1(K+1))}{(K+1)} A^0 + a_2.$$

*Proof.* Let us define a weighting sequence $w^k := \frac{w^{k-1}}{1+a_1}$, where $w^{-1} = 1$. Then

$$w^k A^{k+1} \leq w^k (1 + a_1) A^k - w^k \sum_{i=1}^p B_i^k + w^k a_2 = w^{k-1} A^k - w^k \sum_{i=1}^p B_i^k + w^k a_2,$$

and hence

$$\begin{aligned}
\min_{k=0,\ldots,K} \sum_{i=1}^p B_i^k &\leq \frac{1}{\sum_{k=0}^K w^k} \sum_{k=0}^K w^k \sum_{i=1}^p B_i^k \\
&\leq \frac{1}{\sum_{k=0}^K w^k} \sum_{k=0}^K \left(w^{k-1} A^k - w^k A^{k+1}\right) + \frac{1}{\sum_{k=0}^K w^k} \sum_{k=0}^K w^k a_2 \\
&= \frac{1}{\sum_{k=0}^K w^k} \left(w^{-1} A^0 - w^K A^{K+1}\right) + a_2.
\end{aligned}$$

Using the fact that $w^{-1} = 1$ and $\sum_{k=0}^K w^k = \sum_{k=0}^K \frac{1}{(1+a_1)^{k+1}} \geq \frac{K+1}{(1+a_1)^{K+1}}$, we get

$$\min_{k=0,\ldots,K} \sum_{i=1}^p B_i^k \leq \frac{(1+a_1)^{K+1}}{(K+1)} \left(A^0 - w^K A^{K+1}\right) + a_2 \leq \frac{\exp(a_1(K+1))}{(K+1)} A^0 + a_2,$$

which finishes the proof. $\qquad\square$

# G    EXPERIMENTS

This section provides additional experimental results and setup details complementing Section 5.

## G.1    SETUP DETAILS

Tables 3 to 5 summarize the model and optimizer hyperparameters. The *scale* parameters (Hidden/-Head Scale) in Table 5 specify the LMO trust-region radius as

$$radius = scale \times learning\ rate,$$

following Pethick et al. (2025c); Riabinin et al. (2025b).

Table 3: `NanoGPT`-124M model configuration.

| Hyperparameter | Value |
| --- | --- |
| Total Parameters | 124M |
| Vocabulary Size | 50,304 |
| Number of Transformer Layers | 12 |
| Attention Heads | 6 |
| Hidden Size | 768 |
| FFN Hidden Size | 3,072 |
| Positional Embedding | RoPE (Su et al., 2024) |
| Activation Function | Squared ReLU (So et al., 2021) |
| Normalization | RMSNorm (Zhang & Sennrich, 2019) |
| Bias Parameters | None |

Table 4: `MediumGPT`-335M model configuration.

| Hyperparameter | Value |
| --- | --- |
| Total Parameters | 335M |
| Vocabulary Size | 50,304 |
| Number of Transformer Layers | 24 |
| Attention Heads | 16 |
| Hidden Size | 1024 |
| FFN Hidden Size | 4096 |
| Positional Embedding | RoPE (Su et al., 2024) |
| Activation Function | Squared ReLU (So et al., 2021) |
| Normalization | RMSNorm (Zhang & Sennrich, 2019) |
| Bias Parameters | None |

## G.2    TOP$K$ COMPRESSION DETAILS

Top$K$ compressor requires transmitting both the selected values and their corresponding indices to reconstruct the original tensors. At high compression levels, this introduces significant communication overhead, especially in compositional schemes such as Top$K$ combined with the Natural compressor, where the cost of transmitting indices can even exceed that of the quantized values. To illustrate this effect, we analyze the largest parameter matrices in the `NanoGPT` model: the token embedding layer and the classification head, each of size $50,304 \times 768$. Representing an index for any element in these matrices requires $\log_2(50{,}304 \cdot 768) < 26$ bits. We use this calculation when visualizing communication costs.

## G.3    LEARNING RATE ABLATION

To ensure a fair and robust comparison, we perform a learning rate hyperparameter sweep for each compression configuration, as detailed in Figure 3. For every method, the search space is initialized

Table 5: Optimizer configuration.

| Hyperparameter | Value |
|---|---|
| Sequence Length | 1024 |
| Batch Size | 256 |
| Optimizer | EF21-Muon |
| Weight Decay | 0 |
| Hidden Layer Norm | Spectral norm |
| Hidden Layer Scale | 50 |
| Newton–Schulz Iterations | 5 |
| Embedding and Head Layers Norm | $\ell_\infty$ norm |
| Embedding and Head Layers Scale | 3000 |
| Initial Learning Rate | For non-compressed: $3.6 \times 10^{-4}$ |
| Learning Rate Schedule | Constant followed by linear decreasing |
| Learning Rate Constant Phase Length | 40% of tokens |
| Momentum | 0.9 |

at the optimal learning rate of the uncompressed baseline (taken from the Gluon repository (Riabinin et al., 2025a)) and spans downward by up to an order of magnitude. We consistently observe that more aggressive compression schemes require a smaller learning rate for stable convergence.

This tuning protocol is applied uniformly across all experiments for models trained with 2.5B (Section G.5) and 5B token budgets.

### G.4 COMPRESSION LEVEL ABLATION

This section presents an ablation study on the compression ratio, governed by the parameter $K$. Figures 4 and 5 illustrate the convergence curves for various compression configurations, each trained with its optimal learning rate (see Section G.3). Figure 6 summarizes the final loss as a function of $K$.

Our results show that for $\text{Top}K$ and $\text{Rand}K$ compressors, an aggressive compression ratio of $K = 5\%$ quite severely impairs convergence (see Figure 6), while configurations with $K \geq 10\%$ achieve satisfactory loss reduction. When these compressors are composed with the Natural compressor, convergence degradation is more pronounced for $K = 10\%$ than for the less aggressive $K = 15\%$ setup.

We also examine a more challenging loss threshold of 3.28 (Figure 7). The communication cost improvement at this threshold is even more pronounced than for 3.31 (Figure 1), but this comes at a cost: only a subset of compressors can reach the threshold within the 5B token budget.

### G.5 2.5B TOKENS EXPERIMENT

In Section 5, we report runs with a 5B token budget ($> 40\times$ model size). Testing convergence over a large number of tokens is important, as the limitations of compressors relative to the baseline become more pronounced after many steps. At the same time, evaluating compressed runs with a smaller token budget is useful for cases with limited resources. We provide a learning rate ablation in Figure 3, a summarized comparison in Figure 6, and convergence trajectories for the 2.5B-token setup in Figures 8 and 9.

### G.6 MEDIUMGPT EXPERIMENT

To assess whether the patterns observed on NanoGPT scale to larger models, we conduct experiments on MediumGPT (335M parameters) (Karpathy, 2023) with 2.5B token budget. The model configuration is provided in Table 4. We compare the uncompressed baseline to EF21-Muon with the Natural compressor and evaluate convergence in terms of both tokens and bytes communicated.

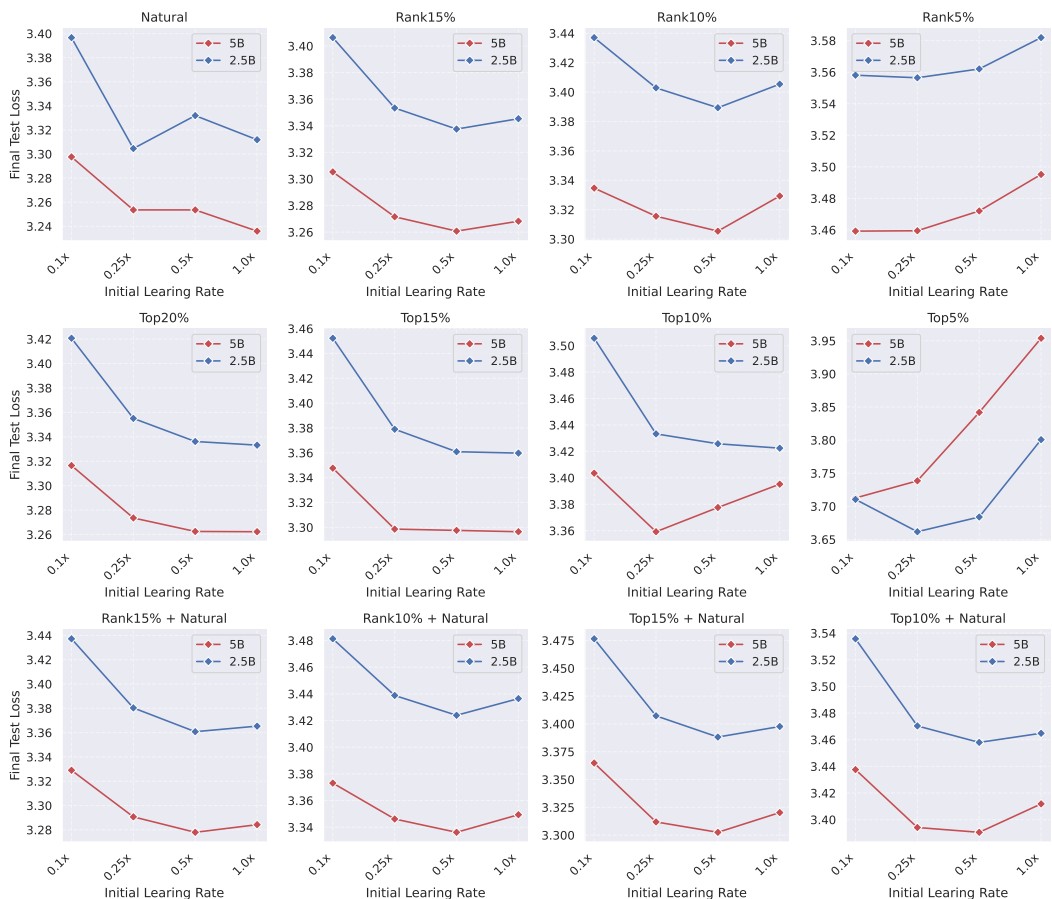

Figure 3: **Learning rate ablation.** The grid spans from the optimal learning rate of the non-compressed baseline, $3.6 \times 10^{-4}$ (denoted as $1.0\times$), down to $0.1\times$. Red curves correspond to experiments processing 5B tokens (Section 5), while blue curves correspond to 2.5B tokens (Section G.5).

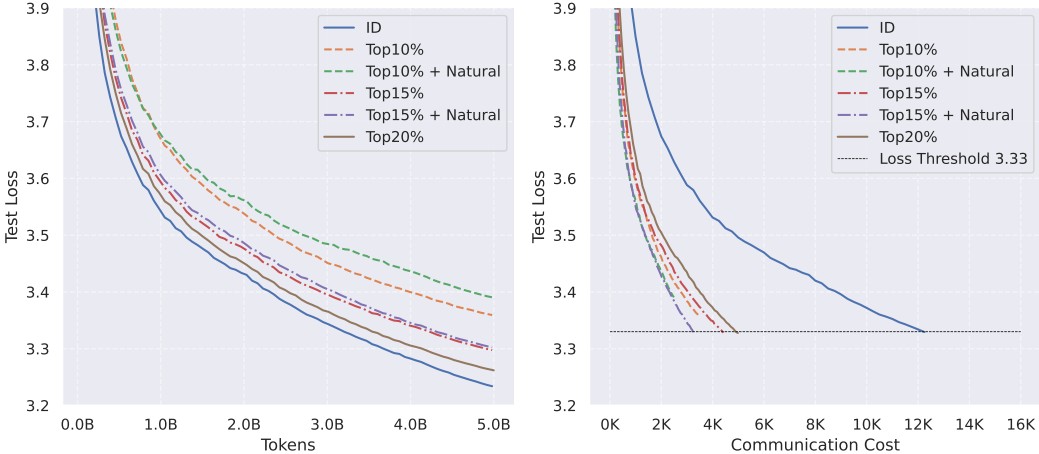

Figure 4: Left: **Test loss vs. # of tokens processed.** Right: **Test loss vs. # of bytes sent to the server from each worker** normalized by model size to reach test loss 3.33. Top$X\%$ = Top$K$ compressor with sparsification level $X\%$; ID = no compression.

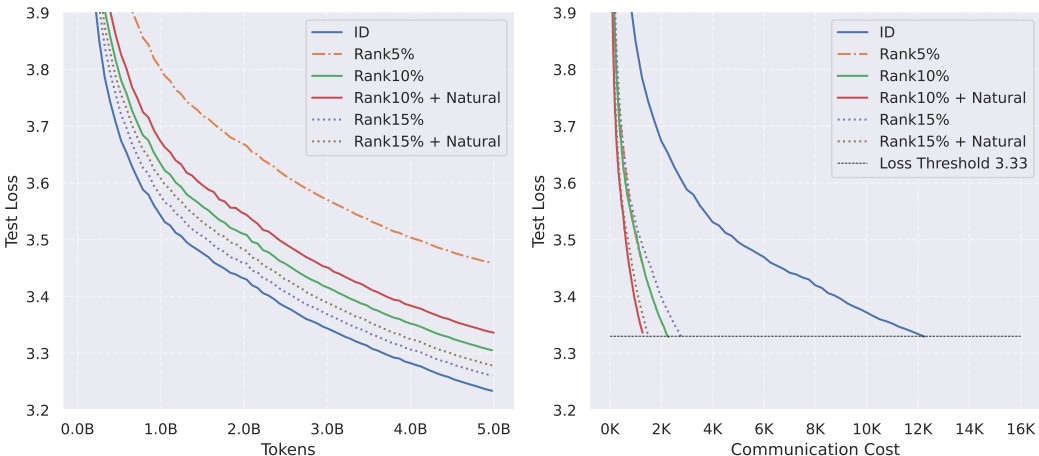

Figure 5: Left: **Test loss vs. # of tokens processed.** Right: **Test loss vs. # of bytes sent to the server from each worker** normalized by model size to reach test loss 3.33. Rank$X\%$ = Rank$K$ compressor with sparsification level $X\%$; ID = no compression.

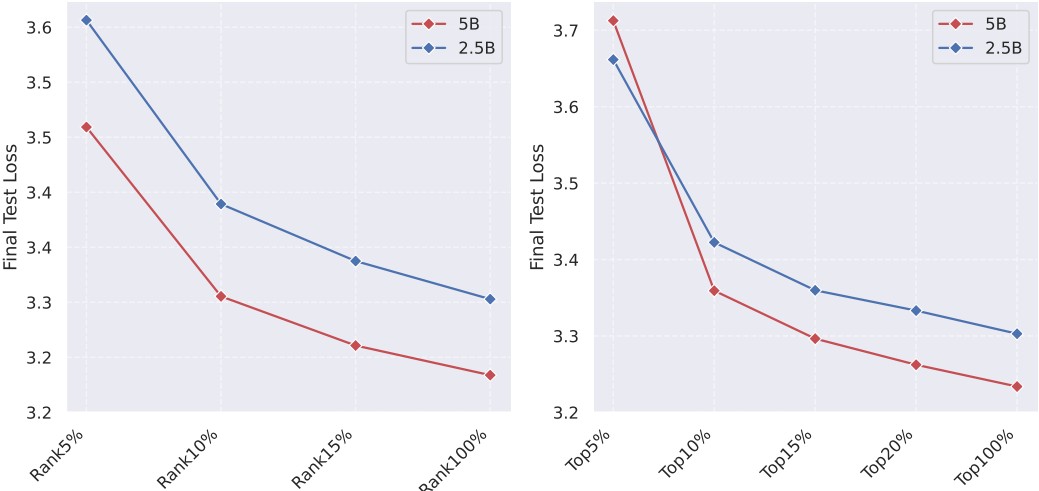

Figure 6: **Final test loss vs. compression parameter $K$.** Results are shown after processing 5B tokens (red) and 2.5B tokens (blue) for Rank$K$ (left) and Top$K$ (right) compressors. $K = 100\%$ corresponds to the non-compressed baseline. In the Top$K$ plot, the 2.5B setup outperforms 5B due to differences in scheduler behavior, as the runs execute a different number of steps.

We adopt the learning rate obtained from the sweep described in Section G.3 and use the same optimization and training setup as in the NanoGPT experiments. The LMO step scaling mechanism (Pethick et al., 2025b) is applied to ensure adaptivity across weight matrices of varying sizes.

The resulting convergence curves are shown in Figure 10. We observe qualitatively similar behavior to the NanoGPT setting: Natural compression achieves close-to-baseline loss while substantially reducing communication cost.

### G.7  BIDIRECTIONAL COMPRESSION

To complement the unidirectional compression experiments, we evaluate EF21-Muon in a fully bidirectional setup in which both server-to-worker and worker-to-server communication are compressed. We apply the Natural compressor in both directions. The training task matches the setup in Section G.5 (NanoGPT with a 2.5B token budget).

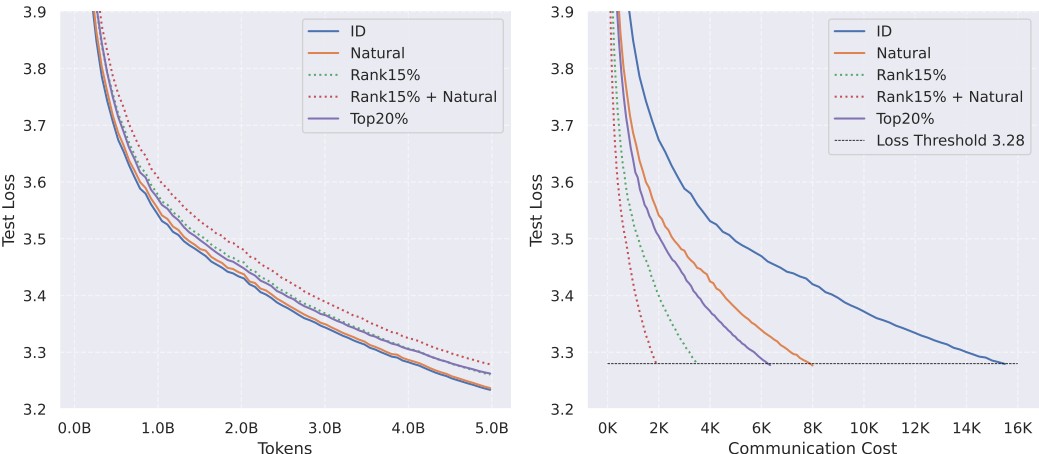

Figure 7: Left: **Test loss vs. # of tokens processed.** Right: **Test loss vs. # of bytes sent to the server from each worker** normalized by model size to reach test loss 3.28. Rank$X$%/Top$X$% = Rank$K$/Top$K$ compressor with sparsification level $X$%; ID = no compression.

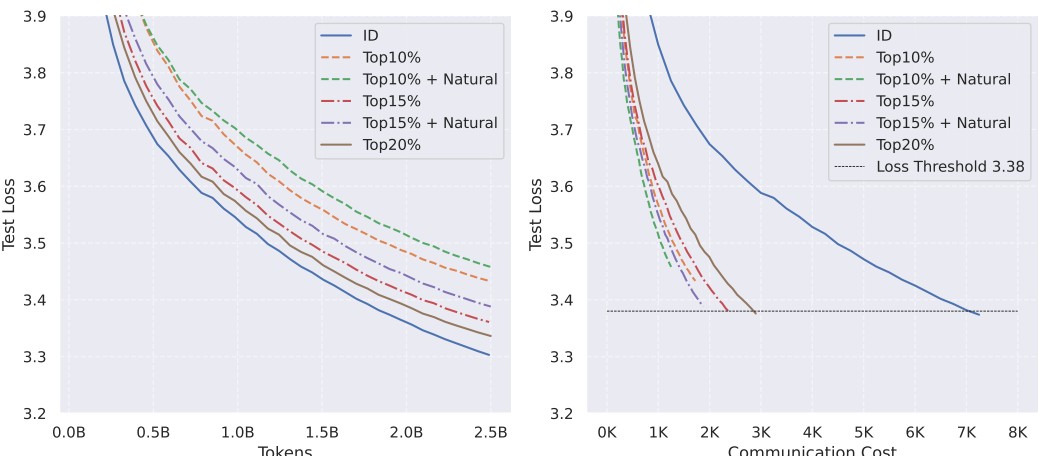

Figure 8: Left: **Test loss vs. # of tokens processed.** Right: **Test loss vs. # of bytes sent to the server from each worker** normalized by model size to reach test loss 3.38. Top$X$% = Top$K$ compressor with sparsification level $X$%; ID = no compression. "+ Natural" corresponds to applying Natural compression after Top$K$ compressor.

We follow the same hyperparameter selection protocol described in Section G.3. The corresponding learning rate sweep is shown in Figure 11. After tuning, we find that EF21-Muon remains effective in this more challenging bidirectional configuration, improving communication efficiency by approximately $2\times$ relative to the uncompressed baseline while achieving comparable convergence, as shown in Figure 12.

## G.8 LIMITATIONS

Reporting results for all compressors on the same token budget (for instance, 5B) and then measuring the prefix needed to reach a given loss threshold may not be fully consistent, as results can be affected by the scheduler. To mitigate this, we use a relatively strong loss threshold that ensures a significant number of tokens are processed beyond the constant learning rate phase. Additionally, tuning the initial learning rate can help stabilize the results.

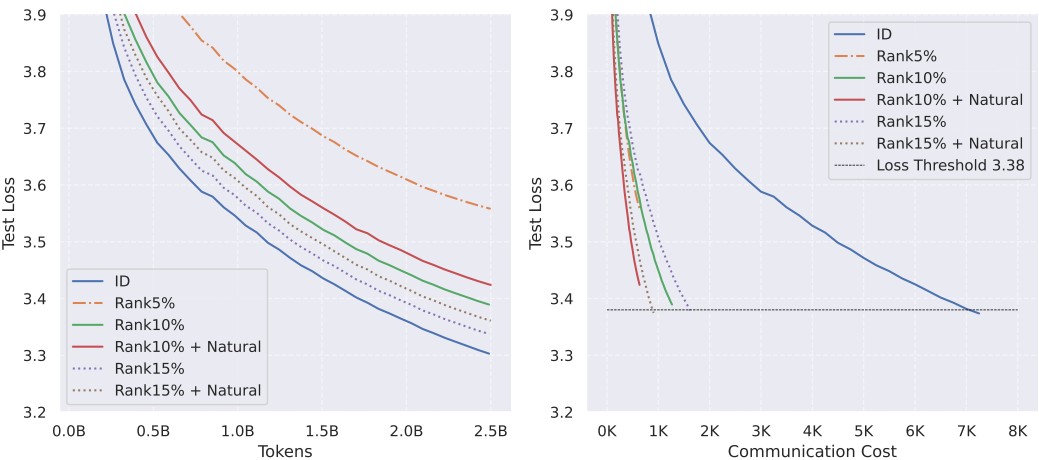

Figure 9: Left: **Test loss vs. # of tokens processed.** Right: **Test loss vs. # of bytes sent to the server from each worker** normalized by model size to reach test loss 3.38. Rank$X\%$ = Rank$K$ compressor with sparsification level $X\%$; ID = no compression. "+ Natural" corresponds to applying Natural compression after Rank$K$ compressor.

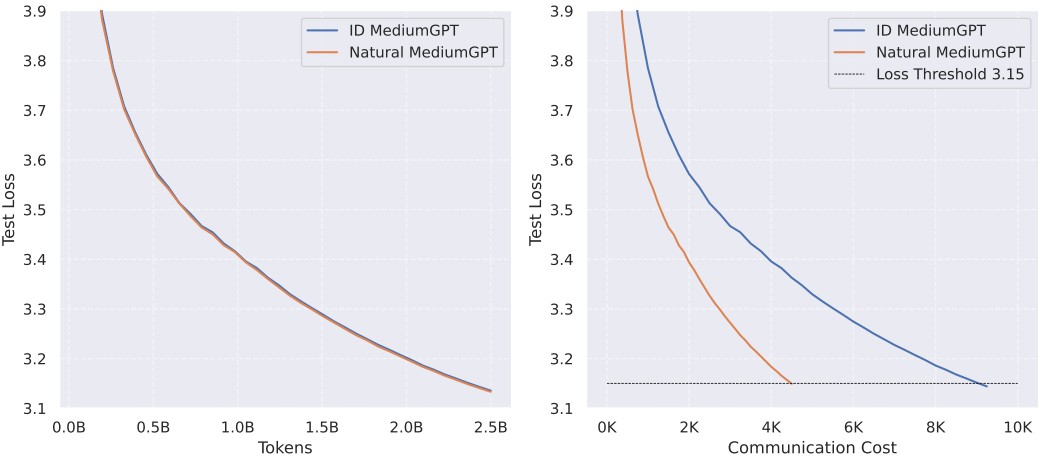

Figure 10: Left: **Test loss vs. # of tokens processed.** Right: **Test loss vs. # of bytes sent to the server from each worker** normalized by model size to reach test loss 3.15. ID = no compression.

**Note on LLM Usage.** Large Language Models were used to assist in polishing the writing of the manuscript. LLM assistance did not contribute to the scientific content of the paper.

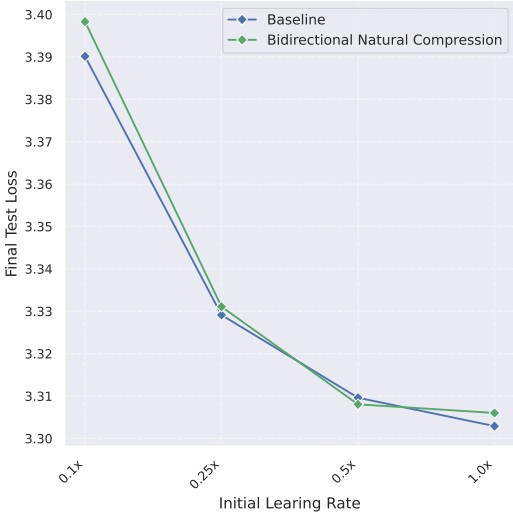

Figure 11: **Learning rate ablation for the bidirectional setup.** The grid spans from the optimal learning rate of the non-compressed baseline, $3.6 \times 10^{-4}$ (denoted as $1.0\times$), down to $0.1\times$.

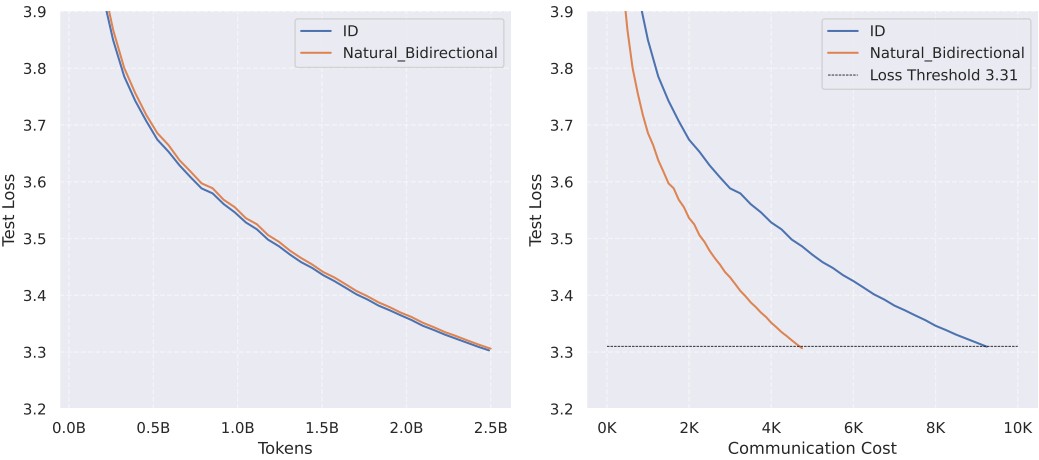

Figure 12: Left: **Test loss vs. # of tokens processed.** Right: **Test loss vs. # of bytes sent to the server from each worker** normalized by model size to reach test loss 3.31. ID = no compression. Both s2w and w2s directions are compressed using the Natural compressor.

