# OpenReview forum: "Error Feedback for Muon and Friends"
_ICLR.cc/2026/Conference — ICLR 2026 Poster_

### Official Review · Reviewer_VseT · 2025-10-16

**Soundness:** 3
**Presentation:** 2
**Contribution:** 3
**Rating:** 6
**Confidence:** 3

**Summary:**

The paper proposes EF21-Muon, which aims to be a communication-efficient distributed learning framework for the Muon family of optimizers. A key insight is to reformulate the LMO update using the sharp operator which recasts the update as a normalized steepest descent step. As Muon is steadily gaining popularity, this area of research is particularly timely and well-placed. Additionally, it is important to provide a principles framework for Muon, including in the setting of distributed training which suffers from communication inefficiencies.

**Strengths:**

1. The topic is particularly timely, and addresses the constraints of distributed training which is not typically encountered in centralized settings.

2. The experiment setup is quite relevant. I believe that NanoGPT on FineWeb is a good choice for optimizer evaluation in the context of model LMs.

**Weaknesses:**

1. Not much of a weakness, but the authors argue that due to increasing size, all training is distributed. This is true, but most frameworks tend to be parallelism (pipeline, model, etc) rather than the FL-type.

2. Experiments are conducted on a single model/scale (NanoGPT 124M). Demonstrating the effectiveness of EF21-Muon on other domains (e.g., ViT) or at a larger scale would make the claims of general applicability significantly more robust.

**Questions:**

Please see weaknesses. Not much questions to add.

---

> ### Author Response · Authors · 2025-11-21
>
> We would like to thank the reviewer for evaluating our submission. We appreciate the time and effort spent and are pleased that the review highlights our work's strengths. We will address each point raised and provide clarifications.
>
> ---
>
> **Weaknesses:**
> > Not much of a weakness, but the authors argue that due to increasing size, all training is distributed...
>
> Our statement was intended to highlight the need for distributed optimization as model sizes grow. Distributed optimization is not synonymous with federated learning; rather, federated learning represents a subset of distributed training. The workers/clients in EF21-Muon need not be mobile devices or edge clients, they could equally be GPUs in a data center cluster. EF21-Muon is agnostic to the specific distributed paradigm and applies broadly across both traditional parallelism-based and federated settings.
>
> > Experiments are conducted on a single model/scale (NanoGPT 124M)...
>
> We appreciate the reviewer's interest. Our experimental setup largely follows the established protocol used in prior Muon-related work [1,2], and we believe these experiments already provide strong validation of our theoretical claims. Nonetheless, given the reviewer's interest, we have conducted additional larger-scale experiments on GPT-Medium. These results are now included in Section G.6 of the revised paper.
>
> We would like to emphasize the primary aim of our work. This paper is fundamentally theoretical, and it should be evaluated as such. The main contribution is a new algorithmic framework along with a suite of state-of-the-art convergence guarantees (Theorems 3, 4, 5, 6, 14, 16, 19, 24). These results stand fully independently of any empirical evaluation. Even without experiments, they would remain strong and self-contained.
>
> That said, we have taken an additional step to demonstrate that our theory is reflected in practice: our empirical results show that the proposed method is indeed very promising. We are not an industrial lab, and industrial-scale experiments are not the focus of this work. Our goal is to provide rigorous theoretical foundations, complemented by a set of targeted experiments.
>
> We expect the benefits of our algorithm to carry over to other architectures and larger models, though confirming this empirically would require additional resources.
>
> [1] Pethick, Thomas, et al. "Training deep learning models with norm-constrained LMOs." arXiv preprint arXiv:2502.07529 (2025).
>
> [2] Riabinin, Artem, et al. "Gluon: Making Muon & Scion Great Again!(Bridging Theory and Practice of LMO-based Optimizers for LLMs)." arXiv preprint arXiv:2505.13416 (2025).
>
> ---
>
> We appreciate the recognition of the importance and timeliness of our work. We believe that our clarifications address all raised concerns. We would like to highlight that EF21-Muon is **the first communication-efficient, distributed Muon-type optimizer** with **state-of-the-art convergence guarantees and strong empirical performance**, filling an important gap in the literature. In light of this, we would be grateful if the reviewer could consider raising their score.

---

> > ### Comment · Reviewer_VseT · 2025-11-26
> >
> > Thank you for your response. My original review did not point out many significant weaknesses. The main issue would perhaps be the scale and scope of the experiments, and this weakness seems to have been flagged by all three reviewers, which the authors responded to by including a new experiment setting. The authors argue this is a theory paper, and this is not necessarily critical in my view. I will keep my borderline positive rating as the final score.

---

> > > ### Author Response · Authors · 2025-11-30
> > >
> > > Thank you for revisiting our submission. We would like to stress that **we have addressed all points raised across the reviews** and have updated the paper accordingly. Since no additional weaknesses were identified, we are unsure what further issues the present comment refers to. As noted previously, we believe the NanoGPT experimental evaluation is sufficient for supporting the theoretical contributions, and we have strengthened it further in response to reviewers' requests by adding larger-scale experiments.
> > >
> > > We hope these revisions make the contribution and its validation clear.

---

### Official Review · Reviewer_PPbv · 2025-10-31

**Soundness:** 3
**Presentation:** 2
**Contribution:** 3
**Rating:** 6
**Confidence:** 2

**Summary:**

This paper presents EF21-Muon, a communication-efficient distributed optimizer based on recent LMO optimizers with rigorous convergence guarantees. This method achieves up to 7x communications savings compared to uncompressed baselines.

Furthermore, a few notable insights include:
1. Bidirectional compression with error feedback
2. Support for non-Euclidean geometries through arbitrary norm choices
3. Layer-wise treatment of neural network parameters
4. Theoretical guarantees under many settings

**Strengths:**

1. Significant theoretical contribution, rigorous convergence analysis for compressed, distributed LMO-based methods in non-Euclidean settings.
2. Addresses a real bottleneck in the distributed training of large models, where Muon is gaining popularity.
3. Comprehensive theory, a very long appendix with many proofs.

**Weaknesses:**

1. Only one model size was trained, and it is very small by modern standards (120M).
2. Only one evaluation (Loss) was used. It's good practice to include a few downstream evaluations just in case for bold claims.
3. Compression overhead is not clearly addressed, topk requires transmitting indices and not thoroughly analyzed for the distributed setting with varying model architectures. This is especially important in optimization as a general-purpose optimizer should work on different model shapes.
4. Very heavy and dense notation, difficult to read.
5. The algorithm itself is not especially novel, error feedback comes from EF21.

**Questions:**

1. How does this method scale when applied to bigger models? How does wall-clock time scale as you increase the model size? Is it a constant factor increase or is it quadratic, log, etc?
2. How does EF21-Muon compare to other compressed distributed training variants of AdamW or SGD?
3. The method proposed has a lot of hyperparameters, how sensitive is the training to choices beyond the limited ablations shown?

---

> ### Author Response · Authors · 2025-11-21
>
> We would like to thank the reviewer for the thoughtful review, we appreciate the time you invested in reading our submission, and we are grateful that you recognized the strengths of our work. We would like to address each point raised in the review and provide clarifications.
>
> ---
>
> **Weaknesses:**
>
> > Compression overhead is not clearly addressed...
>
> Our paper already includes a dedicated section in the appendix (Appendix G.2, "Top$K$ compression details") that addresses precisely the topic of transmitting indices. We explicitly address the overhead of index transmission, which requires $\lceil \log_2(d) \rceil$ bits for a vector of dimension $d$. For the architectures evaluated in this work, 26 bits are sufficient to encode these indices (refer to Appendix G.2). Given the logarithmic dependence on vector size, this overhead remains negligible and does not present a scaling bottleneck, even for models using extremely large weight matrices.
>
> > Very heavy and dense notation, difficult to read.
>
> We recognize that the algorithm is not the simplest and involves several components, which requires introducing some notation. However, we did our best to present these elements as clearly as possible: we carefully explained each component, highlighted key connections using color-coding, and defined all the concepts in the main paper, with additional details provided in the appendix.
>
> The algorithm shown in the main paper is already a simplified version of the more general layer-wise version introduced and analyzed in the appendix. We intentionally focus on this simpler setup in the main text to keep the notation as reader-friendly as possible. Overall, we believe the exposition is clear and accessible (as acknowledged by Reviewers NH9Y and y1YB), but of course no presentation is perfect, and we would be happy to incorporate any specific suggestions the reviewer may have.
>
> > The algorithm itself is not especially novel, error feedback comes from EF21.
>
> We appreciate the comment. As with any research, our work builds upon prior contributions. However, **our results go substantially beyond what was previously known**. While error feedback itself is not new and has inspired a large body of follow-up work, our use of it is novel. The EF21 mechanism in its original form differs significantly from what we need for our non-Euclidean, layer-wise Muon-style updates. In particular, our method required (i) a non-trivial generalization to the non-Euclidean setting, leading us to introduce a **new class of non-Euclidean compressors** (Appendix D), (ii) an **extension to the layer-wise regime** (Appendix B), and (iii) combination of primal compression, dual compression, stochasticity and momentum.
>
> Our algorithm is far from a simple reuse of EF21, and it required substantial theoretical and algorithmic innovations. Composing all the components we rely on: non-Euclidean LMOs, compression, error feedback, layer-wise modeling, and distributed communication efficiency, is far from trivial, as discussed in Section 2 "The challenges of distributing the LMO". Many algorithmic ideas do not compose well together, and showing that this particular combination leads to a theoretically strong and empirically competitive method is a substantial novelty. EF21-Muon is the first communication-efficient, non-Euclidean LMO-based optimizer with rigorous convergence guarantees, providing the first efficient distributed implementation of the very promising Muon-type family.
>
> **Questions:**
> > The method proposed has a lot of hyperparameters, how sensitive is the training to choices beyond the limited ablations shown?
>
> EF21-Muon has the same hyperparameters as Muon, plus the compression level. Learning rate ablations across 12 different compression settings are provided in Appendix G.3, and compression level ablations can be found in Appendix G.4. Momentum is fixed to the standard value of $0.9$. We believe these experiments already demonstrate robustness of our algorithm, but if the reviewer wishes to see any particular additional ablations, we are open to incorporating them.

---

> > ### Author Response · Authors · 2025-11-21
> >
> > **Comments/questions about experiments:**
> >
> > We appreciate the reviewer's suggestion. Our current experimental setup follows the established protocol used in prior Muon-related work [1,2], and we believe these experiments already provide strong validation of our theoretical claims. Nonetheless, given the reviewer’s interest, we have conducted additional larger-scale experiments on GPT-Medium; these results are now included in Section G.6 of the revised paper.
> >
> > We would like to briefly emphasize the primary aim of our work. This paper is fundamentally theoretical, and it should be evaluated as such. The main contribution is a new algorithmic framework along with a suite of state-of-the-art convergence guarantees (Theorems 3, 4, 5, 6, 14, 16, 19, 24). These results stand fully independently of any empirical evaluation. Even without experiments, they would remain strong and self-contained.
> >
> > That said, we have taken an additional step to demonstrate that our theory is reflected in practice: our empirical results show that the proposed method is indeed very promising. We are not an industrial lab, and industrial-scale experiments are not the focus of this work. Our goal is to provide rigorous theoretical foundations, complemented by a set of targeted experiments.
> >
> > We agree that downstream evaluations can be interesting, but they are not the focus of this work. Our primary claim is that EF21-Muon achieves substantial communication savings over Muon, and our theoretical guarantees are stated in terms of the training loss. Consequently, the most relevant way to evaluate our method empirically is through the loss. Our experiments are intentionally aligned with our theory: they aim to validate the behaviors our analysis predicts. While broader experiments may be of independent interest, they do not relate to our theoretical contributions and are therefore outside the scope of this paper.
> >
> > [1] Pethick, Thomas, et al. "Training deep learning models with norm-constrained LMOs." arXiv preprint arXiv:2502.07529 (2025).
> >
> > [2]  Riabinin, Artem, et al. "Gluon: Making Muon \& Scion Great Again!(Bridging Theory and Practice of LMO-based Optimizers for LLMs)." arXiv preprint arXiv:2505.13416 (2025).
> >
> > ---
> >
> > We again thank the reviewer for recognizing the significance of our contribution. We hope that our responses clarify any doubts. Overall, EF21-Muon provides **the first communication-efficient distributed implementation of Muon-type optimizers** with **state-of-the-art convergence guarantees and strong empirical performance**, filling an important gap in the literature. We would be grateful if the reviewer could consider raising their score in light of these clarifications.

---

### Official Review · Reviewer_NH9Y · 2025-11-01

**Soundness:** 3
**Presentation:** 3
**Contribution:** 3
**Rating:** 8
**Confidence:** 3

**Summary:**

This work studies a distributed non-Euclidean LMO-based optimization method that generalizes and recovers Muon and Scion in the non-compressed regime. Specifically, the authors introduce EF21-Muon, a unified and communication-efficient algorithm that incorporates stochastic gradients, momentum, and bidirectional compression with error feedback, while encompassing several existing compressed methods as special cases. Furthermore, the authors present comprehensive convergence guarantees for multiple settings, including both deterministic and stochastic cases, as well as for non-Euclidean smooth and generalized non-Euclidean smooth functions, under both layer-wise and joint parameter treatments. Experimental results on nanoGPT under various worker compressors show significant communication savings for the proposed algorithm.

**Strengths:**

- This work provides rigorous and comprehensive convergence guarantees for a wide range of settings.
- It supports bidirectional compression for the smooth regime.
- The proposed algorithm further supports non-Euclidean contractive compressors, thus enhancing generality.
- The analysis of the algorithms is conducted in non-Euclidean norms.
- The proposed algorithms achieve the state-of-the-art convergence rates.
- The paper is clearly structured and easy to follow.
- The discussion of the contribution of each term in the convergence guarantees is insightful.

**Weaknesses:**

- The results in the generalized smoothness regime do not include primal compression.
- The experimental results could additionally include some other baseline algorithms for comparison to better contextualize performance of EF21-Muon.
- It may also have been beneficial to include wall-clock time experiments.

**Questions:**

- Can the authors elaborate a bit on why in the generalized smooth setup the convergence guarantees are established without primal compression? Can these results potentially be extended to that setting as well?
- Could the authors clarify how the combinations of compressors are applied in the experimental setup?

---

> ### Author Response · Authors · 2025-11-21
>
> We would like to thank the reviewer for the careful review. We appreciate the time you took to evaluate our submission and are glad that you recognized the strengths of our work. We address the points raised and provide clarifications below.
>
> ---
>
> **Weaknesses:**
> > The results in the generalized smoothness regime do not include primal compression.
>
> It is true that Theorems 3 and 5 explicitly include primal compression, while Theorems 4 and 6 do not. However, as we discuss in Section 4 and elaborate in Appendix D.1, **server-to-worker communication can still be made efficient through appropriate choice of norms**. Specifically, LMOs under certain norms naturally induce compression-like behavior, meaning that our theory already captures bidirectional communication compression in both the smooth and generalized smooth settings, even without explicitly applying compression operators.
>
> The reason we do not include explicit primal compression in Theorems 4 and 6 stems from theoretical complications associated with Assumption 4 in the context of bidirectional updates. Without primal compression, the assumption only needs to be applied with $X=X^{k+1}$ and $Y=X^k$, so the gradient on the right-hand side of the inequality depends on the $k$-th iterate. However, introducing primal compression requires applying the assumption with $X=X^{k+1}$ and $Y=W^{k+1}$, introducing $X^{k+1}$ or $W^{k+1}$ on the right-hand side. This shift from dependence on the $k$-th iterate to the $(k+1)$-th iterate cannot be handled by our current proof technique. While it may be possible to extend these convergence guarantees using an alternative proof approach, this is beyond the scope of the present work and is left for future work.
>
> Finally, we note that most prior works on compressed optimization do not consider primal compression at all.
>
> > Experimental results
>
> We appreciate the reviewer's suggestion. We would like to briefly emphasize the primary aim of our work. This paper is fundamentally theoretical, and it should be evaluated as such. The main contribution is a new algorithmic framework along with a suite of state-of-the-art convergence guarantees (Theorems 3, 4, 5, 6, 14, 16, 19, 24). These results stand fully independently of any empirical evaluation. Even without experiments, they would remain strong and self-contained.
>
> We agree that additional evaluations can be interesting, but they are not the focus of this work. Our primary claim is that EF21-Muon achieves substantial communication savings over Muon. Consequently, the most relevant way to evaluate our method empirically is through the comaprison of EF21-Muon and standard Muon in terms of communication costs. Our experiments are intentionally aligned with our theory: they aim to validate the behaviors our analysis predicts. While broader experiments may be of independent interest, they do not relate to our theoretical contributions and are therefore outside the scope of this paper.
>
> We also appreciate this comment regarding computational efficiency. In our setup, the Natural and Top$K$ compressors incur negligible computational overhead. We acknowledge that Rank$K$ compressors are more computationally intensive, resulting in a 30–50\% overhead due to the SVD computations. This overhead could potentially be reduced by replacing the standard PyTorch SVD implementation with a more efficient approach, such as the PowerSGD method.
>
> **Questions:**
> >Can the authors elaborate a bit on why in the generalized smooth setup the convergence guarantees are established without primal compression?
>
> Please, see our response above.
>
> > Could the authors clarify how the combinations of compressors are applied in the experimental setup?
>
> We evaluate two different compression configurations: Top$K$+Natural and Rank$K$+Natural. In the Top$K$+Natural setup, we apply a Top$K$ operator to the original tensor to obtain the values and indices arrays. The Natural compressor is subsequently applied only to the values array. In the Rank$K$+Natural setup, we perform SVD, and take the top r components to obtain $U_r$, $S_r$ and $V_r$. We then apply the Natural compressor to each component of this decomposition. A high-level overview of these methods is provided on page 9, and we will include a more detailed formulation in the appendix.
>
> ---
>
> We thank the reviewer again for their positive assessment and encouraging feedback. We are glad that our contributions were well received and greatly appreciate the time and effort taken to evaluate our work. Please do not hesitate to let us know if you have any further questions.

---

> > ### Comment · Reviewer_NH9Y · 2025-11-26
> > **Thank you for the detailed response**
> >
> > I thank the authors for their response and clarifications. I have also read the other reviews and the corresponding author responses. I continue to believe that the paper’s contribution is meaningful, and I maintain my recommendation for acceptance, although I note that I am not an expert in distributed optimization.

---

### Official Review · Reviewer_y1YB · 2025-11-03

**Soundness:** 3
**Presentation:** 3
**Contribution:** 2
**Rating:** 4
**Confidence:** 3

**Summary:**

This paper introduces a distributed optimizer that brings error‑feedback (EF) to layer‑wise linear minimization oracle (LMO) methods such as Muon/Scion/Gluon, extending EF beyond the Euclidean setting and enabling bidirectional compression with support for stochastic gradients and momentum.

**Strengths:**

The paper establishes non‑Euclidean EF convergence for LMO‑based methods, and shows how rates recover Euclidean EF21 results when specialized, which is a nontrivial generalization.

On NanoGPT-124M trained on FineWeb, they report up to 7x reduction in worker‑to‑server communication without loss of accuracy.

**Weaknesses:**

The novelty is rather incremental since EF21-P introduces bidirectional EF with momentum/stochasticity, Gluon introduces non‑Euclidean layer‑wise LMO analyses, and Dion introduces distributed Muon‑style optimizers.

Experiments show that compression reduces uplink bytes but degrades token efficiency and final loss at a fixed 5B budget, so the claim of "no accuracy degradation" is unsupported under equal‑budget comparisons.

The convergence results assume exact LMO steps, yet the implementation uses inexact Newton–Schulz updates. Recent literature emphasizes that this gap matters, but the paper neither analyzes nor bounds the approximation error.

**Questions:**

Results are on NanoGPT‑124M, but how would the conclusions change for billion‑parameter LLMs and larger clusters?

The main results and plots focus on w2s, but how is the total communication cost including s2w, and what are the gains from s2w compression?

---

> ### Author Response · Authors · 2025-11-21
>
> We would like to thank the reviewer for their thoughtful feedback. We appreciate the time you took to evaluate our submission and are glad that the review highlights our work's strengths. We address the points raised and provide clarifications below.
>
> ---
> **Weaknesses:**
> > The novelty is rather incremental...
>
> We appreciate the comment. As with any research, our work builds upon prior contributions. However, as the reviewer noted in the Strengths section, our results are far from trivial and extend well beyond what was previously known. Below, we address each of the referenced works in more detail to clarify the conceptual and technical novelty of our EF21-Muon:
>
> - EF21-P, introduced in [1], is fundamentally a server-to-worker compression mechanism. It does not itself incorporate bidirectional compression, momentum, or stochasticity. Although one can combine EF21-P with additional modules to produce more complex methods, the mechanism in its original form differs significantly from what we need for our non-Euclidean, layer-wise Muon-style updates. In particular, our server-to-worker mechanism required (i) non-trivial generalization to the non-Euclidean setting, requiring a new class of compressors, and (ii) adaptation to operate layer-wise (detailed in the appendix). This is not a direct reuse of EF21-P; even this single component required substantial theoretical and algorithmic development.
>
> - We agree that Gluon [2] was the first to introduce the layer-wise perspective. However, Gluon is *not* a distributed algorithm. It does not address communication constraints and worker/server bandwidth limitations, which are the main challenges we address. In contrast, EF21-Muon is explicitly designed for multi-worker scenarios where communication is the main bottleneck. Our compression framework provides a principled way to remove these bottlenecks while retaining strong theoretical convergence guarantees. Thus, while Gluon is valuable as theory for single-machine LMO optimizers, **EF21-Muon applies to an entirely different class of problems**.
>
> - Dion [3] is indeed a distributed Muon-style optimizer, but it is algorithmically unrelated to EF21-Muon. Dion relies on low-rank approximations in its orthonormalized updates. We take a completely different approach using non-Euclidean layer-wise LMOs and EF-based compression. Importantly, existing distributed Muon variants (including Dion) were designed based on heuristics and did not come with any theoretical guarantees. As we emphasize in the introduction, this gap between practice and theory was an open problem. Our work is **the first to introduce a communication-efficient, non-Euclidean, LMO-based distributed optimizer with both rigorous state-of-the-art convergence guarantees, and strong empirical performance**. Moreover, Dion's updates operate only with spectral-norm balls, while our framework handles arbitrary norms and is thus more general.
>
> Finally, **combining all these components: non-Euclidean LMOs, compression, error feedback, layer-wise modeling, and distributed communication efficiency, is highly nontrivial**, as we discuss in Section 2, paragraph "The challenges of distributing the LMO". Many algorithmic ideas do not compose well together, and showing that this particular combination leads to a theoretically strong and empirically competitive method is a substantial novelty. EF21-Muon is **the first efficient distributed implementation of the very promising Muon-type family**.
>
> [1] Gruntkowska, Kaja, et al. "EF21-P and friends: Improved theoretical communication complexity for distributed optimization with bidirectional compression." International conference on machine learning. PMLR, 2023.
>
> [2] Riabinin, Artem, et al. "Gluon: Making Muon & Scion Great Again! (Bridging Theory and Practice of LMO-based Optimizers for LLMs)." arXiv preprint arXiv:2505.13416 (2025).
>
> [3] Ahn, Kwangjun, et al. "Dion: Distributed orthonormalized updates." arXiv preprint arXiv:2504.05295 (2025).

---

> > ### Author Response · Authors · 2025-11-21
> >
> > > Experiments show that compression reduces uplink bytes but degrades token efficiency and final loss...
> >
> > As we mention in the experimental section, it is expected that lossy compression slows progress when measured *in terms of number of training steps*. Entirely "free" compression at arbitrary levels would be rather surprising. However, training steps are *not* the metric that matters here. What matters is *communication cost*, and this is precisely where EF21-Muon shines. EF21-Muon dramatically reduces per-step communication (Table 2), which is the main bottleneck in this regime. The modest reduction in per-iteration progress is more than compensated by the much lower per-iteration communication cost. As a result, **EF21-Muon reaches the same target loss in significantly fewer communication-equivalent units**, as demonstrated in the right panels of Figures 1, 4, 5, 7, 8, 9, and 10. Moreover, with less aggressive compression, even the plots against token count show that EF21-Muon performs virtually identically to the uncompressed baseline (Figures 10 and 12).
> >
> > Overall, our claim of "no accuracy degradation" is well supported.
> >
> > > The convergence results assume exact LMO steps, yet the implementation uses inexact Newton–Schulz updates...
> >
> > We acknowledge that in practice one typically uses inexact Newton–Schulz updates, and these approximations can indeed have an effect. However, our work follows the standard assumption used in nearly all prior theoretical papers in this area, namely, exact LMO computations. This assumption is deliberate: it provides the cleanest framework for isolating the core algorithmic behavior we aim to study. In our case, the central phenomena are the effects of distributed and compressed optimization. Assuming exact LMOs allows us to focus precisely on these aspects without introducing additional noise that would obscure the main insights. We do not believe the absence of an inexact analysis is a weakness. Our contributions target foundational questions and advance an established line of research that has consistently used the exact LMO assumption.
> >
> > To our knowledge, the only theoretical paper analyzing inexact updates for standard Muon is [4], which appeared on arXiv after the ICLR submission deadline (October 22). While incorporating inexactness is an interesting and important direction for future exploration, it deserves a separate line of work and lies beyond the scope of this paper.
> >
> > [4] Shulgin, Egor, et al. "Beyond the Ideal: Analyzing the Inexact Muon Update." arXiv preprint arXiv:2510.19933 (2025).

---

> > > ### Author Response · Authors · 2025-11-21
> > >
> > > **Questions:**
> > > > Results are on NanoGPT‑124M, but how would the conclusions change for billion‑parameter LLMs and larger clusters?
> > >
> > > We appreciate the reviewer's interest. Our experimental setup largely follows the established protocol used in prior Muon-related work [1,2], and we believe these experiments already provide strong validation of our theoretical claims. Nonetheless, given the reviewer's interest, we have conducted additional larger-scale experiments on GPT-Medium. These results are now included in Section G.6 of the revised paper.
> > >
> > > We would like to emphasize the primary aim of our work. This paper is fundamentally **theoretical**, and it should be evaluated as such. The main contribution is a new algorithmic framework along with a suite of state-of-the-art convergence guarantees (Theorems 3, 4, 5, 6, 14, 16, 19, 24). These results stand fully independently of any empirical evaluation. Even without experiments, they would remain strong and self-contained.
> > >
> > > That said, we have taken an additional step to demonstrate that our theory is reflected in practice: our empirical results show that the proposed method is indeed very promising. We are not an industrial lab, and industrial-scale experiments are not the focus of this work. Our goal is to provide rigorous theoretical foundations, complemented by a set of targeted experiments.
> > >
> > > We expect the benefits of our algorithm to carry over to even larger models, though confirming this empirically would require additional resources.
> > >
> > > [1] Pethick, Thomas, et al. "Training deep learning models with norm-constrained LMOs." arXiv preprint arXiv:2502.07529 (2025).
> > >
> > > [2]  Riabinin, Artem, et al. "Gluon: Making Muon \& Scion Great Again!(Bridging Theory and Practice of LMO-based Optimizers for LLMs)." arXiv preprint arXiv:2505.13416 (2025).
> > >
> > > > The main results and plots focus on w2s, but how is the total communication cost including s2w, and what are the gains from s2w compression?
> > >
> > > We thank the reviewer for the suggestion. In response, we have added experiments using bidirectionally compressed EF21-Muon in Section G.7. These results show that the algorithm remains effective under this setting, improving communication efficiency by approximately a factor of 2 compared to the uncompressed baseline, while maintaining comparable convergence speed.
> > >
> > > ---
> > > We thank the reviewer again for acknowledging the significance of our contribution. We hope our responses clarified any concerns. EF21-Muon delivers **the first communication-efficient distributed implementation of Muon-type optimizers** with **state-of-the-art convergence guarantees** and **strong empirical performance**. As such, it fills a clear and important gap in the existing literature.
> > >
> > > Since we have addressed all concerns raised and the work's novelty and importance are now clearly evident, we would be grateful if the reviewer could update their score to reflect these strengths.

---

### Meta-Review · Area_Chair_D53u · 2025-12-24

**Summary:**

This paper considers the problem of designing communication-efficient distributed optimizer for non-Euclidean LMO-based training and the authors propose an EF21-Muon algorithm for this purpose. Both theoretical convergence analysis and preliminary numerical validations are done in the paper.

After the rebuttal, most of the concerns from the reviewers are resolved or justified. Yet there still remain a few issues that are, according to the AC's reading, not well justified. In particular, several reviewers mention that the experiments are only on a small model NanoGPT‑124M, it is not clear for larger LLMs and larger clusters. The authors did add the 355M model in the revision, but is still a small model, this does not fully resolve the reviewers' questions on experiments. However, the AC understand that the rebuttal period may not be sufficient to add extra experiments on larger models, and the AC believes that the paper’s contribution is meaningful.

Combining all the feedback from the reviewers and discussion in the rebuttal, the AC thinks that the paper is still marginally above the accept threshold. And the AC would like to suggest an accept to the paper.

**Reviewer Concerns:**

Addressed or justified:
(1) PPbv: Compression overhead is not clearly addressed (addressed).
(2) y1YB & PPbv: The novelty is rather incremental (partially justified).
(3) y1YB: The gap between LMO step and N-S update (justified).

Not Addressed:
(1) y1YB & PPbv & VseT: Experiments are only on a small model NanoGPT‑124M, it is not clear for larger LLMs and larger clusters (partially resolved: the authors do add the 355M model in the revision, but is still a small model, this does not really resolve the reviewers' questions on experiments).

**Reviewer Scores:**

With the response from the authors, the score from y1YB may slightly increase to 5, according to the AC's understanding. The AC thinks it may not increase to 6 because the reviewer's concerns are partially but not fully addressed. For the other reviewers, their score will most likely remain unchanged.

Overall, the AC think's the paper will still be marginally above the threshold.

---

### Decision · Program_Chairs · 2026-01-26

Accept (Poster)